# SafeOR-Gym: A Benchmark Suite for Safe Reinforcement Learning Algorithms on Practical Operations Research Problems

## Abstract

Most existing safe reinforcement learning (RL) benchmarks focus on robotics and control tasks, offering limited relevance to high-stakes domains that involve structured constraints, mixed-integer decisions, and industrial complexity. This gap hinders the advancement and deployment of safe RL in critical areas such as energy systems, manufacturing, and supply chains. To address this limitation, we present SafeOR-Gym, a benchmark suite of nine operations research (OR) environments tailored for safe RL under complex constraints. Each environment captures a realistic planning, scheduling, or control problems characterized by cost-based constraint violations, planning horizons, and hybrid discrete-continuous action spaces. The suite integrates seamlessly with the Constrained Markov Decision Process (CMDP) interface provided by OmniSafe. We evaluate several state-of-the-art safe RL algorithms across these environments, revealing a wide range of performance: while some tasks are tractable, others expose fundamental limitations in current approaches. SafeOR-Gym provides a challenging and practical testbed that aims to catalyze future research in safe RL for real-world decision-making problems.

## 1 Introduction

Real-world reinforcement learning (RL) applications often demand that agents respect strict safety requirements at all times. *Safe reinforcement learning* (Garcıa & Fernández, 2015; Gu et al., 2022) addresses this need by maximizing long-term rewards while satisfying safety constraints. In contrast to standard RL, which might treat safety violations as mere negative rewards, safe RL explicitly enforces constraints throughout training and deployment. This capability is vital in safety-critical domains such as autonomous driving, robotics, and power systems, where an agent's actions can lead to irreversible damage or hazards if constraints such as speed limits, stability margins, or resource capacities are violated. The formalism of constrained Markov decision processes (CMDPs) (Altman, 2021) provides a natural framework for such problems, requiring the learned policy to remain within a set of safe outcomes at all times. In practice, safe RL algorithms incorporate cost signals or penalties for unsafe behavior and aim to ensure constraint satisfaction during both learning and execution. This paradigm has gained traction as a cornerstone for deploying RL in high-stakes environments.

Progress in safe RL has been accelerated by the development of specialized libraries and benchmark suites. A prominent example is *OmniSafe* (Ji et al., 2024), an open-source infrastructure designed specifically for safe RL research. OmniSafe offers a unified, modular framework with built-in support for constraint handling and a comprehensive collection of constrained RL algorithms. Notably, it extends the standard OpenAI Gym interface by supporting constrained Markov decision processes (CMDPs), enabling explicit modeling of safety constraints through cost signals. These capabilities make OmniSafe and similar frameworks highly valuable for evaluating safe RL methods in a standardized and extensible setting. However, despite advances in algorithms and software infrastructure, the diversity and realism of benchmark environments tailored for safe RL remain limited.

Most existing RL benchmarks were not designed with safety constraints in mind. Classic control tasks and popular continuous control domains such as CartPole, Pendulum, and MuJoCo locomotion simulations (Todorov et al., 2012; Brockman et al., 2016) offer simple dynamics with no explicit

safety constraints. Crucially, these environments rarely involve the mixed discrete–continuous decision-making or constraints that characterize real operational problems. Even specialized safe RL environment suites focus primarily on robotic control and do not capture the structured complexity of industrial decision problems. This gap in benchmarks makes it challenging to rigorously evaluate how well safe RL algorithms would perform on realistic, safety-critical tasks that involve complex constraints and decision structures.

Operations research (OR) problems (Rardin & Rardin, 1998) present an appealing opportunity to fill this gap. OR encompasses a broad class of decision-making tasks with rich combinatorial structure, explicit constraints, and often long planning horizons. These problems inherently require balancing long-term objectives with immediate feasibility and safety. For instance, planning the operation of an energy storage system demands making multi-year investments and dispatch decisions that remain robust to short-term operational constraints and rare events(Ramanujam & Li, 2025; Li et al., 2022). In general, OR formulations like scheduling, resource allocation, and supply chain management force agents to respect resource capacities, timing deadlines, and logical constraints at every step, exactly the kind of requirements safe RL is meant to handle. Moreover, OR problems frequently arise in safety-critical domains such as energy systems, transportation, supply chains, and chemical process operations. In these settings, violating a constraint such as overloading a network, missing a maintenance schedule, running a process outside safe limits, etc., can lead to severe real-world consequences. The structured nature and practical relevance of OR tasks make them well-suited as benchmark environments to stress-test safe RL algorithms on realistic problems that go beyond the toy examples commonly used.

We introduce SafeOR-Gym, a benchmark suite for safe reinforcement learning that features a diverse set of operations research environments spanning industrial planning and real-time control. The suite includes nine environments with varying structures, time horizons, and decision complexities.

In summary, this work makes the following contributions:

1. We develop a suite of nine OR-based benchmark environments tailored for safe RL, addressing the need for more realistic and structured evaluation tasks.

2. We release Gym-compatible implementations of these environments with native integration into the OmniSafe framework, enabling out-of-the-box use of constraint-handling algorithms.

3. We evaluate the environments using several on-policy algorithms in OminiSafe and discuss the current limitations of existing safe RL algorithms in solving highly constrained OR problems.

## 2 RELATED WORK

Standard RL benchmarks, such as the OpenAI Gym toolkit (Brockman et al., 2016) with classic control and MuJoCo-based continuous control tasks, have been foundational for evaluating reinforcement learning algorithms. However, they typically lack built-in safety constraints or the structured decision-making found in operations research (OR) problems. Most environments are designed as simplified tasks with minimal realism or constraints.

To address the need for safety-aware evaluation, specialized benchmark libraries have emerged. OpenAI's *Safety Gym* (Ray et al., 2019) introduced a set of environments that simulate continuous control tasks with hazards, cost signals, and explicit safety constraints. More recently, the *Safety Gymnasium* (Ji et al., 2023) suite has extended this idea to include both single- and multi-agent safety-critical environments. The *OmniSafe* (Ji et al., 2024) framework further consolidates these contributions by integrating a variety of safe RL environments and algorithms into a unified and modular platform. It supports constraint-handling algorithms, constrained policy optimization methods, and provides compatibility with Gym-style APIs.

In parallel, a growing body of work has focused on bringing OR and classical control problems into Gym-compatible environments. *OR-Gym* (Hubbs et al., 2020) provides a library of OR formulations, such as knapsack, bin packing, and supply chain management, as RL tasks, enabling comparison between RL policies and traditional optimization techniques. *PC-Gym* (Bloor et al., 2024) and (Park et al., 2025) introduce environments that model realistic chemical process dynamics, control

disturbances, and enforce operational safety. *SustainGym* (Yeh et al., 2023) contributes several sustainability-focused environments such as electric vehicle charging and data center scheduling, which feature complex operational constraints, distribution shifts, and hybrid discrete-continuous action spaces.

Despite these efforts, existing benchmarks for operations research problems exhibit important limitations. Existing works are only compatible with standard Gymnasium environments, rather than the CMDP interface required by frameworks like OmniSafe. They typically handle constraint violations by penalizing the reward function, rather than modeling them as explicit cost signals. Furthermore, many environments simplify or abstract away industrial complexity, limiting their utility for evaluating algorithms in realistic, safety-critical scenarios. *OR-Gym*, for example, primarily consists of toy problems such as knapsack and bin packing, which lack the structural and constraint richness of real-world applications. *SustainGym* centers on sustainability-oriented problems, but does not incorporate the nonlinear nonconvex constraints in many OR applications, such as those found in the blending problem. These gaps motivate the development of a benchmark suite featuring rich, constrained, and practically relevant OR problems that can serve as a rigorous testbed for safe RL algorithms. These comparisons are summarized in Table 1. A more detailed comparison of the environments is show in Appendix C.

Table 1: Comparison of Gym environments with operations research applications

(a) Environment class, constraints, and SafeRL compatibility

| Work | Env. Class | Constraint Handling | SafeRL |
|------|-----------|---------------------|--------|
| OR-Gym | Gymnasium | truncation, reward penalties | × |
| SustainGym | Gymnasium | truncation, reward penalties | × |
| SafeOR-Gym | Gymnasium + CMDP | truncation, reward penalties, explicit costs | ✓ |

(b) problem size, constraint complexity, and application domains

| Work | Obs / Action (mean, max) | Nonconvex | Domain |
|------|--------------------------|-----------|--------|
| OR-Gym | (242, 2501) / (57, 200) | × | classical OR |
| SustainGym | (79, 150) / (33, 72) | × | sustainable energy |
| SafeOR-Gym | (86, 4280) / (32, 272) | ✓ | planning, power, chemical, scheduling |

## 3 ENVIRONMENTS

This section provides an overview of two illustrative examples of the environments in *SafeOR-Gym*. The detailed descriptions of all the environments including the mathematical details, explanations, and illustrative figures can be found in Appendix A.

### 3.1 MULTIPERIOD BLENDING PROBLEM (BLENDINGENV)

**Problem Description** The multiperiod blending problem arises in industries such as refining and chemical processing, where raw materials with different properties must be blended over time to produce saleable products meeting strict quality constraints (Chen & Maravelias, 2020). In BlendingEnv, the agent decides how much of each source stream to purchase, how to route flows through a network of blenders, and how much product to sell at each time step. Safety constraints include property bounds on final products and storage limits on inventories. The environment reflects operational complexities such as nonlinear blending effects, capacity limits, and quality enforcement.

**State Space** The state includes current inventory levels of sources, blenders, and demand nodes; material properties of blender; future availability of sources and product demand over a lookahead window, and the current time step.

**Action Space** The action specifies: source stream purchase quantities; product sale quantities; flow rates from source inventories to blenders, between blenders, and from blenders to product inventories.

**Transition Dynamics** Updates inventories using material balances; clips values to inventory bounds; updates material properties in blenders based on flow composition; rolls forward source availability and demand forecasts. The dynamics of mixing material properties involve nonlinear nonconvex constraints.

**Cost** The environment penalizes: violating inventory bounds; violating the no simultaneous "in-out" flow rule for blenders; and producing out-of-spec blends.

**Reward** The reward includes: revenue from product sales; minus cost of purchasing source streams; minus variable and fixed costs for flow operations.

## 3.2 Integrated Scheduling and Maintenance (SchedMaintEnv)

**Problem Description** The Integrated Scheduling and Maintenance environment models the daily operation of compressors in an Air Separation Unit (ASU), where gaseous product demand must be met without inventory buffers (Xenos et al., 2016). The agent must coordinate production and maintenance actions for each compressor while optionally procuring external supply to satisfy demand. At each time step, the agent decides the fraction of each compressor's capacity to operate, whether to initiate maintenance, and how much external product to purchase. Maintenance policies are constrained by compressor-specific conditions, such as mean time to failure (MTTF), maintenance duration, and cooldown periods. Note that the machine failures can also be uncertain. Safety arises from the need to avoid compressor breakdowns due to delayed maintenance, premature maintenance interventions, or ramping during repair periods.

**State Space** The state includes: forecasted product demand and electricity prices over a fixed horizon; compressor-level indicators for time since last maintenance, time remaining to complete maintenance, and eligibility to enter maintenance.

**Action Space** The action at each time step consists of: a binary vector for maintenance decisions (schedule or not); a continuous vector for compressor production rates; and a continuous external purchase action representing the fraction of a predefined maximum external capacity.

**Transition Dynamics** Compressor state evolves based on maintenance initiation, progress, and cooldown periods. Upate electricity prices and product demand follow forecasts. Maintenance eligibility resets once a cooldown period has elapsed. The environment enforces repair continuity and halts production during repair.

**Cost** The environment penalizes: early maintenance actions; failure to perform maintenance before MTTF; ramping while under maintenance; and disruption of ongoing maintenance. Additionally, unmet or overmet demand incurs a penalty proportional to the deviation from forecast.

**Reward** The agent receives a negative reward equal to the total cost incurred: production cost (based on power price and compressor load), external purchase cost.

## 4 Implementation, Compatibility, and Extensibility

All environments are implemented on top of the Gymnasium API (Brockman et al., 2016), with an additional Constrained Markov Decision Process (CMDP) wrapper (Ji et al., 2024). This wrapper consists of fewer than 50 lines of code but provides compatibility with Safe RL algorithms that explicitly handle constraints. While OmniSafe is used as the primary reference implementation due to its breadth and active development, SafeOR-Gym can be adapted to other Safe RL libraries with only minimal modifications, since most Safe RL algorithms are CMDP-based and rely on Gymnasium as the base environment class.

Each environment in SafeOR-Gym is initialized with a small illustrative instance, but the underlying data structures are independent of any specific problem setup. Instance-specific parameters such as network topologies, renewable generation profiles, or equipment characteristics are stored in external JSON files. This separation of code and data makes it straightforward to create new problem instances or modify existing ones by editing the JSON inputs, without altering the environment code. As a

result, stochastic elements such as randomized demand realizations or sampled renewable generation profiles can be introduced naturally by providing alternative data inputs.

Although the present work focuses on deterministic environments, SafeOR-Gym is designed to accommodate stochasticity and non-stationarity. This is particularly important for real-world OR problems, which often involve renewable generation variability, equipment failures, or evolving market conditions. As a demonstration, we extended the maintenance scheduling environment to incorporate random machine failures, producing `SchedMaintEnv-v1`, a stochastic variant of the original deterministic environment (`SchedMaintEnv-v0`). Importantly, this extension was achieved by inheriting the original environment rather than reimplementing it from scratch, underscoring the ease of extending SafeOR-Gym to uncertainty-aware benchmarks. In this work, we focus primarily on deterministic environments because most existing safe RL algorithms already face substantial challenges in solving even deterministic OR problems with complex constraints, which will be shown in section 5. Demonstrating these limitations in a controlled deterministic setting helps establish a clear baseline before introducing additional sources of uncertainty.

## 5 EXPERIMENTS

We evaluate a suite of safe reinforcement learning (RL) algorithms across multiple environments subject to safety constraints. Each experiment involves one or more deterministic case studies, designed to test algorithmic robustness and generalization. In addition, we also include `SchedMaintEnv-v1`, a stochastic variant of the original deterministic environment (`SchedMaintEnv-v0`). We adopt the CMDP formulation used in the OmniSafe package (Ji et al., 2024). We adapt the safe RL algorithms in OmniSafe, which supports parallelized training and evaluation via its `Experimental Grid` feature. The safe RL algorithms tested include Constrained Policy Optimization (CPO) (Achiam et al., 2017), the Lagrangian version of trust regions policy optimization (TRPOLag) (Ray et al., 2019), Penalized Proximal Policy Optimization (P3O) (Zhang et al., 2022), Constraint Rectified Policy Optimization (OnCRPO) (Xu et al., 2021), the Lagrangian version of Deep Deterministic Policy Gradient (DDPGLag) (Lillicrap et al., 2019), First Order Constrained Optimization in Policy Space (FOCOPS) (Zhang et al., 2020), PID version of SACLag (SACPID) (Stooke et al., 2020), and the Lagrangian version of Soft Actor-Critic (SAC) algorithm (SACLag) (Haarnoja et al., 2018). All the experiments were conducted on AWS servers using g4dn.xlarge instances equipped with NVIDIA T4 GPUs, providing sufficient computational resources for training and evaluation of all the algorithms. The training times for the various algorithms across different experiments are provided in the supplementary material.

### 5.1 RESULTS AND DISCUSSION

We benchmark each algorithm's performance across environments during both training and evaluation. Figure 10 presents the average reward and cost per training epoch, with shaded regions indicating one standard deviation around the mean. Table 2 summarizes evaluation results, averaged over 10 episodes, for a representative subset of experiments and environments. The standard deviation of the evaluation rewards and costs is generally small in the deterministic environments, indicating consistent performance in the deterministic environments.

**Optimal reward of the environments:** One of the advantages of benchmarking safe RL algorithms using environments based on deterministic opeartions research problems is that state-of-the-art optimization solvers such as Gurobi (Gurobi Optimization, LLC, 2025) can be used to solve the nonconvex problems to global optimality while strictly enforcing all the constraints. The optimal reward from the optimization solvers can be seen as the "ground truth" of the envrionments, shown in the first column of Table 2. The optimal reward of `SchedMaintEnv-v1` is obtained by assuming a "perfect information" lookahead using the Gurobi solver, which provide a theoretical upper bound of the optimal expected reward.

**Evaluation Criteria:** To evalute the safe RL algorithms, for each environment, we identify the best- and worst-performing algorithms using a systematic selection criterion. In the table, values highlighted in green correspond to the evaluation reward and cost of the best-performing algorithms, and values in red indicate those of the worst-performing algorithms. This subset of results is chosen to illustrate key behavioral differences among algorithms and to highlight environment-specific

challenges that affect learning and generalization. Results for additional case studies are included in Appendix B. To identify the best and worst performing algorithms in each environment, we adopt a two-step filtering strategy based on evaluation cost and reward. This approach is motivated by the fact that reward and cost can differ significantly in scale, making it inappropriate to combine them into a single metric without normalization or weighting. To select the best algorithm, we first filter those with evaluation costs either within five times the lowest cost or within 25 units of the lowest cost. If the lowest cost is zero, we instead include all algorithms with costs below a fixed threshold of 25. Among this filtered set, the algorithm with the highest evaluation reward is selected as the best. To determine the worst algorithm, if an algorithm's evaluation cost is at least ten times greater than the second-highest and 100 units greater than the second-highest, it is directly selected as the worst. Otherwise, we consider all algorithms whose costs fall within the three highest unique values and select the one with the lowest reward.

A gap is considered significant if the absolute difference between training and evaluation results exceeds 100, and the relative difference (calculated as the absolute gap divided by the magnitude of the training result) exceeds 30%. When the magnitude of the training result is less than 0.1, only the absolute difference is considered. The training result refers to the average episode reward from the final epoch. Additionally, we say an algorithm solves an environment to reasonable-optimality if it consistently achieves reasonably high rewards (less than 35% gap with the optimal reward) while maintaining low costs, allowing for a small number of constraint violations. All the calculations are done based on the average evaluation reward and cost.

### 5.1.1 ENVIRONMENTS WITH REASONABLE PERFORMANCE

The following environments illustrate scenarios in which at least one algorithm successfully trains a policy that yields a reasonably high rewards with low associated cost:

- **InvMgmtEnv:** This case study involves a multi-echelon inventory network comprising one market, one retailer, two distributors, three producers, and two raw-material suppliers, connected via eleven reorder routes and governed by one-period lead times. Each episode spans 30 days, and training is performed with 10 episodes per epoch. P3O exhibits a significant gap between training and evaluation costs and SACPID exhibits a significant gap between training and evaluation rewards.

- **SchedMaintEnv-v0:** We consider an air separation unit with three compressors and optional external product purchases to compensate for maintenance downtime. The planning horizon is 30 days, and each episode is 31 days long. Training is conducted using 25 episodes per epoch. A significant gap between training and evaluation costs is observed for P3O and FOCOPS.

- **SchedMaintEnv-v1:** We consider a similar setup to the **SchedMaintEnv-v0**. We include uncertainty in the environment by incorporating random machine failures. Training is conducted using 25 episodes per epoch. We observe a significant gap between training and evaluation costs for P3O and FOCOPS. We also find that the performance of most algorithms in this stochastic environment closely matches their behavior in the deterministic counterpart, likely due to the underlying physics of the environment. However, a subset of algorithms shows elevated evaluation standard deviations, reflecting the inherent stochasticity of the environment.

- **UCEnv:** This example considers a single-bus unit commitment power system with five generators operating over a 24-hour horizon, with hourly updates to demand forecasts and generator states. Training is done with 100 episodes per epoch. A significant gap between training and evaluation costs is observed for SACLag.

- **GridStorageEnv:** This case study includes a three-bus network with transmission limits of 80MW, 120MW (de-energized under wildfire risk), and 90MW; one generator per bus rated at 100MW, 90MW, and 80MW; and batteries sized 1.0, 1.2, and 0.8 p.u. respectively. The horizon spans 24 hours, with perfect charging, discharging, and carry-over efficiency. Experiments were run with 10 episodes per epoch. We observe a significant gap between training and evaluation costs for SACPID and SACLag.

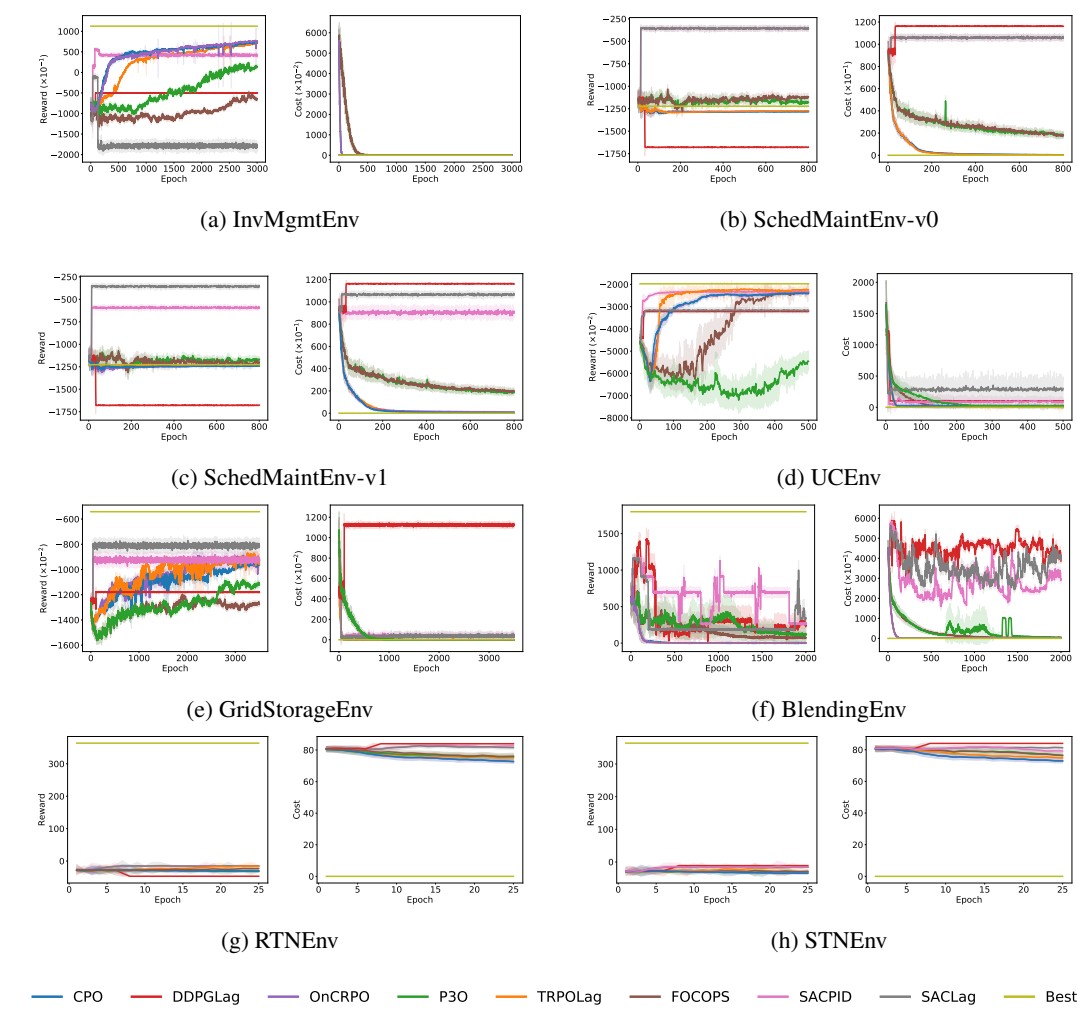

Figure 1: Training curves showing the average reward and cost per episode over training epochs across all case studies.

### 5.1.2 ENVIRONMENTS NOT TRAINED TO REASONABLE OPTIMALITY

The following environments presented significant challenges during training, preventing the agents from learning reasonably optimal policies.

- **BlendingEnv:** The case study includes 2 input streams, 2 output streams, 2 quality properties, 4 blenders, and 6 time periods per episode with the prop strategy used to handle infeasible actions. No flow is allowed between 2 of the blenders. Experiments were run with 100 episodes per epoch. A significant gap is observed between training and evaluation for costs in P3O and FOCOPS, and for rewards in P3O, DDPGLag and SACLag. As shown in Figure 1f, some algorithms exhibit considerable oscillations in both reward and cost over training epochs. This instability may be attributed to the highly non-convex nature of the underlying optimization problem and the resulting non-smooth feasible action space, which makes consistent policy improvement more difficult.

- **RTNEnv:** This case study includes 3 raw materials, 2 intermediates, 2 products, 3 tasks, 3 pieces of equipment, and 2 utilities. The planning horizon spans 30 time periods with fixed product demands. Training used 128 episodes per epoch. As shown in Figure 1g, both reward and cost exhibit slow learning across all algorithms. This sluggish progress may be

Table 2: Evaluation results for 10 episodes

| | Optimal | CPO | | DDPGLag | |
|---|---|---|---|---|---|
| Environment | Reward | Reward | Cost | Reward | Cost |
| InvMgmtEnv | 11265.97 | 7303.2 | 0 | -4858.26 | 0 |
| SchedMaintEnv-v0 | -1221.85 | -1283.96 | 29.59 | -1700.98 | 11675.99 |
| SchedMaintEnv-v1 | -1226.96 | -1240.39 | 7.88 | -1700.98 | 11675.99 |
| UCEnv | -197258 | -236553 | 5.56 | -337153 | 108 |
| GridStorageEnv | -54173 | -95198.2 | 0.02 | -118704 | 120440 |
| BlendingEnv | 1800 | 0 | 190 | 0 | 32763.44 |
| RTNEnv | 363.78 | -31.37 | 71 | -47.76 | 84 |
| STNEnv | 363.78 | -36.76 | 71 | -10.38 | 84 |

| | OnCRPO | | P3O | |
|---|---|---|---|---|
| Environment | Reward | Cost | Reward | Cost |
| InvMgmtEnv | 7598.66 | 0 | 1499.21 | 0 |
| SchedMaintEnv-v0 | -1277.08 | 25.56 | -1187.52 | 751.82 |
| SchedMaintEnv-v1 | -1244.07±1.49 | 103.21 | -1178.95±1.29 | 938.5±10.61 |
| UCEnv | -236989 | 8.55 | -556101 | 0 |
| GridStorageEnv | -90589.6 | 0.02 | -111255 | 0.02 |
| BlendingEnv | 0.03 | 200 | 0.04 | 270.08 |
| RTNEnv | -33.04 | 71 | -31.7 | 71 |
| STNEnv | -37.01 | 71 | -30.83 | 73 |

| | TRPOLag | | FOCOPS | |
|---|---|---|---|---|
| Environment | Reward | Cost | Reward | Cost |
| InvMgmtEnv | 7198.26 | 0 | -6434.03 | 0 |
| SchedMaintEnv-v0 | -1272.61 | 19.85 | -1142.69 | 616.14 |
| SchedMaintEnv-v1 | -1217.24 | 21.35 | -1188.19±4.27 | 962.3±37.61 |
| UCEnv | -218334 | 9.19 | -220312 | 0 |
| GridStorageEnv | -90265.5 | 0.02 | -126974 | 0.03 |
| BlendingEnv | 0.03 | 190 | 0.01 | 180.22 |
| RTNEnv | -12.16 | 71 | -19.56 | 73 |
| STNEnv | -11.34 | 71 | -28.79 | 73 |

| | SACPID | | SACLag | |
|---|---|---|---|---|
| Environment | Reward | Cost | Reward | Cost |
| InvMgmtEnv | 5555.74 | 0 | -14386.99 | 0 |
| SchedMaintEnv-v0 | -274.46 | 11461.29 | -274.46 | 11461.29 |
| SchedMaintEnv-v1 | -524.05 | 8600.12±31.62 | -274.46 | 11521.29±51.64 |
| UCEnv | -221922 | 41.68 | -298228 | 153.06 |
| GridStorageEnv | -100407 | 2122.38 | -68690.5 | 0.03 |
| BlendingEnv | 225.31 | 33101.34 | 523.75 | 33596.83 |
| RTNEnv | -14.7 | 83 | -14.7 | 80 |
| STNEnv | -14.7 | 81 | -14.7 | 81 |

attributed to the combination of non-convexities introduced by integer-based constraints such as task-equipment assignments and equipment availability and the indirect, time-coupled linear constraints that the action space must satisfy. The presence of temporal dependencies, material balances, and combinatorial task scheduling further compounds the difficulty of policy learning in these environments.

- **STNEnv:** This case study builds on the same network as RTNEnv but includes extended task-to-equipment mappings and product-specific processing times. To enable comparison with RTNEnv, mappings were kept unique. The planning horizon and demand profiles remain unchanged. Experiments were run with 128 episodes per epoch. TRPOLag performs the best, while OnCRPO performs the worst. Similar to RTNEnv, both reward and cost exhibit slow learning across all the algorithms in STNEnv, likely due to a similar combination of non-convexities introduced by integer-based constraints and the indirect linear constraints imposed on the action space.

### 5.1.3 DISCUSSION

Across most environments, TRPOLag perform best, while DDPGLag consistently underperforms. Interestingly, the P3O, FOCOPS, and SACLag algorithms showed a significant gap between training and evaluation performance in several settings. While some environments allowed agents to learn policies that achieved near-optimal rewards with minimal constraint violations, other environments posed significant challenges. These more difficult environments highlight the limitations of current methods and underscore the need for more sophisticated approaches to safe reinforcement learning.

## 6 CONCLUSIONS AND FUTURE DIRECTIONS

We presented SafeOR-Gym, a suite of nine operations research environments tailored to benchmark safe RL algorithms in complex, realistic settings. These environments introduce structured constraints, mixed-integer decisions, and discrete-continuous actions, going beyond the scope of conventional safe RL benchmarks. While existing algorithms can solve some tasks, they perform poorly on problems involving mixed-integer variables or nonlinear, nonconvex constraints. These limitations point to broader challenges in applying safe RL to industrial domains.

To bridge this gap, future research can pursue several promising directions. One avenue is to broaden the benchmark environments to cover uncertainty and multiagent interactions, which are central to many operations research applications. Another direction is to extend the benchmark beyond traditional optimization-based solutions. For example, we have shown how the classical base-stock reorder policy can serve as a heuristic baseline for the inventory management environment in Appendix B.3. Developing heuristics that can observe the nontrivial constraints for operations research problems is an ongoing research direction. In this spirit, SafeOR-Gym could be enriched with algorithms drawn from unified framework of sequential decision-making proposed by Powell (2022).

Finally, *SafeOR-Gym* highlights fundamental limitations of current safe RL approaches that require sustained research. Safe RL methods often fail when problem structures involve nonconvex or mixed-integer constraints, raising questions about their robustness and reliability in safety-critical domains. Algorithms like CPO require sensitive hyperparameter tuning to balance performance and feasibility, suggesting that automated approaches to constraint-aware parameter selection could reduce manual overhead and improve reproducibility. Moreover, current methods often rely on penalty-based formulations or post-hoc projection, which cannot guarantee feasibility. Recent work on action-constrained RL (Hung et al., 2025) points toward approaches that can enforce hard safety constraints directly during action selection. Another promising direction lies in encoding safety constraints within neural network architectures themselves, for instance through differentiable constraint satisfaction layers (Chen et al., 2024) that enforce feasibility by design.

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

# A  PROBLEM ENVIRONMENT DESCRIPTION

## A.1  RESOURCE TASK NETWORK ENVIRONMENT (RTNENV)

### A.1.1  OVERVIEW

The Resource Task Network (RTN) (Pantelides, 1994) is a mathematical modeling framework used for plant scheduling problems. It optimally schedules a set of interdependent production *tasks* executed on equipment, which transform reactants into products via intermediates. The full set of reactants, intermediates, products, and equipment are collectively referred to as *resources*. Products are produced to fulfill time-varying demand while minimizing operational costs. The RTNEnv simulates this system in discrete time, where the agent chooses task batch sizes at each timestep and observes the resulting state transitions, constraint violations, and rewards. The schematic of RTN has been illustrated in 2

### A.1.2  PROBLEM SETUP

The RTN is defined over a finite time horizon $T$. Reactants, intermediates, products, equipment, tasks, and utilities are defined as follows:

- **Reactants**: Consumable raw materials that can be ordered (at a cost).
- **Intermediates**: Internally produced and consumed materials. Cannot be ordered or sold.
- **Products**: Final deliverables with external demand and associated revenue and penalty structures.
- **Equipments**: Units required for task execution. Each is consumed when a task starts and returned after the task completes.
- **Tasks**: Tasks produce a subset of resources from another distinct subset of resources using a subset of equipments after a specified time period.
- **Utilities**: Time-varying operational costs incurred per unit utility consumed by tasks.
- **Demand**: Defined only for products, specifying $d_{p,t}$ at each timestep $t$.

**Sets**

- $\mathcal{R}$: All resources (reactants, intermediates, products, equipment)
- $\mathcal{R}^{\text{react}}$: Reactants
- $\mathcal{R}^{\text{int}}$: Intermediates
- $\mathcal{R}^{\text{prod}}$: Products
- $\mathcal{R}^{\text{equip}}$: Equipments
- $\mathcal{R}^{\text{pend}} \subset \mathcal{R}^{\text{int}} \cup \mathcal{R}^{\text{prod}} \cup \mathcal{R}^{\text{equip}}$: Resources which are produced/replenished.
- $\mathcal{I}$: Tasks
- $\mathcal{K}_i \subset \mathcal{R}^{\text{equip}}$: Equipment used by task $i$
- $\mathcal{U}$: Utilities
- $\mathcal{U}_i \subset \mathcal{U}$: Utilities consumed by task $i$

**Parameters**

- $T$: Time horizon
- $\tau_i$: Processing time of task $i$
- $\tau_{\max} = \max_i \tau_i$ : Maximum processing time.
- $V_i^{\min}, V_i^{\max}$: Batch size bounds for task $i$
- $X_r^0, X_r^{\min}, X_r^{\max}$: Initial and bounded inventories for resource $r$

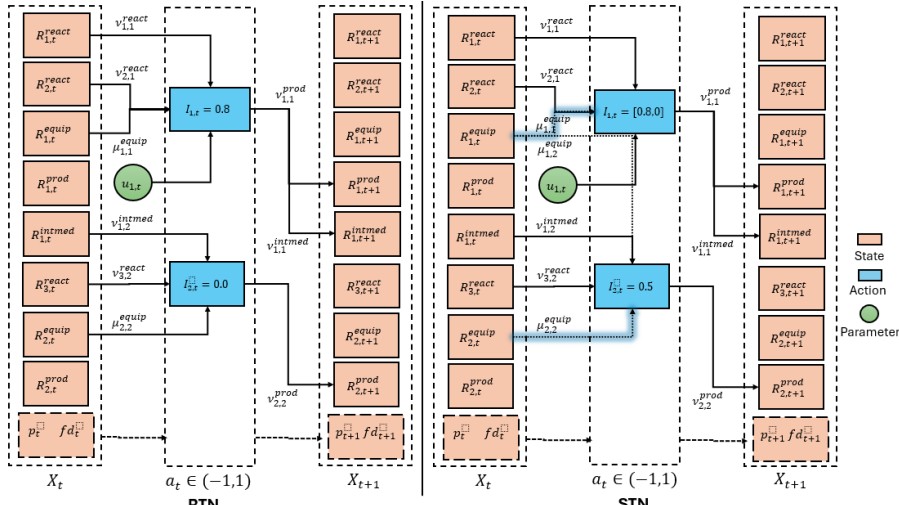

Figure 2: Schematic of RTN and STN: Solid lines in both RTN and STN represent the flow of material through the network. Other inventory levels are included in the state vector. The dashed lines in STN from equipments to tasks represent the choices available where the highlighted lines show the choice taken. The actions are scaled as described in the transition dynamics subsection.

- $\nu_{i,r}$: Stoichiometric coefficient of resource $r$ in task $i$
- $\text{cost}_r$: Unit cost of ordering reactant $r$
- $\text{price}_p$: Unit price of product $p$
- $d_{p,t}$: Demand for product $p$ at time $t$
- $u_{u,t}$: Cost of utility $u$ at time $t$
- $\lambda_{\text{sanit}}$: Sanitization penalty coefficient
- $\mathbb{I}[\cdot]$: Indicator function
- $\oplus$ : vector concatenation

### A.1.3 STATE SPACE

At timestep $t$, the agent observes:

- Inventory vector $X_t \in \mathbb{R}^{|\mathcal{R}|}$ : Inventory of all resources.
- Pending outputs $p_t \in \mathbb{R}^{\tau_{\max} \times |\mathcal{R}^{\text{pend}}|}$ : Intermediates and products which will be delivered in upcoming timesteps.
- Future demand $fd_t \in \mathbb{R}^{T \times |\mathcal{R}^{\text{prod}}|}$ : Demand of all products from timestep $t$ to $T$, post-padded with 0s to maintain consistent shapes.

### A.1.4 ACTION SPACE

The action $a_t \in [-1, 1]^{|\mathcal{I}|}$ is scaled to batch sizes via:

$$a_{i,t}^{\text{scaled}} = \frac{a_{i,t} + 1}{2} \cdot (V_i^{\max} - V_i^{\min}) + V_i^{\min}$$

For numerical stability, any $|a_{i,t}| \leq 10^{-3}$ is set to 0.

### A.1.5 TRANSITION DYNAMICS

At each step:

1. **Sanitize Action**: Prevent resource violations and enforce equipment availability by calculating the maximum inventory available for a resource, maximum batch size that can be processed based on inventory levels, and clipping between the batch size bounds accordingly.

$$b_{i,r,t} = \frac{\max(0, X_{r,t} - X_r^{\min})}{|\nu_{i,r}|}$$

$$b_{i,t} = \min_r b_{i,r,t}$$

$$a_i^{\text{clip}} = 0 \ \text{if} \ b_{i,t} < V_i^{\min}$$

$$a_{i,t}^{\text{final}} = \begin{cases} 0, & \text{if} \ \exists e \in \mathcal{K}_i \ \text{with} \ X_{e,t} = 0 \\ a_i^{\text{clip}}, & \text{otherwise} \end{cases}$$

2. **Inventory Update**: Consumes the inputs to a task immediately.

$$X_{r,t+1} = X_{r,t} + \sum_i \min(\nu_{i,r}, 0) \cdot a_{i,t}^{\text{final}}$$

3. **Pending Outputs**: Add outputs of a task to the pending output buffer and update inventory of resources that are being delivered in the next timestep.

$$
\begin{aligned}
X_{r,t+1} &= X_{r,t} + p_{t,r,0} \\
p_t &= p_{t-1,r,1:\tau_{\max}} \oplus \sum_i \max\{\nu_{i,r}, 0\} \cdot a_{i,t}^{\text{final}}
\end{aligned}
\tag{1}
$$

4. **Inventory Enforcement**: Ensures inventory bounds are not violated. A part of the cost is calculated based on this. Refer to A.1.6.

$$X_{r,t+1} = \min(X_r^{\max}, \max(X_r^{\min}, X_{r,t+1}))$$

### A.1.6 COST FUNCTION

The total cost at each timestep $t$ is given by:

$$C_t = C_t^{\text{lb}} + C_t^{\text{ub}} + C_t^{\text{eq}} + \lambda_{\text{sanit}} C_t^{\text{sanit}}$$

- **Bound Violation**: Penalizes the resource inventory going out of bounds. (Reactants going below lower bounds are assumed to imply ordering of reactants).

$$C_t^{\text{lb}} = \sum_{r \in \mathcal{R}^{\text{int}} \cup \mathcal{R}^{\text{prod}}} \mathbb{I}[X_{r,t} < X_r^{\min}]$$

$$C_t^{\text{ub}} = \sum_{r \in \mathcal{R}} \mathbb{I}[X_{r,t} > X_r^{\max}]$$

- **Equipment Feasibility Cost:** Penalizes execution of tasks when any required equipment is unavailable.

$$C_t^{\text{eq}} = \sum_i \sum_{e \in \mathcal{K}_i} \mathbb{I}[a_{i,t}^{\text{final}} > 0] \cdot \mathbb{I}[X_{e,t} < 1]$$

- **Sanitization Cost:** Penalizes deviations between the raw and final actions due to constraint handling.

$$C_t^{\text{sanit}} = \sum_i \left| a_{i,t}^{\text{final}} - a_{i,t}^{\text{scaled}} \right|$$

### A.1.7 REWARD FUNCTION

The total reward at each timestep $t$ is:

$$\Pi_t = \Pi_t^{\text{rev}} - \Pi_t^{\text{util}} - \Pi_t^{\text{unmet}} - \Pi_t^{\text{order}}$$

- **Revenue:** Earned by fulfilling product demand, constrained by available inventory.

$$\Pi_t^{\text{rev}} = \sum_{p \in \mathcal{R}^{\text{prod}}} \min(X_{p,t} - X_p^{\min}, d_{p,t}) \cdot \text{price}_p$$

- **Unmet Demand Penalty:** Penalizes unmet product demand at 1.5× product price.

$$\Pi_t^{\text{unmet}} = 1.5 \cdot \sum_{p \in \mathcal{R}^{\text{prod}}} \max(d_{p,t} - (X_{p,t} - X_p^{\min}), 0) \cdot \text{price}_p$$

- **Reactant Ordering Cost:** Charged when reactant inventory drops below zero (interpreted as external procurement)

$$\Pi_t^{\text{order}} = \sum_{r \in \mathcal{R}^{\text{react}}} \max(-X_{r,t}, 0) \cdot \text{cost}_r$$

- **Utility Cost:** Incurred from executing tasks using utilities.

$$\Pi_t^{\text{util}} = \sum_i \sum_{j \in \mathcal{U}_i} a_{i,t}^{\text{final}} \cdot u_{j,t}$$

### A.1.8 EPISODE TERMINATION

The episode ends when $t = T$. Truncation does not occur even under constraint violations.

## A.2 STATE TASK NETWORK (STNENV)

### A.2.1 OVERVIEW

The State Task Network (STN) (Pantelides, 1994) is a mathematical modeling framework for short-term production scheduling. It focuses on scheduling a set of production *tasks* on processing units, where each task transforms material *states* over time. Unlike the RTN, which models resources more abstractly, the STN emphasizes the evolution of *material states* through task execution and transfer between units. The objective is to satisfy time-varying demand for final products while minimizing operational costs. The STNEnv simulates this system in discrete time, where an agent selects task batch sizes on specific units at each timestep and observes the resulting state transitions, constraint violations, and rewards. The schematic of STN has been illustrated in 2

### A.2.2 PROBLEM SETUP

The STN is defined over a finite time horizon $T$. Material states, processing units, tasks, and utilities are defined as follows:

- **Reactants**: Consumable raw materials that can be externally procured at a cost. They serve as initial inputs to tasks.
- **Intermediates**: Internally produced and consumed materials. They are used to link tasks within the network. Intermediates cannot be purchased or sold externally.
- **Products**: Final deliverables for which external demand is specified. Products may generate revenue if delivered on time and incur penalties if demand is unmet.
- **Units**: Processing equipment on which tasks are scheduled. Each unit can process at most one task at a time and becomes available again after the task's processing duration.
- **Tasks**: Operations that transform a subset of input states into a subset of output states on a designated unit. Each task has a fixed processing time, specific unit assignment, defined stoichiometry, and utility consumption profile.
- **Utilities**: Operational resources (e.g., electricity, steam) with time-varying costs. Each task consumes a fixed amount of utilities per unit batch size, incurring operational costs accordingly.
- **Demand**: Defined only for product states, specifying $d_{s,t}$ for state $s$ at time $t$. Demand fulfillment yields revenue, while shortages incur penalties.

**Sets**

- $\mathcal{S}$: All material states (reactants, intermediates, products, equipments)
- $\mathcal{S}^{\text{react}} \subset \mathcal{S}$: Reactants (raw materials)
- $\mathcal{S}^{\text{int}} \subset \mathcal{S}$: Intermediates (internal materials)
- $\mathcal{S}^{\text{prod}} \subset \mathcal{S}$: Products (deliverables)
- $\mathcal{E}$: Equipment units
- $\mathcal{S}^{\text{pend}} \subset \mathcal{S}^{\text{int}} \cup \mathcal{S}^{\text{prod}} \cup \mathcal{E}$: States that are produced (i.e., replenished by tasks)
- $\mathcal{I}$: Tasks
- $\mathcal{K}_i \subset \mathcal{E}$: Set of units on which task $i$ can be scheduled
- $\mathcal{U}$: Utilities
- $\mathcal{U}_i \subset \mathcal{U}$: Utilities consumed by task $i$

**Parameters**

- $T$: Time horizon
- $\tau_i$: Processing time of task $i$
- $\tau_{\max} = \max_i \tau_i$: Maximum processing time
- $V_{i,e}^{\min}, V_{i,e}^{\max}$: Batch size bounds for task $i$ by using equipment $e$.
- $X_s^0, X_s^{\min}, X_s^{\max}$: Initial and bounded inventories for state $s$
- $\nu_{i,s}$: Stoichiometric coefficient of state $s$ in task $i$ (negative if consumed, positive if produced)
- $\text{cost}_s$: Unit cost of ordering reactant $s \in \mathcal{S}^{\text{react}}$
- $\text{price}_p$: Unit price of product $p \in \mathcal{S}^{\text{prod}}$
- $d_{p,t}$: Demand for product $p$ at time $t$
- $u_{u,t}$: Cost of utility $u$ at time $t$
- $\lambda_{\text{sanit}}$: Sanitization penalty coefficient (for equipment reuse)
- $\mathbb{I}[\cdot]$: Indicator function
- $\oplus$ : vector concatenation

### A.2.3 STATE SPACE

At timestep $t$, the agent observes:

- Inventory vector $X_t \in \mathbb{R}^{|\mathcal{S}|}$ : Inventory of all material states.
- Pending outputs $p_t \in \mathbb{R}^{\tau_{\max} \times |\mathcal{S}^{\text{pend}}|}$ : Intermediates and products scheduled to be produced in future timesteps due to ongoing tasks.
- Future demand $fd_t \in \mathbb{R}^{T \times |\mathcal{S}^{\text{prod}}|}$ : Demand for all product states from timestep $t$ to $T$, post-padded with 0s to maintain consistent shapes.

### A.2.4 ACTION SPACE

The action $a_t \in [-1, 1]^{|\mathcal{I} \times \mathcal{E}|}$ represents a normalized matrix over task-equipment pairs and is scaled to batch sizes via:

$$a_{i,e,t}^{\text{scaled}} = \frac{a_{i,e,t} + 1}{2} \cdot (V_{i,e}^{\max} - V_{i,e}^{\min}) + V_{i,e}^{\min}$$

For numerical stability, any $|a_{i,e,t}| \leq 10^{-3}$ is set to 0.

### A.2.5 TRANSITION DYNAMICS

At each step:

1. **Sanitize Action**: Prevent state violations and enforce unit availability by calculating the maximum available inventory for each input state, the maximum feasible batch size given current inventories, and clipping between the allowed batch size bounds accordingly.

$$b_{i,s,t} = \frac{\max(0, X_{s,t} - X_s^{\min})}{|\nu_{i,s}|}$$

$$b_{i,t} = \min_{s \in \mathcal{S}:\nu_{i,s}<0} b_{i,s,t}$$

$$a_{i,e,t}^{\text{clip}} = 0 \text{ if } b_{i,t} < V_{i,e}^{\min}$$

$$a_{i,e,t}^{\text{final}} = \begin{cases} 0, & \text{if } e \notin \mathcal{K}_i \text{ or } X_{e,t} = 0 \\ a_{i,e,t}^{\text{clip}}, & \text{otherwise} \end{cases}$$

2. **Inventory Update**: Immediately consumes the input states required by the activated tasks.

$$X_{s,t+1} = X_{s,t} + \sum_e \sum_i \min(\nu_{i,s}, 0) \cdot a_{i,e,t}^{\text{final}}$$

3. **Pending Outputs**: Add the output states of tasks to the pending output buffer and update inventories of materials delivered at the current timestep.

$$X_{s,t+1} = X_{s,t} + p_{t,s,0}$$

$$p_{t+1,s} = p_{t,s,1:\tau_{\max}} \oplus \sum_e \sum_i \max(\nu_{i,s}, 0) \cdot a_{i,e,t}^{\text{final}} \tag{2}$$

4. **Inventory Enforcement**: Enforces inventory bounds to prevent overflow or underflow. Violations of these bounds contribute to the constraint cost.

$$X_{s,t+1} = \min(X_s^{\max}, \max(X_s^{\min}, X_{s,t+1}))$$

### A.2.6 COST FUNCTION

The total cost at each timestep $t$ is given by:

$$C_t = C_t^{\text{lb}} + C_t^{\text{ub}} + C_t^{\text{eq}} + \lambda_{\text{sanit}} C_t^{\text{sanit}}$$

- **Bound Violation**: Penalizes material inventory going out of bounds. (Reactants going below their lower bounds are interpreted as triggering procurement costs.)

$$C_t^{\text{lb}} = \sum_{s \in \mathcal{S}^{\text{int}} \cup \mathcal{S}^{\text{prod}}} \mathbb{I}[X_{s,t} < X_s^{\min}]$$

$$C_t^{\text{ub}} = \sum_{s \in \mathcal{S}} \mathbb{I}[X_{s,t} > X_s^{\max}]$$

- **Equipment Feasibility Cost**: Penalizes task executions on unavailable units.

$$C_t^{\text{eq}} = \sum_i \sum_{e \in \mathcal{K}_i} \mathbb{I}[a_{i,e,t}^{\text{final}} > 0] \cdot \mathbb{I}[X_{e,t} < 1]$$

- **Sanitization Cost**: Penalizes deviation between scaled and final actions due to sanitization (inventory or equipment infeasibility).

$$C_t^{\text{sanit}} = \sum_{i,e} |a_{i,e,t}^{\text{final}} - a_{i,e,t}^{\text{scaled}}|$$

### A.2.7 REWARD FUNCTION

The total reward at each timestep $t$ is:

$$\Pi_t = \Pi_t^{\text{rev}} - \Pi_t^{\text{util}} - \Pi_t^{\text{unmet}} - \Pi_t^{\text{order}}$$

- **Revenue:** Earned by fulfilling product demand, constrained by available inventory.

$$\Pi_t^{\text{rev}} = \sum_{p \in \mathcal{S}^{\text{prod}}} \min(X_{p,t} - X_p^{\min}, d_{p,t}) \cdot \text{price}_p$$

- **Unmet Demand Penalty:** Penalizes unmet product demand at $1.5\times$ the product price.

$$\Pi_t^{\text{unmet}} = 1.5 \cdot \sum_{p \in \mathcal{S}^{\text{prod}}} \max(d_{p,t} - (X_{p,t} - X_p^{\min}), 0) \cdot \text{price}_p$$

- **Reactant Ordering Cost:** Incurred when reactant inventory falls below zero (interpreted as external procurement).

$$\Pi_t^{\text{order}} = \sum_{s \in \mathcal{S}^{\text{react}}} \max(-X_{s,t}, 0) \cdot \text{cost}_s$$

- **Utility Cost:** Accrued from task executions that consume utilities, priced per unit usage and per unit batch size.

$$\Pi_t^{\text{util}} = \sum_{i,e} \sum_{j \in \mathcal{U}_i} a_{i,e,t}^{\text{final}} \cdot u_{j,t}$$

### A.2.8 EPISODE TERMINATION

The episode ends when $t = T$. Truncation does not occur even under constraint violations.

## A.3 UNIT COMMITMENT (UCENV)

### A.3.1 OVERVIEW

The unit commitment problem is one of the most widely used optimization problems in power systems, aiming to minimize the operational costs of power generation over a specified planning horizon by determining optimal decisions for switching power units on or off and managing power dispatch (Knueven et al., 2020). These decisions must comply with safety requirements and fulfill various operational goals. In practice, the unit commitment problem is solved in advance on a rolling basis, as future electricity demand is uncertain and must be forecasted continually once new information becomes available. The growth of renewable energy sources and fluctuations in the electricity markets further increase the uncertainties in demand forecasts. Due to the size of power systems in the real world, solving the resulting MILP or MIQCP problem on time can be prohibitively difficult. Therefore, the development of efficient solution methods that can respond to frequent forecast updates is of significant interest. In this study, we formulate the unit commitment problem as a CMDP and implement it as the environment UCEnv. To support safe sequential decision making in power scheduling and dispatch, practical constraints, such as minimum up and down time, ramping constraints, and reserve requirements, are incorporated into the environment. Given the complexity of unit commitment problem, the environment is provided in two versions. The UCEnv-v0 version assumes a single-bus system, while UCEnv-v1 requires an agent to take into account distributed power demands and power flows within a transmission network.

### A.3.2 PROBLEM SETUP

**Sets**

- $\mathcal{G}$: the set of generators.
- $\mathcal{N}$: the set of buses.
- $\mathcal{K}$: the set of transmission lines.

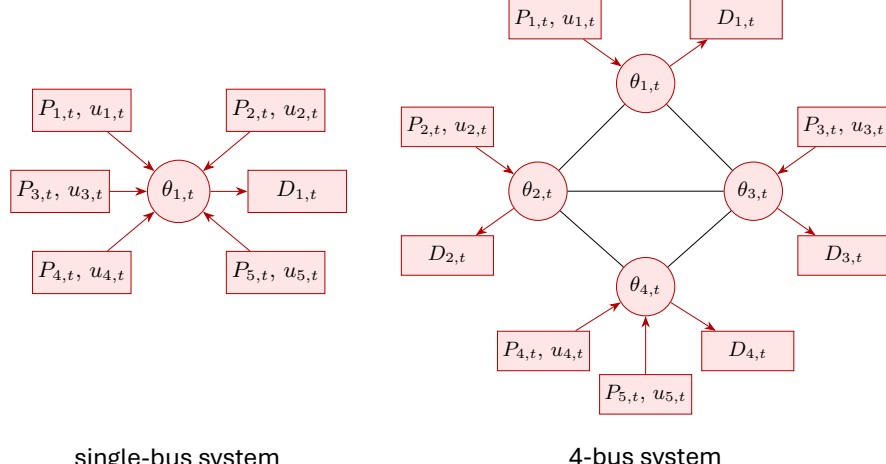

single-bus system          4-bus system

Figure 3: Schematic of UC in a single-bus system and 4-bus system at the time point $t$: Red lines indicate power inflow and outflow. Black lines denote transmission lines. At each time point, forecast demands, current power output, current voltage angle, current and previous on-off status (state) are used to infer the power output, voltage angle, on-off status at the next time point (action).

- $\mathcal{G}_n$: the set of generators that is connected to the $n$ bus.
- $k(n)$: the from-bus for the $k$th transmission line.
- $k(m)$: the to-bus for the $k$th transmission line.
- $\delta^+(n)$: the set of transmission line for the to-bus $n$.
- $\delta^-(n)$: the set of transmission line for the from-bus $n$.

**Parameters**

- $(a_i, b_i, c_i)$: quadratic, linear, and constant cost coefficients of power generation.
- $C_i^v$: startup cost coefficients.
- $C_i^w$: shutdown cost coefficients.
- $UT_i$: minimum up time.
- $DT_i$: minimum down time.
- $RU_i$: ramp-up rate.
- $RD_i$: ramp-down rate.
- $SU_i$: start-up rate.
- $SD_i$: shut-down rate.
- $P_{\max i}$: maximum power output.
- $P_{\min i}$: minimum power output.
- $\Theta_{\max n}$: maximum voltage angle.
- $\Theta_{\min n}$: minimum voltage angle.
- $F_{\max k}$: maximum transmission capacity.
- $F_{\min k}$: minimum transmission capacity.
- $D_{n,t}$; time-varying power demand.
- $C^{LS}$: load shedding cost coefficient for failure to meet load.
- $C^R$: reserve shortfall cost coefficient for failure to meet the reserve requirement.
- $R$: system-wide reserve requirement

### A.3.3 STATE SPACE

The state $s_t$ includes the following components:

- $u_{i,t}^{\text{seq}} = [u_{i,t}, u_{i,t}^{\text{old}}] \in \{0,1\}^{\max(UT_i, DT_i)+1}$: a sequence of binary indicators about the on-off status of the generator $i \in \mathcal{G}$, where $u_{i,t}^{\text{old}}$ records the history in the most recent time periods before $t$, used to track compliance with the up/down time. The dimension of this variable is $\sum_{i \in \mathcal{G}}(\max(UT_i, DT_i) + 1)$.

- $p_{i,t} \in [P_{\min i}, P_{\max i}]$: power output of the generator $i \in \mathcal{G}$. The dimension of this variable is $|\mathcal{G}|$.

- $\theta_{n,t} \in [\Theta_{\min n}, \Theta_{\max n}]$: voltage angle of the bus $n \in \mathcal{N}$. The dimension of this variable is $|\mathcal{N}|$. In the simpler version, UCEnv-v0, the environment is assumed to be a single-bus system where the voltage angle is not considered.

- $D_{n,t+1:t+W} \in \mathbb{R}_{\geq 0}^{W}$: forecast demand of the bus $n \in \mathcal{N}$ with a window length of $W$. The dimension of this variable is $|\mathcal{N}| \times W$. When the window extends beyond the episode horizon, it uses the forecast demand from the following steps.

The full observation is flattened into a continuous vector space for reinforcement learning (RL) training.

### A.3.4 ACTION SPACE

At each time step $t$, the agent takes the following actions $a_t$:

- $u_{i,t+1} \in \{0,1\}$: on/off status of the generator $i \in \mathcal{G}$. The dimension of this variable is $|\mathcal{G}|$.

- $p_{i,t+1} \in [P_{\min i}, P_{\max i}]$: power output of the generator $i \in \mathcal{G}$. The dimension of this variable is $|\mathcal{G}|$.

- $\theta_{n,t+1} \in [\Theta_{\min n}, \Theta_{\max n}]$: voltage angle of the bus $n \in \mathcal{N}$, where $\theta_{1,t+1} = 0$ is always fixed as reference. The dimension of this variable is $|\mathcal{N}| - 1$.

These decision variables are initially normalized to a continuous action space $a_t \in [-1, 1]^n$ and then mapped to their actual values within their bounds.

### A.3.5 TRANSITION DYNAMICS

Given the action $a_t$, the next state $s_{t+1}$ is computed as follows.

**Intermeidate state, turn-on and turn-off status and sequence**: First, we compute the intermediate states $v_{i,t+1}, w_{i,t+1}$ as well as the sequences $v_{i,t+1}^{\text{seq}}$ and $w_{i,t+1}^{\text{seq}}$. These intermediate states serve as a practical representation of whether a generator has recently been activated or deactivated, facilitating the subsequent computation of costs, violations, and rewards.

$$v_{i,t+1} = \max(0, u_{i,t+1} - u_{i,t})$$
$$w_{i,t+1} = -\min(0, u_{i,t+1} - u_{i,t})$$
$$v_{i,t+1}^{\text{seq}} = \left[v_{i,t+1}, v_{i,t}^{\text{old}}[:-1]\right]$$
$$w_{i,t+1}^{\text{seq}} = \left[w_{i,t+1}, w_{i,t}^{\text{old}}[:-1]\right]$$

**Check and repair on-off status** Next, we check the feasibility of the on-off status. The first inequality evaluates the minimum up-time constraint, ensuring the $i$th generator remains on for a specified number of time periods, $UT_i$, before shutdown. The second inequality evaluates the minimum down-time constraint, which requires the $i$th generator to have a minimum off duration, $DT_i$, before restart. These constraints prevent excessive wear-and-tear resulting from frequent cycling. The violation indicates the power generator that must be kept on and off, respectively. The values of $u_{i,t+1}$ of the must-on generators (must-off generators) are then corrected to 1 (0).

$$u_{i,t+1}^{\text{r}} = \begin{cases} 1, & \text{if } \sum v_{i,t+1}^{\text{seq}} > u_{i,t+1} \\ 0, & \text{elif } \sum w_{i,t+1}^{\text{seq}} > 1 - u_{i,t+1} \\ u_{i,t+1}, & \text{otherwise} \end{cases}$$

**Update sequence of on-off status** : After checking and repairing the on-off status, their sequence is updated:

$$\mathbf{u}_{i,t+1} = \left[ u^{\mathrm{r}}_{i,t+1}, u^{\mathrm{old}}_{i,t} [:-1] \right]$$

**Check and repair power output** We proceed to check the violations of the ramping constraints. The first inequality evaluates the ramp-up constraint, and the second inequality evaluates the ramp-down constraint. These constraints impose limits on the rate at which a generator increases or decreases its output between consecutive time steps, reflecting the physical capabilities of generators. The violation indicates the power generation that must be kept within a tighter bound based on the repaired on-off status.

$$p^{\mathrm{r}}_{i,t+1} = \begin{cases} p_{i,t} + RU_i \cdot u_{i,t} + SU_i \cdot v^{\mathrm{r}}_{i,t+1}, & \text{if } p_{i,t+1} - p_{i,t} > RU_i \cdot u_{i,t} + SU_i \cdot v^{\mathrm{r}}_{i,t+1} \\ p_{i,t} - RD_i \cdot u^{\mathrm{r}}_{i,t+1} - SD_i \cdot w^{\mathrm{r}}_{i,t+1}, & \text{elif } p_{i,t} - p_{i,t+1} > RD_i \cdot u^{\mathrm{r}}_{i,t+1} + SD_i \cdot w^{\mathrm{r}}_{i,t+1} \\ p_{i,t+1}, & \text{otherwise} \end{cases}$$

**Update power output** : The power output is then updated by multiplying it with its on-off status.

$$p_{i,t+1} = u^{\mathrm{r}}_{i,t+1} \cdot p^{\mathrm{r}}_{i,t+1}$$

**Update voltage angle**

$$\theta_{n,t+1} = \theta_{n,t+1}$$

**Intermeidate state, load unfulfillment and power reserve** We compute the following intermediate states to ease the computation of rewards.

$$f_{k,t+1} = B_k(\theta_{k(n),t+1} - \theta_{k(m),t+1})$$

$$f^{\mathrm{r}}_{k,t+1} = \max \left( \min(F_{\max k}, f_{k,t+1}), F_{\min k} \right)$$

$$s_{n,t+1} = \max \left( D_{n,t+1} - \sum_{i \in \mathcal{G}_n} p_{i,t+1} - \sum_{k \in \delta^+(n)} f^{\mathrm{r}}_{k,t+1} + \sum_{k \in \delta^-(n)} f^{\mathrm{r}}_{k,t+1}, 0 \right)$$

$$r_{i,t+1} = \max \left( \min(P_{\max i} \cdot u_{i,t+1} - p_{i,t+1}, RU_i \cdot u_{i,t} + SU_i \cdot v_{i,t+1}), 0 \right)$$

Although the power flow $f_{k,t+1}$ is clipped for subsequent computation, the voltage angles $\theta_{n,t+1}$ are not repaired accordingly because future states are independent of $\theta_{n,t+1}$ and they will not accumulate further violations.

**Demand forecast** :

$$D_{n,t+2:t+W+1} = \text{forecast}(D_{n,t+1:t+W})$$

### A.3.6 COST FUNCTION

**Minimum up-time & down-time violations** : The penalty for correcting invalid on-off status decisions is the number of minimum up-time and down-time violations multiplied by a penalty factor. The violations represent that the agent attempted to turn off a power unit that must be kept on or turn on a power unit that must be kept off at the current time step.

$$C^{\mathrm{UTDT}}_t = \sum_i P \cdot \mathbb{I}\left( \sum v^{\mathrm{seq}}_{i,t+1} > u_{i,t+1} \vee \sum w^{\mathrm{seq}}_{i,t+1} > 1 - u_{i,t+1} \right)$$

**Ramp-up & ramp-down violations** : The penalty for correcting invalid power output decisions is the magnitude of ramp-up and ramp-down violations multiplied by a penalty factor. The violations represent that the agent attempted to increase or decrease a power output too aggressively, which exceeds the generator's allowable ramping limits between consecutive time steps.

$$C^{\mathrm{Ramp}}_t = P \cdot \sum_i \max \left( p_{i,t+1} - p_{i,t} - RU_i \cdot u_{i,t} - SU_i \cdot v^{\mathrm{r}}_{i,t+1}, p_{i,t} - p_{i,t+1} - RD_i \cdot u^{\mathrm{r}}_{i,t+1} - SD_i \cdot w^{\mathrm{r}}_{i,t+1}, 0 \right)$$

**Transmission capacity violations** : The penalty for correcting invalid volatage angle decisions is the magnitude of transmission capacity violations multiplied by a penalty factor. The violations represent that the agent attempted to allocate an excessive or insufficient amount of power to other buses, exceeding or falling short of the transmission capacity.

$$C_t^{\text{Cap}} = P \cdot \sum_k \max\left(f_{k,t+1} - F_{\max k}, F_{\min k} - f_{k,t+1}, 0\right)$$

**Total Cost**

$$C_t = C_t^{\text{UTDT}} + C_t^{\text{Ramp}} + C_t^{\text{Cap}}$$

### A.3.7 REWARD FUNCTION

The agent receives a reward equal to the negative of the production generation cost, startup cost, shutdown cost, load shedding cost and reserve shortfall cost.

**Production generation reward**:

$$\Pi_t^{\text{pg}} = -\sum_i (a_i \cdot p_{i,t}^2 + b_i \cdot p_{i,t} + c_i)$$

**Startup reward**:

$$\Pi_t^v = -\sum_i (C_i^v \cdot v_{i,t})$$

**Shutdown reward**:

$$\Pi_t^w = -\sum_i (C_i^w \cdot w_{i,t})$$

**Load shedding reward**:

$$\Pi_t^{LS} = -C^{LS} \cdot \sum_n s_{n,t}$$

where $s_{n,t}$ represents the load unfulfillment at bus $n$.

**Reserve shortfall reward**:

$$\Pi_t^R = -C^R \cdot \max(R - \sum_i r_{i,t}, 0),$$

where $r_{i,t}$ represents the power that can be reserved at generator $i$.

**Total reward**:

$$\Pi_t = \Pi_t^{\text{pg}} + \Pi_t^v + \Pi_t^{LS} + \Pi_t^R$$

### A.3.8 EPISODE DYNAMICS

The episode commences with the known sequence of the on-off status and power output from the preceding time step, and terminates after $T$ time steps.

## A.4 GENERATION AND TRANSMISSION EXPANSION PLANNING(GTEPENV)

### A.4.1 OVERVIEW

The generation and transmission expansion problem is a critical planning task in power systems, aiming to determine when and where to install new generators and transmission lines to ensure adequate power supply amidst growing and spatially distributed demand. This problem is inherently combinatorial and must satisfy constraints such as the maximum number of generators in a region and overall demand satisfaction (Li et al., 2022).

Because electricity demand varies over time and across regions, expansion decisions must be made periodically, turning the problem into a sequential decision-making process. Moreover, future

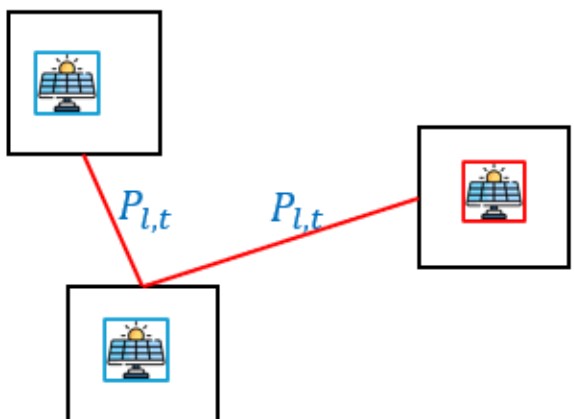

Figure 4: Schematic of GTEPEnv: This is a representation of a network of 3 regions where we install solar panels and transmission lines to deal with the demand for power. The solar panels with blue borders represent those already installed (state), and the one with the red border represents the one being installed in this time period (action)

demand is typically forecasted and therefore uncertain, necessitating a rolling planning framework in which decisions are continually updated as new information becomes available. This need is further heightened by the increasing integration of renewable energy sources and the growing volatility in energy consumption patterns, both of which introduce greater uncertainty into power system operations. As such, effective solutions must explicitly account for these uncertainties and remain flexible enough to adapt to frequent forecast revisions. Keeping these considerations in mind, reinforcement learning (RL) offers a promising approach for addressing the generation and transmission expansion problem (Pesántez et al., 2024). In particular, safe reinforcement learning techniques enable agents to learn actions that are more likely to be both feasible and near-optimal. To this end, we model the problem as a CMDP and implement it as the environment `GTEPEnv`, which supports decision-making regarding the installation of generators and transmission lines across different time periods and regions. We assume generators operate at full capacity and the cost for curtailment is negligible. A schematic of the environment is shown in figure 4.

### A.4.2 PROBLEM SETUP

We consider a multi-period generator and transmission expansion planning problem. The sets and known parameters used in the model are described below.

**Sets**

- $\mathcal{T} = \{1, \ldots, T\}$: Set of discrete time periods.
- $\mathcal{R}$: Set of regions.
- $\mathcal{G}$: Set of generator types.
- $\mathcal{L} \subseteq \mathcal{R} \times \mathcal{R}$: Set of candidate transmission lines, where each line is represented by a single directed pair $(r_1, r_2)$ with $r_1 \neq r_2$. For any unordered pair $\{r_1, r_2\}$, only one direction (e.g., $(r_1, r_2)$) is included in $\mathcal{L}$.

**Parameters**

- $\text{Dem}_{r,t}$: Electricity demand in region $r \in \mathcal{R}$ at time $t \in \mathcal{T}$.

- $\text{Cap}_i^{\text{gen}}$: Capacity of a single unit of generator type $i \in \mathcal{G}$.

- $C_i^{\text{inst,gen}}$: Installation cost of generator type $i$.

- $\text{Cap}_l^{\text{tl}}$: Transmission capacity of line $l \in \mathcal{L}$.

- $C_l^{\text{inst,tl}}$: Installation cost of transmission line $l$.

- $M_{i,r}$: Maximum number of generators of type $i$ that can be installed in region $r$.

- $\lambda_0$: Fixed penalty term (analogous to an $\ell_0$-style penalty to encourage sparsity).

- $\lambda_2$: Quadratic penalty coefficient (analogous to an $\ell_2$-style penalty to discourage overuse or smooth solutions).

- $k$: Window length for demand forecast

- $\epsilon$: A small threshold used to ignore negligible power flows

### A.4.3 STATE SPACE

The state at each time step $t \in \mathcal{T}$, denoted $s_t$, includes the following components:

- $n_{i,r,t} \in \mathbb{Z}_{\geq 0}$, bounded by $[0, M_{i,r}]$: Number of generators of type $i \in \mathcal{G}$ installed in region $r \in \mathcal{R}$ at time $t \in \mathcal{T}$. This component contributes a state dimension of $|\mathcal{G}| \times |\mathcal{R}|$.

- $nt_{l,t} \in \{0, 1\}$: Binary indicator for whether transmission line $l \in \mathcal{L}$ is installed at time $t$. This contributes a dimension of $|\mathcal{L}|$.

- $Dem_{r,t+1:t+k}$: Forecasted demand in region $r \in \mathcal{R}$ from time $t+1$ to $t+k$, where $k$ is the forecasting window. This has a dimension of $|\mathcal{R}| \times k$. Entries are padded with zeros for time periods beyond the episode horizon.

- $t$: Current time index, optionally included to provide temporal context.

All components are concatenated and flattened into a continuous vector for use in reinforcement learning.

### A.4.4 ACTION SPACE

At each time step $t \in \mathcal{T}$, the agent selects the following actions:

- $n_{i,r,t}^{\text{add}} \in [0, M_{i,r}]$: Number of generators of type $i \in \mathcal{G}$ to install in region $r \in \mathcal{R}$ at time $t$. This defines a decision space of dimension $|\mathcal{G}| \times |\mathcal{R}|$.

- $P_{l,t} \in [-\text{Cap}_l^{\text{tl}}, \text{Cap}_l^{\text{tl}}]$: Power flow along transmission line $l = (r_1, r_2) \in \mathcal{L}$ at time $t$, where the direction of flow is from $r_1$ to $r_2$. This defines a space of dimension $|\mathcal{L}|$.

The bounds of the action space are given by the physical constraints on installation limits and transmission capacities.

For reinforcement learning, actions are normalized to $[-1, 1]$ and then scaled back to their original ranges using:

$$a_{\text{actual}} = \frac{(a_{\text{normalized}} + 1)}{2} \cdot (a_{\text{high}} - a_{\text{low}}) + a_{\text{low}}$$

### A.4.5 TRANSITION DYNAMICS

After scaling, we denote the set of actions by

$$\left\{ n_{i,r,t}^{\text{add,action}}, \quad P_{l,t}^{\text{action}} \right\}.$$

We first further process the actions by rounding the number of generators added to the nearest integer and setting negligible power flows to 0.

$$n_{i,r,t}^{\text{add,prebound}} = \text{round}(n_{i,r,t}^{\text{add,action}})$$

$$P_{l,t} = \begin{cases} 0 & \text{if } |P_{l,t}^{\text{action}}| \leq \epsilon \\ P_{l,t}^{\text{action}} & \text{otherwise} \end{cases}$$

We then increment $t$ to $t+1$.

**Checking action for generator bounds:** Next, we proceed to check for violations of bounds in the state. The main focus is on checking for violations related to the number of generators in each region, as everything else has been accounted for in the scaling process. If the action results in a violation of the constraint, we adjust the action to ensure the state reaches the maximum possible configuration within the bounds.

$$n_{i,r,t}^{\text{add}} = \begin{cases} M_{i,r} - n_{i,r,t-1} & \text{if } n_{i,r,t-1} + n_{i,r,t}^{\text{add,prebound}} > M_{i,r} \\ n_{i,r,t}^{\text{add,prebound}} & \text{otherwise} \end{cases}$$

After sanitizing the actions and ensuring the state satisfies its bounds, the state is updated as follows:

**Generator Updates:**
$$n_{i,r,t} = n_{i,r,t-1} + n_{i,r,t}^{\text{add}}$$

**Transmission Line Installation:**

$$nt_{l,t} = \begin{cases} 1 & \text{if } nt_{l,t-1} = 0 \text{ and } |P_{l,t}| > 0 \\ nt_{l,t-1} & \text{otherwise} \end{cases}$$

We then proceed to calculate the cost and reward. Finally, we shift the demand forecast by one step to $\text{Dem}_{r,t+1:t+k}$.

### A.4.6  COST FUNCTION

The total cost at each time step consists of penalties for constraint violations, including those for the bounds of the state, and demand satisfaction violations. The components of the cost are defined as follows:

**Generator Bound Violations:** We apply a combination of $L_0$ and $L_2$ penalties for generating an action before sanitization that violates the maximum number of generators in a region:

$$C_{i,r}^{\text{bound,gen}} = \begin{cases} \lambda_0 + \lambda_2 \cdot \left( n_{i,r,t}^{\text{add,prebound}} + n_{i,r,t-1} - M_{i,r} \right)^2 & \text{if } n_{i,r,t}^{\text{add,prebound}} + n_{i,r,t-1} > M_{i,r} \\ 0 & \text{otherwise} \end{cases}$$

**Demand Violations:** The available power $\text{Pow}_{r,t}^{\text{avail}}$ in region $r$ at time $t$ is calculated as:

$$\text{Pow}_{r,t}^{\text{avail}} = \sum_i n_{i,r,t} \cdot \text{Cap}_i^{\text{gen}} + \sum_{r'|(r,r')\in\mathcal{L}} P_{(r,r'),t} - \sum_{r'|(r',r)\in\mathcal{L}} P_{(r',r),t}$$

**Note:** The power flow obtained from the agent is only in one direction. The power flow along the reverse direction can be interpreted simply the negative of the power flow in the original direction. Specifically, for any transmission line $(r_1, r_2) \in \mathcal{L}$, the power flow from region $r_1$ to region $r_2$ is denoted by $P_{(r_1,r_2),t}$, and the reverse flow from region $r_2$ to region $r_1$ is $P_{(r_2,r_1),t} = -P_{(r_1,r_2),t}$.

We apply a combination of $L_0$ and $L_2$ penalties for unmet demand:

$$C_r^{\text{demand}} = \begin{cases} \lambda_0 + \lambda_2 \cdot \left( \text{Dem}_{r,t} - \text{Pow}_{r,t}^{\text{avail}} \right)^2 & \text{if } \text{Dem}_{r,t} > \text{Pow}_{r,t}^{\text{avail}} \\ 0 & \text{otherwise} \end{cases}$$

**Total cost:** The total cost is the sum of the penalties for all violations, as follows:

$$C_t = \sum_{i,r} C_{i,r}^{\text{bound,gen}} + \sum_r C_r^{\text{demand}}$$

The total cost is used to guide the agent toward optimal decision-making.

### A.4.7 REWARD FUNCTION

The reward function consists of:

**Contribution to reward from installation of generators** :

$$\pi_t^{gen} = -\sum_{i,r} n_{i,r,t}^{\text{add}} \cdot C_i^{\text{inst,gen}}$$

**Contribution to reward from installation of transmission lines** :

$$\pi_t^{tl} = -\sum_{l \in \mathcal{L}} C_l^{\text{inst,tl}} \cdot \mathbb{1}[nt_{l,t-1} = 0 \wedge |P_{l,t}| > 0]$$

**Total reward** :

$$\pi_t = \pi_t^{gen} + \pi_t^{tl}$$

### A.5 MULTIPERIOD BLENDING PROBLEM(BLENDINGENV)

### A.5.1 OVERVIEW

The multiperiod blending problem, a core challenge in chemical engineering, involves optimally blending multiple input streams to produce outputs with desired quality attributes over a series of time periods. Input streams, characterized by specific properties, are procured and stored in inventory vessels. These streams are then transferred to blenders, where they are mixed according to specified blending rules. The resulting output streams must meet property constraints (e.g., quality specifications) and are subsequently stored in output inventory vessels before being sold, subject to upper bounds on product quantities.

The goal is typically to maximize profit while satisfying constraints on inventory levels, property ranges, and operational rules—such as prohibiting simultaneous inflow and outflow in the same blender. These requirements lead to a highly nonlinear and constrained formulation, commonly modeled as a non-convex mixed-integer quadratically constrained program (MIQCP) (Chen & Maravelias, 2020).

Due to the sequential decision-making structure of the problem and the uncertainty in demands or property variations over time, reinforcement learning (RL) emerges as a promising approach. In particular, safe RL techniques can guide agents to learn actions that are not only near-optimal but also likely to satisfy complex constraints. To that end, we model the problem as a Constrained Markov Decision Process (CMDP) and implement it as the environment BlendingEnv, which supports dynamic decisions on stream flows, purchasing, and selling over time. A schematic of the environment is shown in figure 5.

### A.5.2 PROBLEM SETUP

The sets and known parameters used to describe the environment are shown below:

**Sets**

- $\mathcal{T} = \{1, \ldots, T\}$: Set of discrete time periods.
- $\mathcal{S}$: Set of source streams.
- $\mathcal{J}$: Set of blenders.
- $\mathcal{P}$: Set of demand streams.

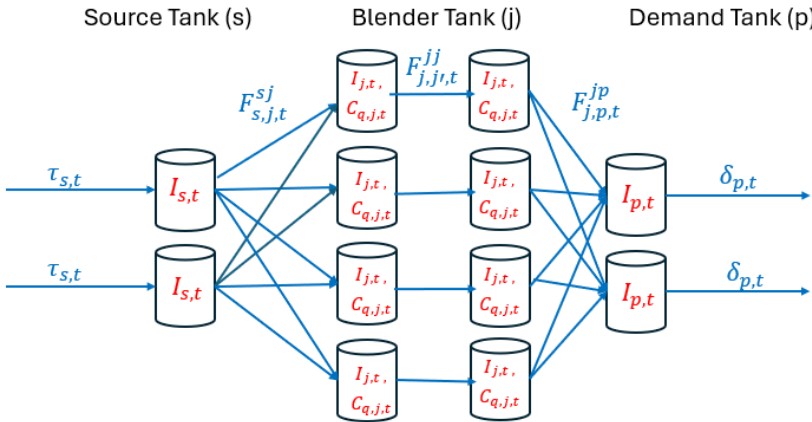

Figure 5: Schematic of BlendingEnv: This is a representation of a blending system with the blue arrows representing the flows between different components (action) and the red variables representing the different inventories and properties of the blender (state)

- $\mathcal{Q}$: Set of stream properties (e.g., chemical or physical characteristics).
- $\mathcal{F}^{s,j}$: Set of tuples representing possible directed flows from source streams to blenders.
- $\mathcal{F}^{j,j}$: Set of tuples representing possible directed flows between blenders. Flows are defined in such a way that only one way is possible.
- $\mathcal{F}^{j,p}$: Set of tuples representing possible directed flows from blenders to demand streams.

**Parameters**

- $F^{\mathrm{max}}$: Upper bound for any flow between nodes.
- $\sigma_{s,q}$: Value of property $q \in \mathcal{Q}$ for source stream $s \in \mathcal{S}$.
- $[s_s^{\mathrm{lb}}, s_s^{\mathrm{ub}}]$: Lower and upper bounds on inventory of source $s \in \mathcal{S}$.
- $\tau_{s,t}^0$: Availability of source stream $s \in \mathcal{S}$ at time $t \in \mathcal{T}$.
- $[b_j^{\mathrm{lb}}, b_j^{\mathrm{ub}}]$: Lower and upper inventory bounds for blender $j \in \mathcal{J}$.
- $[\sigma_{p,q}^{\mathrm{lb}}, \sigma_{p,q}^{\mathrm{ub}}]$: Lower and upper bounds on property $q \in \mathcal{Q}$ for demand stream $p \in \mathcal{P}$.
- $[d_p^{\mathrm{lb}}, d_p^{\mathrm{ub}}]$: Lower and upper bounds on inventory of demand stream $p \in \mathcal{P}$.
- $\delta_{p,t}^0$: Maximum amount of demand stream $p \in \mathcal{P}$ that can be fulfilled at time $t \in \mathcal{T}$.
- $\beta_p^d$: Unit selling price for demand stream $p \in \mathcal{P}$.
- $\beta_s^s$: Unit purchase cost of source stream $s \in \mathcal{S}$.
- $\beta$: Unit cost of intermediate flows (e.g., between nodes).
- $\alpha$: Fixed cost of activating an intermediate flow.
- $k$: Window length for source and demand forecast
- $strategy$: The strategy used in the environment to handle illegal actions caused by constraint violations. We support three options: $prop$, $disable$, and $none$. In the $prop$ strategy, we adjust a subset of actions when they violate the relevant constraints, with actions scaled or clipped depending on the specific constraint. In the $disable$ strategy, a subset of actions is set directly to zero . The $none$ strategy applies no correction, allowing actions to remain unchanged regardless of constraint violations.
- $\lambda_{0,B}$: a fixed cost for violating bound constraint (analogous to $\ell_0$ regularization).

- $\lambda_B$: prefactor to the sum of $\ell_0$ and $\ell_1$ violations for inventory bound constraint (analogous to $\ell_0$ regularization).

- $\lambda_{0,M}$: a fixed cost for violating in-out rule constraint (analogous to $\ell_0$ regularization).

- $\lambda_{0,Q}$: a fixed cost for violating property specifications (analogous to $\ell_0$ regularization).

- $\epsilon$: A small positive threshold used to disregard negligible violations of constraints.

### A.5.3 STATE SPACE

The state $s_t$ at time $t \in \mathcal{T}$ includes the following components, with dimensionality expressed using the sets defined previously:

- Source inventories: $I^s_{s,t} \in [s^{\text{lb}}_s, s^{\text{ub}}_s]$, for all $s \in \mathcal{S}$. This has dimension $|\mathcal{S}|$.

- Blender inventories: $I^b_{j,t} \in [b^{\text{lb}}_j, b^{\text{ub}}_j]$, for all $j \in \mathcal{J}$. This has dimension $|\mathcal{J}|$.

- Demand inventories: $I^d_{p,t} \in [d^{\text{lb}}_p, d^{\text{ub}}_p]$, for all $p \in \mathcal{P}$. This has dimension $|\mathcal{P}|$.

- Blender properties: $C_{j,q,t} \in \mathbb{R}$, for all $j \in \mathcal{J}, q \in \mathcal{Q}$. This has dimension $|\mathcal{J}| \times |\mathcal{Q}|$.

- $\tau^0_{s,t+1:t+k}$: Forecasted source availability for all $s \in \mathcal{S}$ from time t+1 to t+k where $k$ is the forecasting window. This component has a dimension of $|\mathcal{S}| \times k$. Entries are padded with zeros for time periods beyond the episode horizon.

- $\delta^0_{p,t+1:t+k}$ Forecasted demand availability, for all $p \in \mathcal{P}$ from time t+1 to t+k where $k$ is the forecasting window. This component has a dimension of $|\mathcal{P}| \times k$. Entries are padded with zeros for time periods beyond the episode horizon.

- Time step: $t \in \mathcal{T}$. This is a scalar.

### A.5.4 ACTION SPACE

At each time step $t \in \mathcal{T}$, the agent selects the following actions:

- Source purchases: $\tau_{s,t} \in [0, \tau^0_{s,t}]$, for all $s \in \mathcal{S}$. This has dimension $|\mathcal{S}|$.

- Demand sales: $\delta_{p,t} \in [0, \delta^0_{p,t}]$, for all $p \in \mathcal{P}$. This has dimension $|\mathcal{P}|$.

- Source-to-blender flows: $F^{sj}_{s,j,t} \in [0, F^{\max}]$, for all $(s,j) \in \mathcal{F}^{s,j}$. This has dimension $|\mathcal{F}^{s,j}|$.

- Blender-to-blender flows: $F^{jj}_{j,j',t} \in [0, F^{\max}]$, for all $(j,j') \in \mathcal{F}^{j,j}$. This has dimension $|\mathcal{F}^{j,j}|$.

- Blender-to-demand flows: $F^{jp}_{j,p,t} \in [0, F^{\max}]$, for all $(j,p) \in \mathcal{F}^{j,p}$. This has dimension $|\mathcal{F}^{j,p}|$.

Each action is initially normalized to the interval $[-1, 1]$ and mapped to its actual value using affine transformation based on the lower and upper bounds of the corresponding action:

$$a^{\text{actual}} = \frac{(a^{\text{normalized}} + 1)}{2} \cdot (a^{\text{high}} - a^{\text{low}}) + a^{\text{low}}$$

Here, $a^{\text{low}}$ and $a^{\text{high}}$ are the lower and upper bounds for each action dimension as specified above.

### A.5.5 TRANSITION DYNAMICS

After scaling the actions, we increment $t$ to $t + 1$. Let us denote the set of actions at this point as

$$\left\{ \tau^{\text{prebound}}_{s,t}, \quad \delta^{\text{prebound}}_{p,t}, \quad F^{\text{sj,prebound}}_{s,j,t}, \quad F^{\text{jj,preinout}}_{j,j',t}, \quad F^{\text{jp,preinout}}_{j,p,t} \right\}.$$

respectively.

**Checking action for source inventory bounds:** We next proceed to check if the actions provide a source inventory that violates the bounds. If there is a violation, we adjust the actions based on the strategy chosen.

$$I^s_{\text{new},s} = I^s_{s,t-1} - \sum_{(s,j)\in\mathcal{F}^{s,j}} F^{sj,prebound}_{s,j,t} + \tau^{prebound}_{s,t}$$

$$F^{sj}_{s,j,t|(s,j)\in\mathcal{F}^{s,j}} = \begin{cases} F^{sj,prebound}_{s,j,t}\left(\dfrac{I^s_{s,t-1}+\tau^{prebound}_{s,t}-s^{lb}_s}{\sum_{(s,j)\in\mathcal{F}^{s,j}} F^{sj,prebound}_{s,j,t}}\right) & \text{if } I^s_{\text{new},s} < s^{lb}_s - \epsilon, strategy = prop \\ 0 & \text{if } I^s_{\text{new},s} < s^{lb}_s - \epsilon, strategy = disable \\ F^{sj,prebound}_{s,j,t} & \text{otherwise} \end{cases}$$

$$\tau_{s,t} = \begin{cases} \min(s^{ub}_s + \sum_{(s,j)\in\mathcal{F}^{s,j}} F^{sj,prebound}_{s,j,t} - I^s_{s,t-1}, \tau^0_{s,t}) & \text{if } I^s_{\text{new},s} > s^{ub}_s + \epsilon, strategy = prop \\ 0 & \text{if } I^s_{\text{new},s} > s^{ub}_s + \epsilon, strategy = disable \\ \tau^{prebound}_{s,t} & \text{otherwise} \end{cases}$$

**Source inventory updates** We then update the source inventory as follows:

$$I^s_{s,t} = \text{clip}(I^s_{s,t-1} - \sum_{(s,j)\in\mathcal{F}^{s,j}} F^{sj}_{s,j,t} + \tau_{s,t},\ s^{lb}_s,\ s^{ub}_s)$$

**Checking action for in-out rule violations:** We now check if the actions follow the in-out rule for blenders. That is, blenders should not have simultaneous inflow and outflow of streams. If there is such a case, we set all corresponding outflows to 0. The value of other flows are not changed.

$$\text{Inflow}_j = \sum_{(s,j)\in\mathcal{F}^{s,j}} F^{sj}_{s,j,t} + \sum_{(j',j)\in\mathcal{F}^{j,j}} F^{jj,preinout}_{j',j,t}$$

$$\text{Outflow}_j = \sum_{(j,j')\in\mathcal{F}^{j,j}} F^{jj,preinout}_{j,j',t} + \sum_{(j,p)\in\mathcal{F}^{j,p}} F^{jp,preinout}_{j,p,t}$$

$$F^{jp,prebound}_{j,p,t|(j,p)\in\mathcal{F}^{j,p}} = \begin{cases} 0 & \text{if Inflow}_j > \epsilon, \text{Outflow}_j > \epsilon, strategy \neq no \\ F^{jp,preinout}_{j,p,t} & \text{otherwise} \end{cases}$$

$$F^{jj,prebound}_{j,j',t|(j,j')\in\mathcal{F}^{j,j}} = \begin{cases} 0 & \text{if Inflow}_j > \epsilon, \text{Outflow}_j > \epsilon, strategy \neq no \\ F^{jj,preinout}_{j,j',t} & \text{otherwise} \end{cases}$$

**Checking action for blender inventory lower bound:** We next proceed to check if the actions provide a blender inventory that violates the lower bounds.

$$I^b_{\text{new},j} = I^b_{j,t-1} + \sum_{(s,j)\in\mathcal{F}^{s,j}} F^{sj}_{s,j,t} + \sum_{(j',j)\in\mathcal{F}^{j,j}} F^{jj,prebound}_{j',j,t}$$
$$- \sum_{(j,j')\in\mathcal{F}^{j,j}} F^{jj,prebound}_{j,j',t} - \sum_{(j,p)\in\mathcal{F}^{j,p}} F^{jp,prebound}_{j,p,t}$$

If there is a violation, we adjust the actions based on the strategy chosen. We keep the same actions otherwise.

$$F^{jp}_{j,p,t|(j,p)\in\mathcal{F}^{j,p}} = \begin{cases} F^{jp,prebound}_{j,p,t} F^{j,prop}_{j,t} & \text{if } I^b_{\text{new},j} < b^{lb}_j - \epsilon, \\ & strategy = prop \\ 0 & \text{if } I^b_{\text{new},j} < b^{lb}_j - \epsilon, \\ & strategy = disable \\ F^{jp,prebound}_{j,p,t} & \text{otherwise} \end{cases}$$

$$F^{jj}_{j,j',t|(j,j')\in\mathcal{F}^{j,j}} = \begin{cases} F^{jj,prebound}_{j,j',t}F^{j,prop}_{j,t} & \text{if } I^b_{\text{new},j} < b^{lb}_j - \epsilon, \\ & strategy = prop \\ 0 & \text{if } I^b_{\text{new},j} < b^{lb}_j - \epsilon, \\ & strategy = disable \\ F^{jj,prebound}_{j,j',t} & \text{otherwise} \end{cases}$$

$$F^{j,prop}_{j,t} = \frac{I^b_{j,t-1} + \sum_{(s,j)\in\mathcal{F}^{s,j}} F^{sj}_{s,j,t} + \sum_{(j',j)\in\mathcal{F}^{j,j}} F^{jj,prebound}_{j',j,t} - b^{lb}_j}{\sum_{(j,j')\in\mathcal{F}^{j,j}} F^{jj,prebound}_{j,j',t} + \sum_{(j,p)\in\mathcal{F}^{j,p}} F^{jp,prebound}_{j,p,t}}$$

**Blender inventory updates:** After making the adjustments described above, we update the blender inventory.

$$I^b_{j,t} = \text{clip}(I^b_{j,t-1} + \sum_{(s,j)\in\mathcal{F}^{s,j}} F^{sj}_{s,j,t} + \sum_{(j',j)\in\mathcal{F}^{j,j}} F^{jj}_{j',j,t} - \sum_{(j,j')\in\mathcal{F}^{j,j}} F^{jj}_{j,j',t} - \sum_{(j,p)\in\mathcal{F}^{j,p}} F^{jp}_{j,p,t},\ b^{lb}_j,\ b^{ub}_j)$$

**Checking action for demand inventory lower bound:** We now check if the actions provide a demand inventory that violates the lower bounds.

$$I^d_{\text{new},p} = I^d_{p,t-1} + \sum_{(j,p)\in\mathcal{F}^{j,p}} F^{jp}_{j,p,t} - \delta^{prebound}_{p,t}$$

If there is a violation, we adjust the actions based on the strategy chosen.

$$\delta_{p,t} = \begin{cases} I^d_{p,t-1} + \sum_{(j,p)\in\mathcal{F}^{j,p}} F^{jp}_{j,p,t} - d^{lb}_p & \text{if } I^d_{\text{new},p} < d^{lb}_p - \epsilon, strategy = prop \\ 0 & \text{if } I^d_{\text{new},p} < d^{lb}_p - \epsilon, strategy = disable \\ \delta^{prebound}_{p,t} & \text{otherwise} \end{cases}$$

**Demand inventory updates:** After making the adjustments described above, we update the demand inventory.

$$I^d_{p,t} = \text{clip}(I^d_{p,t-1} + \sum_{(j,p)\in\mathcal{F}^{j,p}} F^{jp}_{j,p,t} - \delta_{p,t},\ d^{lb}_p,\ d^{ub}_p)$$

**Blender property updates** We then update the property of the materials in each blender as follows:

$$C_{j,q,t} = \begin{cases} = \frac{1}{I^b_{j,t}}\Big(C_{j,q,t-1}\cdot I^b_{j,t-1} + \sum_{(s,j)\in\mathcal{F}^{s,j}} F^{sj}_{s,j,t}\cdot\sigma_{s,q} \\ + \sum_{(j',j)\in\mathcal{F}^{j,j}} F^{jj}_{j',j,t}\cdot C_{j',q,t-1} \\ - \sum_{(j,j')\in\mathcal{F}^{j,j}} F^{jj}_{j,j',t}\cdot C_{j,q,t-1} \\ - \sum_{(j,p)\in\mathcal{F}^{j,p}} F^{jp}_{j,p,t}\cdot C_{j,q,t-1}\Big) & \text{if } I^b_{j,t+1} > \epsilon \\ 0 & \text{otherwise} \end{cases}$$

We then proceed to calculate the cost and reward. Finally, we shift the future schedules forecast by one step to $\tau^0_{s,t+1:t+k}$ and $\delta^0_{p,t+1,t+k}$.

### A.5.6 COSTS

The total cost at each time step consists of penalties for constraint violations, including those for the violations of the bounds of the state and the in-out rule. The components of the cost are defined as follows:

**Inventory Bound Violation Costs** :

Inventory levels at sources, blenders, and demands are required to remain within prescribed upper and lower bounds. Inventory levels computed with actions before the relevant upadates are used to assess constraint violations.

**Sources**:

$$C_{s,t}^{source} = \begin{cases} \lambda_B \cdot (\lambda_{0,B} + I_{new,s}^s - s_s^{ub}), & \text{if } I_{new,s}^s > s_s^{ub} + \epsilon \\ \lambda_B \cdot (\lambda_{0,B} + s_s^{lb} - I_{new,s}^s), & \text{if } I_{new,s}^s < s_s^{lb} - \epsilon \\ 0, & \text{otherwise} \end{cases}$$

**Blenders**:

$$C_{j,t}^{blender} = \begin{cases} \lambda_B \cdot (\lambda_{0,B} + I_{new,j}^b - b_j^{ub}), & \text{if } I_{new,j}^b > b_j^{ub} + \epsilon \\ \lambda_B \cdot (\lambda_{0,B} + b_s^{lb} - I_{new,j}^b), & \text{if } I_{new,j}^b < b_j^{lb} - \epsilon \\ 0, & \text{otherwise} \end{cases}$$

**Demands**:

$$C_{p,t}^{demand} = \begin{cases} \lambda_B \cdot (\lambda_{0,B} + I_{new,p}^d - d_p^{ub}), & \text{if } I_{new,p}^d > d_p^{ub} + \epsilon \\ \lambda_B \cdot (\lambda_{0,B} + d_p^{lb} - I_{new,j}^b), & \text{if } I_{new,p}^d < d_p^{lb} - \epsilon \\ 0, & \text{otherwise} \end{cases}$$

**In-Out Rule Violation Costs** :

Blenders must not simultaneously receive and send flow within the same time step. If both incoming and outgoing flows are non-zero (beyond a tolerance $\epsilon$), a penalty is applied:

$$C_{j,t}^{in-out} = \begin{cases} \lambda_{0,M}, & \text{if } \text{Inflow}_j > \epsilon, \text{Outflow}_j > \epsilon \\ 0, & \text{otherwise} \end{cases}$$

**Property Violation Costs** :

Each demand $p$ requires product properties $q$ within bounds. If blender $j$'s content violates these bounds and is being delivered to $p$, a penalty is applied:

$$C_{j,q,p,t}^Q = \begin{cases} \lambda_{0,Q}, & \text{if } (C_{j,q,t} < \sigma_{p,q}^{lb} - \epsilon \vee C_{j,q,t} > \sigma_{p,q}^{ub} + \epsilon), (j,p) \in \mathcal{F}^{j,p}, F_{j,p,t}^{jp} > 0 \\ 0, & \text{otherwise} \end{cases}$$

**Total Cost**  The total penalty at time $t$ is:

$$C_t = \sum_{s \in S} C_{s,t}^{source} + \sum_{j \in J} C_{j,t}^{blender} + \sum_{p \in P} C_{p,t}^{demand} + \sum_{j \in J} C_{j,t}^{in-out} + \sum_{j \in J} \sum_{q \in Q} \sum_{p \in P} C_{j,q,p,t}^Q$$

A.5.7  REWARD FUNCTION

The reward function consists of:

**Revenue from selling streams** :

$$\pi_t^{sale} = \sum_{p \in P} \beta_p^d \cdot \delta_{p,t}$$

**Contribution to reward from buying streams** :

$$\pi_t^{purchase} = -\sum_{s \in S} \beta_s^s \cdot \tau_{s,t}$$

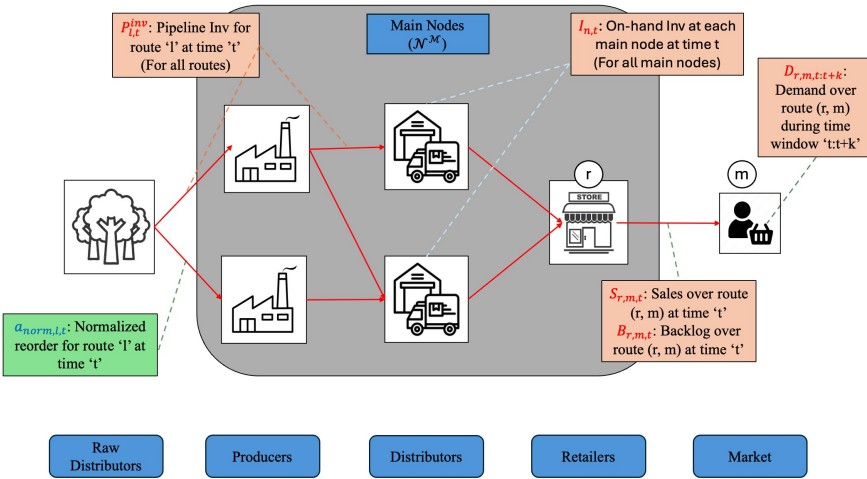

Figure 6: Schematic of *InvMgmtEnv*. A multi-echelon supply-chain network with raw distributors, producers, distributors, and retailers serving a market. Directed red arrows depict transportation routes $\ell \in \mathcal{L}$. Red callouts indicate observation components: *on-hand inventory* $I_{n,t}$ for all main nodes $n \in \mathcal{N}^{\mathcal{M}}$; *pipeline inventory* $P_{\ell,t}^{\mathrm{inv}}$ (time-indexed along each route $\ell$); *demand window* $D_{r,m,t:t+k}$ on retail–market pairs $(r,m) \in \mathcal{RM}$; *sales* $S_{r,m,t}$; and *backlog* $B_{r,m,t}$. The blue callout marks the action—the *normalized reorder* $a_{\mathrm{norm},\ell,t}$ on each route $\ell$. The shaded region highlights the main-node set $\mathcal{N}^{\mathcal{M}}$

**Contribution to reward from flows** :

$$\pi_t^{flow} = -\alpha \cdot Q_{bin} - \beta \cdot Q_{float}$$

where

$$Q_{bin} = \sum_{(s,j) \in \mathcal{F}^{s,j}} \mathbb{1}[F_{s,j,t}^{sj} > 0] + \sum_{(j,p) \in \mathcal{F}^{j,p}} \mathbb{1}[F_{j,p,t}^{jp} > 0]$$

$$Q_{float} = \sum_{(s,j) \in \mathcal{F}^{s,j}} F_{s,j,t}^{sj} + \sum_{(j,p) \in \mathcal{F}^{j,p}} F_{j,p,t}^{jp}$$

**Total Reward** :

$$\pi_t = \pi_t^{sale} + \pi_t^{purchase} + \pi_t^{flow}$$

## A.6 MULTI-ECHELON INVENTORY MANAGEMENT ENVIRONMENT (INVMGMTENV)

### A.6.1 OVERVIEW

The multi-echelon inventory management problem involves coordinating replenishment orders across a five-tier supply network—raw-material suppliers, producers, distributors, retailers, and end-markets—under time-varying demand. At each period, the decision maker chooses continuous order quantities along each transportation route, with orders subject to fixed lead times before arrival. On-hand and in-transit inventories incur holding costs, while unmet demand is backlogged and penalized. The objective is to maximize cumulative net profit—total sales revenue minus procurement, operating, holding, and shortage penalty costs—over a finite planning horizon.

We adopt a centralized, single-product framework (Perez et al., 2021) in which all replenishment decisions are made by a central planner, and customer demand at each retailer-to-market link is drawn from a known stationary distribution, such as a Gaussian with specified mean and variance. Transportation is lossless with deterministic lead times, and replenishment quantities are bounded by per-route capacity, with any excess actions penalized. Episodes terminate after a fixed number of periods. A schematic of the environment is shown in figure 6.

A.6.2   PROBLEM SETUP

The sets and known parameters used to describe `InvMgmtEnv` are shown below.

**Sets**

- $\mathcal{T} = \{1, \ldots, T\}$: set of discrete time periods.
- $\mathcal{M}$: set of market nodes.
- $\mathcal{R}$: set of retailer nodes.
- $\mathcal{D}$: set of distributor nodes.
- $\mathcal{P}$: set of producer nodes.
- $\mathcal{S}$: set of raw-material supplier nodes.
- $\mathcal{N} = \mathcal{M} \cup \mathcal{R} \cup \mathcal{D} \cup \mathcal{P} \cup \mathcal{S}$: set of all nodes.
- $\mathcal{N}^{\mathcal{M}} = \mathcal{R} \cup \mathcal{D} \cup \mathcal{P}$: set of main nodes.
- $\mathcal{L}$: set of directed replenishment routes. Each route is written

$$\ell = \big(\mathrm{orig}(\ell), \, \mathrm{dest}(\ell)\big),$$

  an ordered pair of nodes with $\mathrm{orig}(\ell) \in \mathcal{N}$ the origin (shipping) node and $\mathrm{dest}(\ell) \in \mathcal{N}$ the destination (receiving) node.
- $\mathcal{R}\mathcal{M}$: set of retailer-to-market demand links.

**Parameters**

- $I_n^{\mathrm{init}}$, $n \in \mathcal{N}$: initial on-hand inventory at node $n$.
- $\mathrm{Cap}_\ell^{\mathrm{route}}$, $\ell \in \mathcal{L}$: capacity limit of route $\ell$.
- $C_\ell^{\mathrm{hold,mat}}$, $\ell \in \mathcal{L}$: per-unit holding cost for pipeline inventory on route $\ell$.
- $C_\ell$, $\ell \in \mathcal{L}$: procurement cost per unit ordered on route $\ell$.
- $LT_\ell$, $\ell \in \mathcal{L}$: fixed lead time (in periods) for route $\ell$.
- $C_n^{\mathrm{hold,inv}}$, $n \in \mathcal{N}$: per-unit holding cost for on-hand inventory at node $n$.
- $C_p^{\mathrm{oper}}$, $p \in \mathcal{P}$: operating cost per unit of production activity at producer $p$.
- $\eta_p$, $p \in \mathcal{P}$: production yield at producer $p$.
- $\mu_{r,m}$, $\sigma_{r,m}$, $(r,m) \in \mathcal{R}\mathcal{M}$: mean and standard deviation of demand on link $(r,m)$.
- $P_{r,m}$, $(r,m) \in \mathcal{R}\mathcal{M}$: selling price per unit on link $(r,m)$.
- $C_{r,m}^{\mathrm{penalty}}$, $(r,m) \in \mathcal{R}\mathcal{M}$: penalty cost per unit of unmet demand on link $(r,m)$.
- $k$: look-ahead window length for demand forecast.
- $\varepsilon$: small threshold below which reorder quantities are treated as zero.
- $\phi_{\mathrm{action}}$: penalty factor for action-bound violations.
- $\phi_{\mathrm{on\_hand}}$: penalty factor for on-hand inventory violations.
- $\phi_{\mathrm{pipeline}}$: penalty factor for pipeline inventory violations.
- $\phi_{\mathrm{sales}}$: penalty factor for sales-state violations.
- $\phi_{\mathrm{backlog}}$: penalty factor for backlog-state violations.

A.6.3   STATE SPACE

The state $s_t$ at time $t \in \mathcal{T}$ is represented by the tuple

$$s_t = \big(I_{n,t}, \, P_{\ell,t}^{\mathrm{inv}}, \, S_{r,m,t}, \, B_{r,m,t}, \, D_{r,m,t:t+k}, \, t\big),$$

where:

- *On-hand inventory levels:*
$$I_{n,t} \in \mathbb{R}_{\geq 0}, \quad n \in \mathcal{N}^{\mathcal{M}},$$
contributing $|\mathcal{N}^{\mathcal{M}}|$ dimensions.

- *Pipeline inventory:*
$$P_{\ell,t}^{\text{inv}} = \big(P_{\ell,1,t}, \ldots, P_{\ell,LT_\ell,t}\big), \quad \ell \in \mathcal{L},$$
contributing $\sum_{\ell \in \mathcal{L}} LT_\ell$ dimensions.

- *Sales:*
$$S_{r,m,t} \in \mathbb{R}_{\geq 0}, \quad (r,m) \in \mathcal{RM},$$
contributing $|\mathcal{RM}|$ dimensions.

- *Backlog:*
$$B_{r,m,t} \in \mathbb{R}_{\geq 0}, \quad (r,m) \in \mathcal{RM},$$
contributing $|\mathcal{RM}|$ dimensions.

- *Demand window:*
$$D_{r,m,t:t+k} = \big(D_{r,m,t}, \ldots, D_{r,m,t+k}\big), \quad (r,m) \in \mathcal{RM},$$
where, for offsets $h$ with $t + h > T$, the unavailable future demand $D_{r,m,t+h}$ is padded with $0$. This contributes $|\mathcal{RM}| \times k$ dimensions.

- *Time step:* (optional) the scalar $t$, contributing $1$ dimension.

All components are concatenated and flattened into a continuous observation vector of total dimension

$$|\mathcal{N}^{\mathcal{M}}| \;+\; \sum_{\ell \in \mathcal{L}} LT_\ell \;+\; 2\,|\mathcal{RM}| \;+\; |\mathcal{RM}| \times k \;+\; 1.$$

### A.6.4 ACTION SPACE

At each decision epoch $t \in \mathcal{T}$ the agent outputs a $|\mathcal{L}|$–tuple of normalized actions

$$\mathbf{a}_{\text{norm},t} \;=\; \big(a_{\text{norm},\ell,t}\big)_{\ell \in \mathcal{L}} \;\in\; [-1,1]^{|\mathcal{L}|},$$

whose components correspond one-to-one with the directed replenishment routes $\ell \in \mathcal{L}$.

**Scaling to Physical Reorder Quantities.** Each normalized component is linearly mapped to a true reorder quantity, respecting the capacity of its route:

$$Q_{\ell,t} \;=\; \frac{a_{\text{norm},\ell,t} + 1}{2}\, \text{Cap}_\ell^{\text{route}}, \qquad \ell \in \mathcal{L}.$$

A small-order cutoff $\varepsilon$ is then applied:

$$Q_{\ell,t} \;=\; \begin{cases} 0, & Q_{\ell,t} \leq \varepsilon, \\ Q_{\ell,t}, & \text{otherwise.} \end{cases}$$

Collecting all routes, the environment works with the physical reorder vector

$$\mathbf{Q}_t \;=\; \big(Q_{\ell,t}\big)_{\ell \in \mathcal{L}} \;\in\; \big[0,\, \text{Cap}_\ell^{\text{route}}\big]^{|\mathcal{L}|}.$$

**Effective Action Space.** After scaling and cutoff, the admissible actions lie in the $|\mathcal{L}|$-dimensional box

$$\mathcal{A} \;=\; \Big\{ \mathbf{Q}_t \;\big|\; Q_{\ell,t} \in \big[0, \text{Cap}_\ell^{\text{route}}\big], \; \ell \in \mathcal{L} \Big\}.$$

### A.6.5 TRANSITION DYNAMICS

At each time step $t \in \mathcal{T}$, after observing $s_t$ and selecting normalized actions $\mathbf{a}_{\text{norm}} \in [-1,1]^{|\mathcal{L}|}$, the environment updates to $s_{t+1}$ as follows:

**Action Processing, Clipping, and Penalty Recording.** Each component $a_{\mathrm{norm},\ell}$ is first mapped to a preliminary reorder quantity

$$Q_{\ell,t}^{\mathrm{pre}} = \frac{a_{\mathrm{norm},\ell} + 1}{2}\,\mathrm{Cap}_\ell^{\mathrm{route}}.$$

A small-order cutoff $\varepsilon$ is applied:

$$Q_{\ell,t}^{\mathrm{cut}} = \begin{cases} 0, & Q_{\ell,t}^{\mathrm{pre}} \le \varepsilon, \\ Q_{\ell,t}^{\mathrm{pre}}, & \text{otherwise.} \end{cases}$$

Any remaining violation of the action bounds is then clipped and recorded:

$$Q_{\ell,t} = \begin{cases} 0, & Q_{\ell,t}^{\mathrm{cut}} < 0, \\ Q_{\ell,t}^{\mathrm{cut}}, & 0 \le Q_{\ell,t}^{\mathrm{cut}} \le \mathrm{Cap}_\ell^{\mathrm{route}}, \\ \mathrm{Cap}_\ell^{\mathrm{route}}, & Q_{\ell,t}^{\mathrm{cut}} > \mathrm{Cap}_\ell^{\mathrm{route}}, \end{cases}$$

and the action-bound violation penalty is computed as

$$C_{\ell,t}^{\mathrm{bound,action}} = \begin{cases} \phi_{\mathrm{action}}\,\bigl|Q_{\ell,t}^{\mathrm{cut}}\bigr|, & Q_{\ell,t}^{\mathrm{cut}} < 0, \\ \phi_{\mathrm{action}}\,\bigl|Q_{\ell,t}^{\mathrm{cut}} - \mathrm{Cap}_\ell^{\mathrm{route}}\bigr|, & Q_{\ell,t}^{\mathrm{cut}} > \mathrm{Cap}_\ell^{\mathrm{route}}, \\ 0, & \text{otherwise,} \end{cases}$$

with $\phi_{\mathrm{action}}$ the penalty factor for action violations. Finally, increment the period: $t \leftarrow t + 1$.

**Pipeline Update.** After incrementing the period $t \leftarrow t + 1$ at the end of previous step, the in-transit slots are shifted and the newest order is injected:

$$P_{\ell,\tau,t} = \begin{cases} P_{\ell,\tau+1,\,t-1}, & \tau = 1, 2, \ldots, LT_\ell - 1, \\ Q_{\ell,\,t-1}, & \tau = LT_\ell, \\ 0, & \text{otherwise.} \end{cases}$$

The total in-transit inventory on route $\ell$ is

$$P_{\ell,t}^{\mathrm{inv}} = \sum_{\tau=1}^{LT_\ell} P_{\ell,\tau,t},$$

and the arrivals at node $n$ are

$$\mathrm{Arrivals}_{n,t} = \sum_{\substack{\ell \in \mathcal{L} \\ \mathrm{dest}(\ell)=n}} P_{\ell,1,t}.$$

**Arrival and Inventory Update.** For each node $n \in \mathcal{N}$,

$$I_{n,t} = I_{n,t-1} + \mathrm{Arrivals}_{n,t}.$$

**4. Demand Realization.** For each $(r,m) \in \mathcal{RM}$,

$$D_{r,m,t} \sim \mathcal{N}(\mu_{r,m}, \sigma_{r,m}).$$

**Sales and Backlog Update.** Sales are

$$S_{r,m,t} = \min\bigl\{ D_{r,m,t} + B_{r,m,t-1},\, I_{r,t} \bigr\},$$

then

$$I_{r,t} \leftarrow I_{r,t} - S_{r,m,t}, \quad B_{r,m,t} = D_{r,m,t} + B_{r,m,t-1} - S_{r,m,t}.$$

**Demand Forecast Shift.**

$$D_{r,m,t+1:t+k} = \bigl(D_{r,m,t+1}, \ldots, D_{r,m,t+k}\bigr) \quad \forall\,(r,m) \in \mathcal{RM}.$$

**Observation Clipping.** After constructing all next-state components, any $x_{i,t+1}$ outside its bounds $[L_i, U_i]$ is clipped to $L_i$ or $U_i$ as appropriate. Each such clipping is recorded as an observation-bound violation and will incur the corresponding penalty in the Cost Function.

### A.6.6 COST FUNCTION

The total cost at time $t$ is the sum of the penalties incurred for any state-observation violations:

$$C_t = C_t^{\text{on\_hand}} + C_t^{\text{pipeline}} + C_t^{\text{sales}} + C_t^{\text{backlog}}$$

Each category cost $C_t^c$ (for $c \in \{\text{on\_hand}, \text{pipeline}, \text{sales}, \text{backlog}\}$) is computed by applying the same piecewise linear penalty to each pre-clipped component $x_{i,t}$ in that category. Let $\mathcal{I}_c$ be the set of component indices for category $c$, each with bounds $[L_i, U_i]$ and penalty factor $\phi_c$. For each category $c \in \{\text{on\_hand}, \text{pipeline}, \text{sales}, \text{backlog}\}$, let $\phi_c$ denote its penalty factor (e.g. $\phi_{\text{pipeline}}$ for pipeline-inventory violations). Then

$$C_t^c = \sum_{i \in \mathcal{I}_c} \begin{cases} \phi_c \left(L_i - x_{i,t}\right), & x_{i,t} < L_i, \\ \phi_c \left(x_{i,t} - U_i\right), & x_{i,t} > U_i, \\ 0, & \text{otherwise,} \end{cases}$$

so that any observation below its lower bound or above its upper bound contributes linearly to the total cost.

**Action-bound penalty.** If a preliminary order $Q_{\ell,t}^{\text{cut}}$ violates the interval $\left[0, \text{Cap}_\ell^{\text{route}}\right]$ (Step 1 of the transition dynamics), it is clipped to the nearest bound and the deviation is recorded as an *action-bound penalty* $C_{\ell,t}^{\text{bound,action}}$, weighted by the scalar factor $\phi_{\text{action}}$. The sum over all routes yields $C_t^{\text{action}}$ is added to the total cost $C_t$ above.

### A.6.7 REWARD FUNCTION

We express the per-step reward as the sum of five components, each denoted by $\Pi$:

$$\Pi_t = \Pi_t^{\text{rev}} - \Pi_t^{\text{proc}} - \Pi_t^{\text{hold}} - \Pi_t^{\text{oper}} - \Pi_t^{\text{backlog}}.$$

- $\Pi_t^{\text{rev}} = \sum_{(r,m) \in \mathcal{RM}} S_{r,m,t} \, P_{r,m}$, revenue from sales.

- $\Pi_t^{\text{proc}} = \sum_{\ell \in \mathcal{L}} Q_{\ell,t} \, C_\ell$, procurement cost for orders.

- $\Pi_t^{\text{hold}} = \sum_{n \in \mathcal{N}} I_{n,t} \, C_n^{\text{hold,inv}} + \sum_{\ell \in \mathcal{L}} P_{\ell,t}^{\text{inv}} \, C_\ell^{\text{hold,mat}}$, holding cost for on-hand and pipeline inventories.

- $\Pi_t^{\text{oper}} = \sum_{p \in \mathcal{P}} \dfrac{C_p^{\text{oper}}}{\eta_p} \sum_{\substack{\ell \in \mathcal{L} \\ \text{orig}(\ell)=p}} Q_{\ell,t}$, operating cost at each producer $p$, proportional to the total quantity $Q_{\ell,t}$ dispatched along routes $\ell$ that originate at $p$; the factor $1/\eta_p$ converts finished-good output to required production input.

- $\Pi_t^{\text{backlog}} = \sum_{(r,m) \in \mathcal{RM}} B_{r,m,t} \, C_{r,m}^{\text{penalty}}$, penalty for backlog.

### A.6.8 EPISODE TERMINATION

An episode ends when the final time step $t = T$ is reached. It is *not truncated*, even if the system encounters infeasible states.

## A.7 GRID-INTEGRATED ENERGY STORAGE (GRIDSTORAGEENV)

### A.7.1 OVERVIEW

The grid-integrated energy-storage environment (GridStorageEnv) models the hourly operation of a transmission-connected battery fleet co-located with conventional generators on a network subject to time-varying loads and deterministic line de-energisation schedules (Piansky et al., 2024). At each period, the agent simultaneously chooses (i) real-power outputs for every thermal generator, (ii) battery charge rates, (iii) battery discharge rates, (iv) deliberate load shedding, and (v) bus voltage angles at all non-reference buses (Bus 1 is fixed at 0 rad). Given these angles, line flows are computed through DC power-flow equations, and flows on deterministically de-energised lines are forced to zero.

Batteries follow a "bucket" state-of-charge (SOC) dynamic that applies charging/discharging inefficiency and an inter-period carry-over factor.

The objective is to minimise total cost over a finite horizon of $T$ hours, comprising (a) generator fuel cost modelled as a polynomial in power output and (b) linear penalties on slack generation, unserved load, bus-angle limits, line-loading ratio, SOC and demand violations, as well as a nodal power-balance penalty that discourages infeasible angle choices. Physical limits on generators, transmission lines, buses, and batteries are enforced by clipping out-of-bounds actions or state variables and charging a proportional penalty.

We assume perfect foresight of hourly demand and line de-energisation schedules, a single central decision maker, lossless transmission on energised lines, and a fixed $k$-hour demand-forecast window embedded in the state. Episodes last exactly $T$ steps and are never truncated. A schematic of the environment is shown in figure 7.

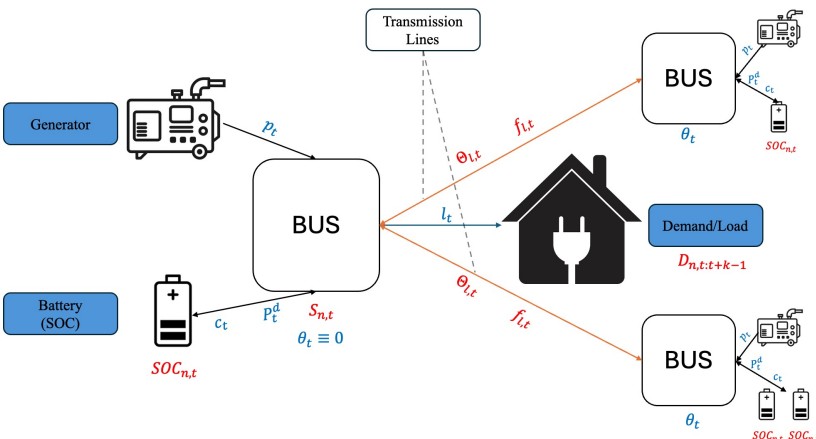

Figure 7: Schematic of *GridStorageEnv*. A power grid with buses, generators, and battery storage, subject to time-varying demand and deterministic line de-energization. Agent actions (blue) at each time step include generator output $p_{g,t}$, battery charge $c_{n,t}$, discharge $p_{n,t}^d$, load shed $\ell_{n,t}$, and voltage angles $\theta_{n,t}$. Observed state (red) comprises battery SOC $SOC_{n,t}$, line flows $f_{\ell,t}$, voltage-angle differences $\Theta_{\ell,t}$, slack $s_{n,t}$, and a $k$-period demand forecast $D_{n,t:t+k-1}$.

### A.7.2 PROBLEM SETUP

The environment is defined by the following sets and parameters.

**Sets**

- $\mathcal{T} = \{1, \ldots, T\}$: discrete time periods.
- $\mathcal{N}$: set of buses in the network, $|\mathcal{N}| = N$.

- $\mathcal{G}$: set of generators, $|\mathcal{G}| = G$.
- $\mathcal{L} \subseteq \mathcal{N} \times \mathcal{N}$: set of transmission lines.
- $\mathcal{D}_t \subseteq \mathcal{L}$: deterministic subset of lines de-energized at time $t$.

**Parameters**

- BusGeneratorLink: a mapping $g \mapsto n$ for $g \in \mathcal{G}$, $n \in \mathcal{N}$, indicating which bus each generator sits on.
- $B_{ij}$, $(i,j) \in \mathcal{L}$: line susceptance.
- $\overline{f}_\ell$, $\ell \in \mathcal{L}$: maximum power-flow on line $\ell$.
- $\underline{\theta}_\ell$, $\overline{\theta}_\ell$, $\ell \in \mathcal{L}$: bounds on voltage-angle difference.
- $d_{n,t}$, $(n,t) \in \mathcal{N} \times \{1, \ldots, T\}$: demand at bus $n$ and time $t$.
- $p_g^{\min}$, $p_g^{\max}$, $g \in \mathcal{G}$: generator output limits.
- $E_n^{\min}$, $E_n^{\max}$, $n \in \mathcal{N}$: battery state-of-charge (SOC) bounds.
- $E_{n,0}$, $n \in \mathcal{N}$: initial SOC.
- $p_n^{c,\min}$, $p_n^{c,\max}$, $n \in \mathcal{N}$: battery charge-rate bounds.
- $p_n^{d,\min}$, $p_n^{d,\max}$, $n \in \mathcal{N}$: battery discharge-rate bounds.
- $\eta$: battery charge/discharge efficiency.
- $\gamma$: SOC carry-over rate between periods.
- PolynomialDegree: number of generator-cost coefficients (the highest exponent is PolynomialDegree $- 1$).
- $C_{g,j}$, $g \in \mathcal{G}$, $j = 0, \ldots, \text{PolynomialDegree} - 1$: generator cost coefficients.
- $K_{\text{slack}}$ (Kslack): per-unit penalty for slack generation.
- $K_{\text{ls}}$ (Kls): per-unit penalty for load-shedding.
- $\Theta^{\max}$: absolute bound on controllable bus voltage angle (radians).
- $\phi_{\text{bal}}$: penalty factor for nodal power-balance violations arising when injections do not match DC power flow.
- $\phi_{\theta,\text{act}}$: penalty factor for node-angle *action* clipping.
- $\phi_{\text{power}}$: penalty factor for generator output bounds violations.
- $\phi_{\text{charge}}$: penalty factor for battery charge-rate bounds violations.
- $\phi_{\text{discharge}}$: penalty factor for battery discharge-rate bounds violations.
- $\phi_{\text{slack}}$: penalty factor for slack generation bounds violations.
- $\phi_{\text{shed}}$: penalty factor for load-shedding bounds violations.
- $\phi_{\text{soc}}$: penalty factor for state-of-charge observation bounds violations.
- $\phi_\theta$: penalty factor for voltage-angle observation-bound violations.
- $s^{\max}$: maximum slack-generation allowed at each bus.
- $d_{\text{global}}^{\max} = \max\limits_{n \in \mathcal{N},\, t \in \mathcal{T}} d_{n,t}$: system-wide peak demand, used as the upper bound for all load-shedding actions.

A.7.3   STATE SPACE

The state $s_t$ at time $t \in \mathcal{T}$ is the tuple

$$s_t = \big(\text{SOC}_{n,t},\ \Theta_{\ell,t},\ f_{\ell,t},\ s_{n,t},\ D_{n,t:t+k-1},\ \tau_t\big),$$

with components defined and sized as follows:

- $\text{SOC}_{n,t} = E_{n,t}/E_n^{\max} \in [0,1]$ (normalized battery state of charge at bus $n$; dimension $N$).

- $\Theta_{\ell,t} = \frac{2(\theta_{i,t}-\theta_{j,t})-(\underline{\theta}_\ell+\overline{\theta}_\ell)}{\overline{\theta}_\ell-\underline{\theta}_\ell}$ (normalized voltage-angle difference for $\ell = (i,j)$; dimension $L$).

- $f_{\ell,t} \in [-\overline{f}_\ell, \overline{f}_\ell]$ (actual power flow on line $\ell$ in MW; dimension $L$).

- $s_{n,t} \in [0, s^{\max}]$ (slack generation at bus $n$; dimension $N$).

- $D_{n,t:t+k-1} = (d_{n,t}, d_{n,t+1}, \ldots, d_{n,t+k-1})$ (length-$k$ demand-forecast window; dimension $N \times k$).

- $\tau_t = (t-1)/(T-1) \in [0,1]$ (normalized time index; dimension 1).

All components combine into a flattened observation vector of length

$$2N \;+\; 2L \;+\; Nk \;+\; 1.$$

### A.7.4 ACTION SPACE

At the start of each period $t$ the agent chooses the vector

$$a_t = \left(p_t,\; c_t,\; p_t^d,\; \ell_t,\; \theta_t\right)^\top, \qquad \theta_{1,t} \equiv 0.$$

Its components obey the compact interval constraints

$$
\begin{aligned}
p_t &\in \left[p^{\min}, p^{\max}\right], & &\text{(generator outputs, dim. } G), \\
c_t &\in \left[p^{c,\min}, p^{c,\max}\right], & &\text{(battery charge, } N), \\
p_t^d &\in \left[p^{d,\min}, p^{d,\max}\right], & &\text{(battery discharge, } N), \\
\ell_t &\in [0, d_{\text{global}}^{\max}]^N, & &\text{(load shedding, } N), \\
\theta_t &\in [-\Theta^{\max}, \Theta^{\max}]^{N-1}, & &\text{(bus angles, } N-1).
\end{aligned}
$$

**Normalized interface.** The policy operates in the cube $[-1,1]^{G+3N+(N-1)}$ and outputs $a_{\text{norm},t}$. An affine map rescales it to a preliminary action

$$a_{\text{pre},t} = \tfrac{1}{2}\left(a_{\text{norm},t}+1\right) \odot (a_{\max} - a_{\min}) + a_{\min},$$

with block-wise bounds

$$a_{\min} = \left(p^{\min}, p^{c,\min}, p^{d,\min}, 0_N, -\Theta^{\max}1_{N-1}\right)^\top$$

$$a_{\max} = \left(p^{\max}, p^{c,\max}, p^{d,\max}, d_{\text{global}}^{\max}1_N, \Theta^{\max}1_{N-1}\right)^\top$$

Any element that exceeds its limits after mapping is clipped to the nearest bound; the clipping distance is multiplied by the corresponding penalty factor $\phi$ and accumulated into the action-bound penalty $C_t^{\text{action}}$.

### A.7.5 TRANSITION DYNAMICS

At each time step $t \in \mathcal{T}$, the environment moves from state $s_t$ to $s_{t+1}$ after receiving a normalized action $\mathbf{a}_{\text{norm},t} \in [-1,1]^{G+3N+(N-1)}$. The update proceeds through seven ordered steps:

**1. Action decoding, clipping, and penalty logging.** Each normalized component is mapped back into its physical range:

$$a_{i,t}^{\text{pre}} = \frac{a_{\text{norm},i,t}+1}{2}\left(a_i^{\max} - a_i^{\min}\right) + a_i^{\min},$$

and clipped to remain within bounds $[a_i^{\min}, a_i^{\max}]$. Let the resulting action vector be

$$\mathbf{a}_t = \{p_{g,t}\}_{g \in \mathcal{G}} \;\|\; \{c_{n,t}\}_{n \in \mathcal{N}} \;\|\; \{p_{n,t}^d\}_{n \in \mathcal{N}} \;\|\; \{\ell_{n,t}\}_{n \in \mathcal{N}} \;\|\; \{\theta_{n,t}\}_{n \in \mathcal{N}\setminus\{1\}},$$

with $\theta_{1,t} \equiv 0$. Each clipping violation incurs a penalty multiplied by the corresponding $\phi$..

**2. Battery state-of-charge update.** The battery SOC at each bus $n$ evolves as:

$$E_{n,t+1} = \gamma E_{n,t} + \eta c_{n,t} - \frac{1}{\eta} p_{n,t}^d, \quad n \in \mathcal{N}.$$

**3. Load-shedding enforcement.** Any load shedding exceeding the global maximum is clipped:

$$\ell_{n,t} \leftarrow \min(\ell_{n,t}, d_{\text{global}}^{\max}),$$

penalising excess with factor $\phi_{\text{shed}}$.

**4. Power-flow calculation.** Compute power flows from voltage angles, enforcing zero flow on de-energised lines:

$$f_{\ell,t} = \begin{cases} B_{ij}(\theta_{i,t} - \theta_{j,t}), & \ell \notin \mathcal{D}_t \\ 0, & \ell \in \mathcal{D}_t \end{cases}, \quad \ell = (i,j).$$

**5. Slack generation calculation.** Slack generation $s_{n,t}$ is computed to enforce exact network balance:

$$s_{n,t} = \max\Big\{0,\ d_{n,t} - \ell_{n,t} - \sum_{g:\text{BusGeneratorLink}[g]=n} p_{g,t} + c_{n,t} - p_{n,t}^d$$

$$+ \sum_{(i,n)\in\mathcal{L}} f_{(i,n),t} - \sum_{(n,j)\in\mathcal{L}} f_{(n,j),t}\Big\}.$$

**6. Net nodal-injection and power-balance penalty.** Compute net nodal injection at each bus $n$:

$$P_{n,t} = \sum_{g:\text{BusGeneratorLink}[g]=n} p_{g,t} + s_{n,t} - d_{n,t} + \ell_{n,t} - c_{n,t} + p_{n,t}^d.$$

The nodal power-balance residual is:

$$\Delta_{n,t} = P_{n,t} - \sum_{j\in\mathcal{N}} B_{nj}(\theta_{n,t} - \theta_{j,t}),$$

and the network-balance penalty is:

$$C_t^{\text{bal}} = \phi_{\text{bal}} \sum_{n\in\mathcal{N}} |\Delta_{n,t}|.$$

**7. Demand-forecast and observation reconstruction.** Update the forecast window at each bus $n$:

$$D_{n,t:t+k-1} = (d_{n,t}, d_{n,t+1}, \ldots, d_{n,\min\{t+k-1,T\}}),$$

padded with zeros beyond horizon $T$. Form the next state $s_{t+1}$ from normalized SOC, voltage-angle differences, loading ratios, flows, slack generation $s_{n,t}$, demand window, and normalized time $\tau_{t+1} = t/(T-1)$. Components outside valid bounds are clipped, incurring penalties accordingly.

A.7.6  COST FUNCTION

The total penalty cost at time $t$ is

$$C_t = C_t^{\text{soc}} + C_t^{\theta} + C_t^{\text{flow\_ratio}} + C_t^{\text{slack}} + C_t^{\text{bal}} + C_t^{\text{action}}$$

**Observation-bound terms.** For each category $c \in \{\text{soc}, \theta, \text{flow\_ratio}, \text{slack}\}$, let $\mathcal{I}_c$ be the indices of the corresponding observation sub-vector and $[L_i, U_i]$ its valid range. Then

$$C_t^c = \sum_{i\in\mathcal{I}_c} \begin{cases} \phi_c (L_i - x_{i,t}), & x_{i,t} < L_i, \\ \phi_c (x_{i,t} - U_i), & x_{i,t} > U_i, \\ 0, & \text{otherwise.} \end{cases}$$

**Network-balance term.** $C_t^{\text{bal}}$ is defined in Transition-Step 5 as $\phi_{\text{bal}} \sum_n |\Delta_{n,t}|$, penalising any mismatch between injections and DC power flow.

**Action-bound term.** $C_t^{\text{action}}$ aggregates the clipping penalties accrued in Transition-Step 1 for generator power, charge/discharge rates, load shedding, and bus angles (the latter weighted by $\phi_{\theta,\text{act}}$). It ensures every action component exceeding its hard limit is penalised proportionally to its violation.

### A.7.7    REWARD FUNCTION

We decompose the per-step reward $\Pi_t$ into three components, each denoted by $\Pi$:

$$\Pi_t = \Pi_t^{\text{gen}} \; + \; \Pi_t^{\text{slack}} \; + \; \Pi_t^{\text{shed}}.$$

where

$$\Pi_t^{\text{gen}} = -\sum_{g \in \mathcal{G}} \sum_{j=0}^{\text{PolynomialDegree}-1} C_{g,j} \, p_{g,t}^j,$$

negative generator fuel cost (polynomial in output $p_{g,t}$),

$$\Pi_t^{\text{slack}} = -K_{\text{slack}} \sum_{n \in \mathcal{N}} s_{n,t},$$

negative cost of slack generation at each bus,

$$\Pi_t^{\text{shed}} = -K_{\text{ls}} \sum_{n \in \mathcal{N}} \ell_{n,t}.$$

negative cost of load-shedding.

### A.7.8    EPISODE TERMINATION

An episode ends when the final time step $t = T$ is reached. It is *not truncated*, even if the system encounters infeasible states.

## A.8    INTEGRATED SCHEDULING AND MAINTENANCE (SCHEDMAINTENV)

### A.8.1    OVERVIEW

Energy-intensive chemical processes leverage Demand Response (DR) to adjust electricity usage in response to price fluctuations, typically optimizing production on a rolling basis using forecasted demand and electricity prices. However, optimizing production scheduling alone can be detrimental, as it neglects the operational condition of essential equipment. Recent studies have addressed this by integrating condition-based maintenance into production optimization, notably for Air Separation Units (ASUs) (Xenos et al., 2016), and natural gas plants (Huang & Zheng, 2020).

In this study, we model an Air Separation Unit (ASU) comprising three compressors tasked with meeting aggregated gaseous nitrogen (GAN) and oxygen (GOX) demand over a 31-day episode, leveraging a deterministic 30-day rolling forecast of electricity prices and demands. Each day, the agent decides for each compressor whether to operate—producing at a chosen output level—or to undergo maintenance, incurring downtime and resetting its operational state. If total production falls short of demand, the deficit is met through external purchases at a fixed (though inflated) price. The objective is to minimize total operating expense—comprising electricity costs, downtime losses, and external purchase costs—by optimally trading off short-term production gains against long-term equipment health. The base environment, which is completely deterministic, is termed SchedMaintEnv-v0. To further resemble real-world operations, we introduce uncertainty in compressor failure times. We refer to our stochastic variant of the base environment as SchedMaintEnv-v1. A schematic of the environment is shown in figure 8.

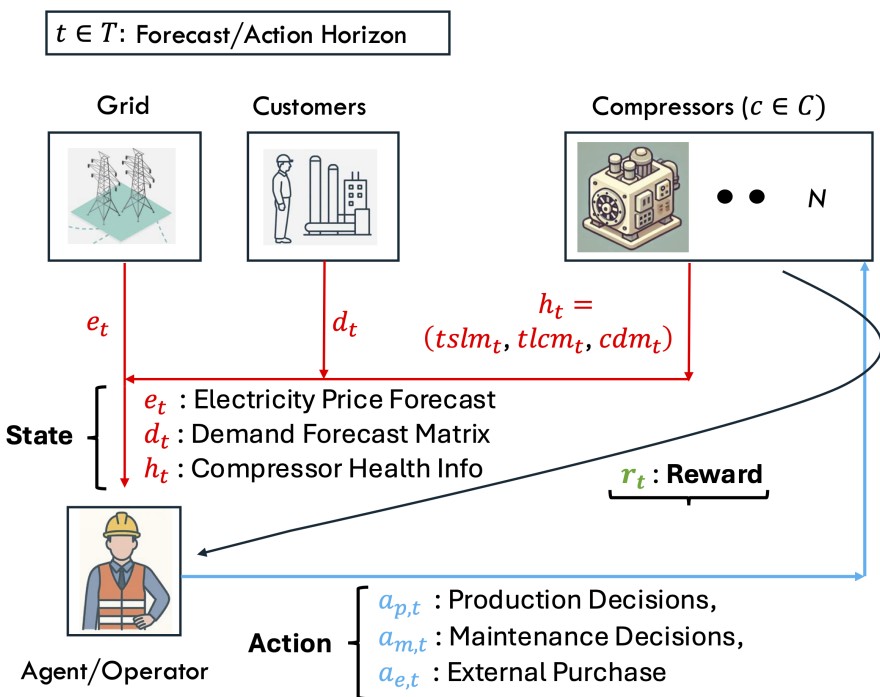

Figure 8: Schematic of `SchedMaintEnv`. At each time step $t$, the agent observes the day-ahead electricity price forecast $e_t$, the demand forecast $d_t$, and the compressor health state $h_t = (\text{tslm}_t, \text{tlcm}_t, \text{cdm}_t)$. It then selects compressor production rates $a_{p,t}$, maintenance scheduling flags $a_{m,t}$, and external purchase fraction $a_{e,t}$ to meet demand while managing cost and maintenance constraints, receiving reward $r_t$ from the compressors.

### A.8.2 PROBLEM SETUP

The sets and known parameters used to describe the environment are shown below:

**Sets**

- $\mathcal{T} = \{0, \ldots, T\}$: Set of discrete time periods.
- $\mathcal{C}$: Set of compressors.

**Parameters**

- $n$: Number of compressors.
- $S$: Forecast horizon.
- $Cap_c$: Maximum daily production capacity of the compressor $c$.
- $\text{SPEN}_c$: Specific energy of compressor $c$ in KWh/t.
- $\text{MTTF}_c$: Mean time to failure represents the maximum number of consecutive operating days before maintenance is required for compressor $c$.
- $\text{MTTR}_c$: Mean time to repair represents fixed duration (in days) of any maintenance outage compressor $c$.
- $\text{MNRD}_c$: Minimum no-repair duration, i.e., the time that must elapse after maintenance before the next service can begin for compressor $c$.
- $tlcm_{c,0}$: Initial time left to complete maintenance for compressor $c$ at start of the episode.
- $tslm_{c,0}$: Initial time since last maintenance for compressor $c$ at start of the episode.

- $cdm_{c,0}$: Initial indicator of whether compressor $c$ is eligible for maintenance at the start of the episode.

- $\alpha_{ext}$: External purchase price of per unit of product.

- $Q_{ext}$: Maximum possible purchase quantity for any given day.

- D: Array of daily forecasted demand over the simulation horizon $T + S$ in ton.

- E: Array of daily forecasted electricity prices over the simulation horizon $T + S$ in \$/KWh.

- $\rho_{\text{MD}}, \rho_{\text{MF}}, \rho_{\text{EM}}, \rho_{\text{RP}}, \rho_{\text{D}}$: Various penalty parameters related to constraint violation.

### A.8.3 STATE SPACE

The observation state at time $t \in \mathcal{T}$ is represented as a vector,

$$s(t) = (d_t, e_t, tslm_t, tlcm_t, cdm_t), \quad t \in [0, T]$$

The state vector $s(t)$ captures the essential operational and maintenance-related information for the Air Separation Unit (ASU) on day $t$. It includes forecasts of production demands and electricity prices over a fixed horizon ($S$ days), along with detailed maintenance indicators for each compressor. These components are defined as follows:

- Demand Forecast ($d_t \in \mathbb{R}_+^{S \times 1}$): a vector of predicted demands from day $(t + 1)$ to $(t + S)$, expressed in tons.

- Electricity Price Forecast ($e_t \in \mathbb{R}_+^{S \times 1}$): a vector of corresponding day-ahead electricity prices from day $t$ to $t + S - 1$, measured in \$/kWh

- Time Since Last Maintenance ($\text{tslm}_t \in \mathbb{Z}_+^{n \times 1}$): the number of days since each compressor $c$ last underwent maintenance

- Time Left to Complete Maintenance ($\text{tlcm}_t \in \mathbb{Z}_+^{n \times 1}$): the remaining time (in days) required to complete maintenance for each compressor $c$; it is strictly positive only when maintenance is in progress.

- Can Do Maintenance Indicator ($\text{cdm}_t \in \{0, 1\}^n$): a binary vector where $\text{cdm}_{ct} = 1$ indicates that compressor $c$ is eligible for maintenance on day $t$

In the base SchedMaintEnv, the agent strives to learn the fixed failure time $\text{MTTF}_c$ of each compressor $c$. In the stochastic variant, we assume each compressor can fail at $\text{MTTF}_c$, $\text{MTTF}_c - 1$, or $\text{MTTF}_c - 2$ with equal probability; hence, the agent should ideally learn a robust preventive maintenance policy. We implement the uncertainty by introducing the aforementioned stochasticity in failure times at the beginning of each episode.

### A.8.4 ACTION SPACE

Given the received observation at the start of each day $t \in \mathcal{T}$, the action space at time $t$ consists of operational decisions related to maintenance scheduling, compressor utilization, and external product procurement. The physical description is as follows:

- Compressor maintenance ($a_{\text{maintenance}}(t) \in \{0, 1\}^n$): a binary vector indicating whether each compressor is scheduled for maintenance at time $t$, with $a_{\text{maintenance},c}(t) = 1$ if compressor $c$ is under maintenance. We occasionally abbreviate this action as $a_{\text{maint.},c}(t)$.

- Compressor production rate ($a_{\text{production}}(t) \in [0, 1]^n$): a continuous vector representing the fraction of the maximum capacity $Cap_c$ utilized by each compressor $c$. We occasionally abbreviate this action as $a_{\text{prod.},c}(t)$.

- External purchase $a_{\text{purchase}}(t) \in [0, 1]$: the fraction of the maximum external product $Q_{\text{ext}}$ purchased to meet demand when internal production is insufficient.

The agent's raw actions are clipped to remain within their specified bounds. In particular, each component of $a_{\text{maintenance}}(t)$ is first generated as a scalar in $[0, 1]$ and then rounded to $\{0, 1\}$, while all other actions are clipped directly to their respective intervals.

A.8.5 TRANSITION DYNAMICS

Here we define how the environment state evolves in response to the agent's actions at each discrete time step $t \in \mathcal{T}$. Let $s(t) = (d_t, e_t, \text{tslm}_t, \text{tlcm}_t, \text{cdm}_t)$ be the observation vector at time $t$, with $s(0)$ denoting the initial observation. The transition to $s(t+1)$ is governed by the following procedures:

**Information State Update:** This update incorporates changes in demand and electricity price signals. The updated states are retrieved from the simulated perfect-forecast arrays of demand (D) and electricity prices (E) for the next $S$ days as follows:

$$\mathbf{d}_{t+1} \leftarrow \text{D}[t+1, t+S], \quad e_{t+1} \leftarrow \text{E}[t+1, t+S] \qquad \forall t \in T$$

It is worth noting that the simulated data is appropriately longer than the episode length to account for the state horizon; therefore, no padding is used at any point.

**Compressor Physical Condition Transition:** The following updates track the evolution of maintenance status and compressor readiness for each compressor $c \in C$, based on operational decisions. The initial physical state at the start of the simulation is given by $(tlcm_{c,0}, tslm_{c,0}, cdm_{c,0})$, and future states are derived accordingly.

$$\text{tslm}_{c,t+1} = \begin{cases} 0, & \text{if } a_{\text{maintenance},c}(t) = 1 \\ \text{tslm}_{ct} + 1, & \text{otherwise} \end{cases}$$

$$\text{tlcm}_{c,t+1} = \begin{cases} \text{MTTR}_c - 1, & \text{if } a_{\text{maintenance},c}(t) = 1 \wedge \text{cdm}_{ct} = 1 \\ \text{tlcm}_{ct} - 1, & \text{if } a_{\text{maintenance},c}(t) = 1 \wedge \text{cdm}_{ct} = 0 \\ \text{tlcm}_{ct}, & \text{otherwise} \end{cases}$$

$$\text{cdm}_{c,t+1} = \begin{cases} 1, & \text{if } \text{tslm}_{c,t+1} \geq \text{MNRD}_c \\ 0, & \text{otherwise} \end{cases}$$

To ensure feasibility and consistency with compressor state constraints, a sanitization step is applied to the raw agent actions before the state update, but after the associated violation costs are realized. For each compressor $c \in C$, the action is adjusted as follows:

- If $\text{tslm}_{ct} \geq \text{MTTF}_c$ and $a_{\text{maintenance},c}(t) \neq 1$, then:

  $$\Rightarrow \quad a_{\text{maintenance},c}(t) \leftarrow 1 \quad \text{and} \quad a_{\text{production},c}(t) \qquad \leftarrow 0$$

  This rule enforces maintenance when it is overdue (i.e., $\text{tslm}_{ct} \geq \text{MTTF}_c$) but the agent has not scheduled it. Maintenance is forced, and production is halted to ensure feasibility and update the environment state accordingly.

- If $a_{\text{maintenance},c}(t) = 1$ and $a_{\text{production},c}(t) > 0$, then:

  $$\Rightarrow \quad a_{\text{production},c}(t) \leftarrow 0$$

  This rule ensures that production is not allowed during maintenance.

- If $\text{cdm}_{ct} = 0$ and $\text{tlcm}_{ct} = 0$ and $a_{\text{maintenance},c}(t) = 1$, then:

  $$\Rightarrow \quad a_{\text{maintenance},c}(t) \leftarrow 0$$

  To ensure maintenance is not performed after the required duration or prematurely before it is permitted.

- If $\text{tlcm}_{ct} > 0$ and $a_{\text{maintenance},c}(t) \neq 1$, then:

  $$\Rightarrow \quad a_{\text{maintenance},c}(t) \leftarrow 1 \quad \text{and} \quad a_{\text{production},c}(t) \qquad \leftarrow 0$$

  To make sure maintenance remains active for the required maintenance duration.

### A.8.6   COST FUNCTION

The agent may incur various costs at each time step if system constraints are violated, encouraging it to learn an optimal policy. We reiterate that these costs are realized before the action is sanitized. The potential costs are as follows:

**Maintenance Duration Cost:**

This cost is incurred if maintenance is interrupted before it is completed or prolonged, in which case $\text{tlcm}_{ct}$ becomes negative. The cost is denoted by $C_{ct}^{\text{MI}}$, where MI stands for maintenance interruption, defined as:

$$
C_{ct}^{\text{MI}} = \begin{cases} \rho_{\text{MD}} \cdot \exp\left(|\text{tlcm}_{ct}|\right) & \begin{array}{l} \text{if } (a_{\text{maint.},c}(t) \neq 1 \text{ and } \text{tlcm}_{ct} > 0) \\ or \ (a_{\text{maint.},c}(t) = 1 \text{ and } \text{tlcm}_{ct} = 0 \text{ and } \text{tslm}_{ct} = 0) \\ or \ (a_{\text{maint.},c}(t) = 1 \text{ and } \text{tlcm}_{ct} < 0), \end{array} \\ 0 & \text{otherwise.} \end{cases}
$$

**Maintenance Failure Cost:**

This cost occurs if the *Time Since Last Maintenance* ($\text{tslm}_{ct}$) exceeds the *Mean Time to Failure* ($\text{MTTF}_c$) of the compressor. The cost is denoted by $C_{ct}^{\text{MF}}$, where MF stands for maintenance failure, defined as:

$$
C_{ct}^{\text{MF}} = \begin{cases} \rho_{\text{MF}} \cdot (\text{tslm}_{ct} - \text{MTTF}_c) & \text{if } (a_{\text{maint.},c}(t) = 0 \text{ and } \text{tslm}_{ct} > \text{MTTF}_c), \\ \rho_{\text{MF}} & \text{if } (a_{\text{maint.},c}(t) = 0 \text{ and } \text{tslm}_{ct} = \text{MTTF}_c), \\ 0 & \text{otherwise.} \end{cases}
$$

**Early Maintenance Cost:**

This cost is incurred if maintenance is performed on a compressor when it is not yet eligible for maintenance (i.e., when $\text{cdm}_{ct} = 0$, indicating that the compressor has recently undergone maintenance, and $\text{tlcm}_{ct} = 0$). The cost is proportional to the *Time Since Last Maintenance* ($\text{tslm}_{ct}$) of the compressor. The cost is denoted by $C_{ct}^{\text{EM}}$, where EM stands for early maintenance, defined as:

$$
C_{ct}^{\text{EM}} = \begin{cases} -\rho_{\text{EM}} \cdot \text{tslm}_{ct} & \text{if } (a_{\text{maint.},c}(t) = 1 \text{ and } \text{cdm}_{ct} = 0 \text{ and } \text{tlcm}_{ct} = 0), \\ 0 & \text{otherwise.} \end{cases}
$$

**Ramp Cost:**

This cost is incurred when a compressor is ramped up while under maintenance.

$$
C_{ct}^{\text{Ramp}} = \begin{cases} \rho_{\text{RP}} \cdot a_{\text{prod.},c}(t) \cdot \text{Cap}_c & \text{if } (a_{\text{maint.},c}(t) = 1 \text{ and } a_{\text{prod.},c}(t) \neq 0), \\ 0 & \text{otherwise.} \end{cases}
$$

**Demand Cost:**

This penalty is incurred if the total supply from production and external purchases does not meet the demand on the current day. The total supply is the sum of the production and external purchase, and if this is less than or greater than the demand, a penalty is imposed proportional to the absolute difference between demand and supply.

$$
C_t^{\text{Demand}} = \rho_{\text{D}} \cdot |d_t - \sum_c (a_{\text{production},c} \cdot Cap_c)|
$$

**Total Cost:**

$$
C_t^{\text{total}} = \sum_{c \in \mathcal{C}} (C_{ct}^{\text{MI}} + C_{ct}^{\text{MF}} + C_{ct}^{\text{EM}} + C_{ct}^{\text{Ramp}}) + C_t^{\text{Demand}}
$$

### A.8.7 REWARD FUNCTION

The reward function represents the cost incurred by the agent for making decisions related to production and external purchases. At each time step $t$, the reward is defined as:

$$\Pi_t^{action} = -\big(\text{production cost}_t + \text{external purchase cost}_t\big)$$

where,

$$\text{production cost}_t = \sum_{c \in \mathcal{C}} \big(\text{SPEN}_c \, a_{\text{production},c}(t) \, \text{Cap}_c \, \mathbb{E}[t]\big)$$

$$\text{external purchase cost}_t = a_{\text{purchase}}(t) \, Q_{\text{ext}} \, \alpha_{\text{ext}}$$

The production cost at time $t$ is calculated using the production rate, compressor capacity, specific energy consumption, and electricity price. The external purchase cost is incurred when demand exceeds production capacity, calculated by multiplying the purchase amount by the external price.

### A.8.8 EPISODE TERMINATION

The episode terminates when $t + 1 = T$, indicating the start of the day immediately after the final one.

## A.9 PRODUCTION SCHEDULING IN AIR SEPARATION UNIT (ASUENV)

### A.9.1 OVERVIEW

The ASUEnv simulates a liquid air separation unit (ASU) producing liquid nitrogen, oxygen, and argon in a Gym-compatible reinforcement learning framework. We consider the ASU to have hourly production capacity while demand occurs only at 24-hour intervals. To that end, at each hour the agent selects production rates based on inventory levels to minimize electricity and storage costs while ensuring daily demand satisfaction. Production actions are restricted to the convex hull of historically observed operating points, guaranteeing industrial feasibility. We follow the simplified dynamics of Zhang et al. (Zhang et al., 2015), which capture core ASU behavior without the complexity of detailed mode-switching or maintenance constraints.

We make several key assumptions to keep the environment tractable. Only two production modes—"Work" (active production) and "Off" (zero output)—are available each hour, and switching between them is instantaneous and cost-free. Electricity prices and product demands over a short rolling horizon are treated as perfectly known and error-free. Storage capacity for each product is finite and nonnegative, with penalties applied if inventory limits are exceeded or daily demand is unmet. Finally, we do not model any external purchasing option, assuming all demand falls within the ASU's inherent production capability. The simulation runs for a total of seven days with a four-day lookahead, resulting in $24 \times 7$ steps per episode. A schematic of the environment is shown in figure 9.

Figure 9: Schematic of ASUEnv. At each hour $t$, the agent observes the electricity price forecast $e_t$, demand forecast $D_t$, and inventory levels $IV_t$. It then selects production weights $\lambda_t$ (convex-combination coefficients of historical patterns) to meet demand and manage inventories, and receives reward $r_t$ from the ASU.

### A.9.2 PROBLEM SETUP

The sets and known parameters used to describe the environment are shown below:

**Sets**

- $\mathcal{T} = \{0, \ldots, T\}$: Set of hours in the episode.
- $\mathcal{D} = \{1, \ldots, D\}$: Set of days in the episode.
- $\mathcal{J}$: Set of liquid products: liquid nitrogen (LIN), liquid oxygen (LOX) and liquid argon (LAR).
- $\mathcal{X}$: Set of vertices of the convex hull derived from historical operational data.

**Parameters**

- $T$: Episode length in hours.
- $D$: Episode length in days.
- $S$: Lookahead days used in the forecast.
- $m$: Number of products in $\mathcal{J}$.
- $[IV^{j,\text{lb}}, IV^{j,\text{ub}}]$: Lower and upper bounds on the inventory level of each product $j$.
- $N$: Number of historical production data points.
- $v_x$: Extreme points of the convex hull.
- HPQ: Historical hourly production quantities.

- $k$: Number of extreme points of the convex hull.
- $\bar{D}_t$: Matrix of hourly product demands used in the simulation, of dimension $\mathbb{R}_+^{m \times 4(S+T+1)}$.
- $\bar{E}_t$: Array of hourly electricity prices used in the simulation, of dimension $\mathbb{R}_+^{24(S+T+1)}$.
- $PQ_t$: Array of products produced based on the actions.
- $DQ_d$: Array of dispatched product quantities at the end of each day in the simulation.
- $\rho_{\mathrm{IV}}$: Penalty parameter for inventory overflow.
- $\rho_{\mathrm{D}}$: Penalty parameter for unmet demand.
- $C_{\mathrm{fixed}}$: Fixed cost per hour to keep the plant operational.
- $C_{\mathrm{unit}}$: Hourly unit production cost.

### A.9.3   STATE SPACE

At any hour $t$, the observation state is represented by the vector

$$s(t) = (e_t, D_t, IV_t), \quad \forall t \in \mathcal{T}$$

The state vector $s(t)$ captures all relevant information needed for the agent to make production planning decisions at hour $t$ in the Air Separation Unit (ASU). It includes deterministic forecasts of electricity prices and product demands over a lookahead horizon of $S$ future days; therefore, including the current day, the total forecasting horizon becomes $S + 1$ days. It also includes the current inventory levels of liquid products. The components are defined as follows:

- Electricity Price Forecast ($e_t \in \mathbb{R}_+^{24(S+1) \times 1}$): a vector of day-ahead electricity prices for the next $S + 1$ days, given at an hourly resolution (totaling $24(S + 1)$ elements), measured in KWh.
- Demand Forecast ($D_t \in \mathbb{R}_+^{24(S+1) \times m}$): a matrix of demands for each product $j$ over the next $S + 1$ days, where $j \in \mathcal{J}$. Each row represents the hourly demand forecast for one product; however, the demand is non-zero only at 24-hour intervals.
- Inventory Levels ($IV_t \in \mathbb{R}_+^{m \times 1}$): the current inventory levels of all $m$ liquid products at hour $t$. These are bounded between predefined lower and upper capacity limits.

### A.9.4   ACTION SPACE

The action space at each time step $t$ is defined as:

$$a(t) = \lambda(t),$$

where $\lambda(t) \in [0, 1]^k$ represents the weights of the extreme points in the convex hull of the possible production quantities. We assume historical hourly production quantities are given by

$$\mathrm{HPQ} = \left\{ Q_j^1, Q_j^2, \ldots, Q_j^N \mid j \in \mathcal{J} \right\},$$

where $N$ is the number of samples available. The feasible region FR is then approximated as `convhull(HPQ)`, with vertices $v_{jx}$ for $j \in \mathcal{J}$, $x \in \mathcal{X}$, and $|\mathcal{X}| = k$. The quantity of each product produced at time $t$ is

$$PQ_{jt} = \sum_{x \in \mathcal{X}} \lambda_x(t) \, v_{jx}.$$

The received actions are clipped to their bounds $[0, 1]$ and then normalized to enforce the convex-sum property $\sum_{x=1}^{k} \lambda_x(t) = 1$ as described in the next section.

### A.9.5   TRANSITION DYNAMICS

Here we describe how the system state evolves in response to the agent's production decisions at each discrete hourly time step $t \in T$. Let the observation vector at time $t$ be $s(t) = (e_t, D_t, IV_t)$, with $s(0)$ denoting the initial observation. The transition to $s(t + 1)$ is governed by two primary updates: the shifting of forecast windows and the physical evolution of product inventories.

**Forecast Update:** The forecast arrays for electricity prices and product demand are deterministic and span a rolling horizon of $S + 1$ days at hourly resolution. The environment updates the forecast component of the state differently depending on whether a new hour or a new day has begun.

**Hourly Shifting:** At each non-zero hour $(t + 1) \bmod 24 \neq 0$, the environment updates the observation by shifting the forecast vectors leftward by one hour to discard outdated information. This is achieved via the `shift_observation()` routine:

$$D_{t+1} \leftarrow \texttt{ShiftLeft}(D_t), \quad e_{t+1} \leftarrow \texttt{ShiftLeft}(e_t)$$

where `ShiftLeft` removes the earliest hour from the forecast vector and appends a placeholder value—such as the average of the remaining values (for electricity) or zeros (for demand)—to preserve the total horizon length $24(S + 1)$.

**Daily Refresh:** At the start of each new day (i.e., when $(t + 1) \bmod 24 = 0$), the environment invokes `update_demand_and_electricty_state()` to refresh the entire forecast arrays for electricity prices and product demands. This function populates only the end-of-day demand values (i.e., the 24th hour of each day) while keeping the rest of the hourly entries zero, as demand is modeled to be daily:

$$D_{t+1} \leftarrow \left( \bar{\mathrm{D}}[t + 1], \bar{\mathrm{D}}[t + 2], \ldots, \bar{\mathrm{D}}[t + 24\,(S + 1)] \right)$$

$$E_{t+1} \leftarrow \left( \bar{\mathrm{E}}[t + 1], \bar{\mathrm{E}}[t + 2], \ldots, \bar{\mathrm{E}}[t + 24\,(S + 1)] \right)$$

This rolling update mechanism enables the agent to make production decisions with awareness of upcoming price and demand trends while ensuring that outdated data does not persist in the observation state. The simulated data exceeds the episode length to cover the lookahead horizon, eliminating the need for padding during daily refresh.

**Inventory State Transition:** Before updating the inventory, action sanitization rescales the raw weights $\{\lambda_x(t)\}$ to enforce the convex-sum constraint:

$$\lambda_x(t) \leftarrow \frac{\lambda_x(t)}{\sum_{y=1}^{k} \lambda_y(t)} \quad \forall x \in \mathcal{X}, \quad \text{s.t.} \quad \sum_{x=1}^{k} \lambda_x(t) = 1$$

The inventory vector is then updated based on the production action $PQ_t$ at time $t$, itself computed as a convex combination of feasible production profiles. If $t \bmod 24 = 23$ (the last hour of the day), a portion of inventory is shipped to meet daily demand; otherwise, production is simply added:

$$IV_{j,t+1} = \begin{cases} IV_{j,t} + PQ_{j,t} - DQ_{j,d}, & \text{if } t \bmod 24 = 23, d = \left\lfloor \frac{t+1}{24} \right\rfloor \\ IV_{j,t} + PQ_{j,t} & \text{otherwise,} \end{cases}$$

where $DQ_{j,d} = \min(IV_{j,t} + PQ_{j,t}, \ D[j, 23])$ ensures that the shipment quantity does not exceed the sum of available inventory and production.

Finally, state sanitization corrects any inventory overflow by checking whether the inventory penalty $\mathrm{C}_t^{\mathrm{IV}} > 0$. If so, the inventory vector is clipped element-wise to its upper bounds:

$$IV_{t+1} \leftarrow \min\left( IV_{t+1}, \ IV^{\mathrm{ub}} \right),$$

thereby preserving feasibility by preventing storage violations.

### A.9.6 Cost Function

The agent may incur several types of costs at each time step $t$ from production decisions or constraint violations. These costs guide the agent toward learning an efficient and feasible production policy. The individual cost components are as follows:

**Inventory Overflow Cost:**
This cost is incurred if the inventory of any product exceeds its maximum storage capacity:

$$\mathrm{C}_t^{\mathrm{IV}} = \rho_{\mathrm{IV}} \cdot \sum_{j \in \mathcal{J}} \max(IV_{j,t} - IV_j^{\max}, 0)$$

**Demand Shortfall Cost:**

At the end of each day (i.e., every 24 hours), the environment evaluates whether the shipped quantity meets the daily demand. A cost is imposed for any shortfall:

$$\mathrm{C}_t^D = \begin{cases} \rho_\mathrm{D} \cdot \sum_{j \in J} \max\left(D[j,t] - \mathrm{DQ}_{j,d}, 0\right) & \text{if } t \bmod 24 = 23, d = \left\lfloor \frac{t+1}{24} \right\rfloor \\ 0 & \text{otherwise,} \end{cases}$$

where $D[j, 23]$ is the daily non-zero demand for product $j$, and $DQ_{j,d}$ is the quantity shipped.

**Total Cost**

$$\mathrm{C}_t^{\text{total}} = \mathrm{C}_t^{\text{IV}} + \mathrm{C}_t^D$$

### A.9.7  REWARD FUNCTION

The reward at time $t$ is defined as the negative of the production cost:

$$\Pi_t^{production} = -\left(\text{production cost}_t\right)$$

The production cost is formally given by:

$$\text{production cost}_t = \begin{cases} 0 & \text{if } \sum_x \lambda_x(t) = 0, \\ C_{\text{fixed}} + \left(\sum_{j \in J} PQ_{jt}\right) \cdot C_{\text{unit}} \cdot e[t] & \text{otherwise} \end{cases}$$

where $PQ_{jt}$ is the production quantity of product $j$, and $e[t]$ is the electricity price at time $t$. These costs arise from the fixed cost $C_{\text{fixed}}$, incurred if any production occurs and a variable cost, proportional to the total production and electricity price at hour $t$.

### A.9.8  EPISODE TERMINATION

The episode terminates when $(t + 1) = H$ or $(t + 1) = 24D$, i.e., when the next starting hour is the first hour following the final day.

## B  OTHER RESULTS

### B.1  ADDITONAL ENVIRONMENTS AND VARIANTS

We consider results for three additional environments—ASUEnv, UNEnv-v1, and GTEP. Figure 10 presents the average reward and cost per training epoch, with shaded regions indicating one standard deviation around the mean. Table 3 summarizes the evaluation results, averaged over 10 episodes, for the additional environments. For the set of environments considered, the values in green correspond to the evaluation reward and cost of the best-performing algorithms, while values in red indicate those of the worst-performing algorithms. The optimal reward is calculated by solving an optimization counterpart of the environment at each step. The strategy used to determine the algorithm with the best and worst performance, as well as the criteria to segregate the environments based on performance, remains the same as described in the main paper.

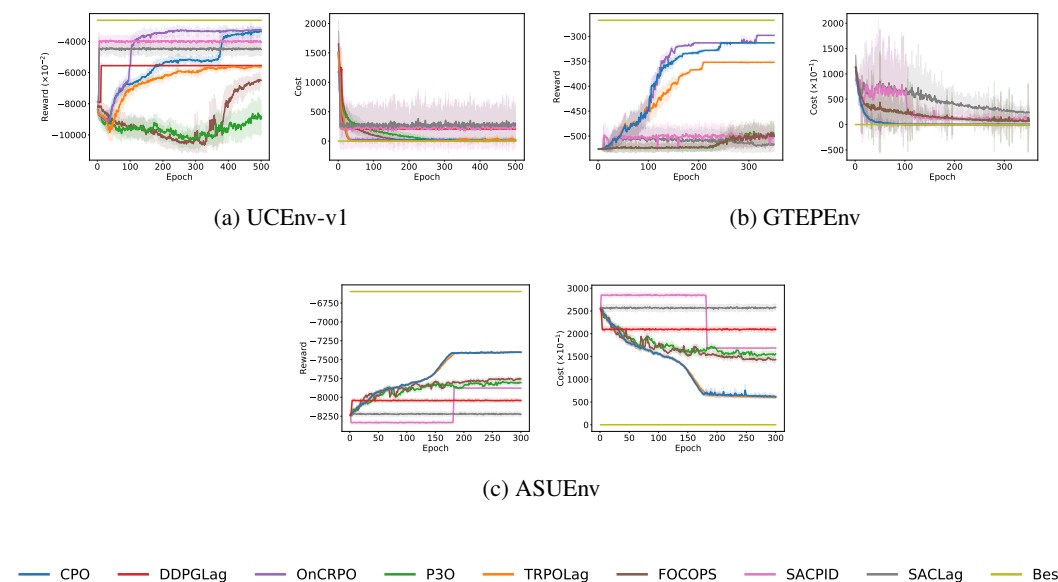

(a) UCEnv-v1      (b) GTEPEnv

(c) ASUEnv

CPO    DDPGLag    OnCRPO    P3O    TRPOLag    FOCOPS    SACPID    SACLag    Best

Figure 10: Training curves of average reward and cost per episode across three additional environments and BlendingEnv variants with different strategies

### B.1.1 DESCRIPTION OF ENVIRONMENTS

- **GTEPEnv:** This case study involves a five-region power system with two generators and possible transmission lines between all region pairs over a 10-period planning horizon. We train with 100 episodes per epoch. P3O shows a significant gap between training and evaluation in both reward and cost. Notably, DDPGLag learns a policy that achieves high rewards at the expense of significantly increased costs. To preserve clarity in visual comparisons, we omit the training curve for DDPGLag in Figure 10b. We see a significant gap for training and evaluation costs in the P3O algorithm and for rewards in the P3O, FOCOPS, SACPID, and SACLag algorithms.

- **UCEnv-v1:** This unit commitment version models a multi-bus system with explicit power flows, enabling realistic evaluation of policies where generation and demand locations are critical. This example considers a multi-bus unit commitment power system with five generators distributed across four buses, connected by five transmission lines. It operates over a 24-hour horizon with hourly updates to demand forecasts, generator states, and network power flows. A significant gap is observed for training and evaluation costs in the SACPID and SACLag algorithms.

- **ASUEnv:** This example considers production scheduling in an air separation unit over a one-week horizon, where the agent takes hourly steps to meet daily product demands. We consider a planning horizon of 5 days, with each episode consisting of 168 steps, reflecting the operation of the ASU over a week. We train using 60 episodes per epoch. A significant gap is observed between the training and testing costs for CPO and TRPOLag.We see a significant gap for training and evaluation costs in the CPO, DDPGLag,OnCRPO, and TRPOLag algorithms.

### B.1.2 DISCUSSION

For GTEPEnv and UCEnv-v1, we see that OnCRPO performs the best. However, for the ASUEnv, we observe that CPO marginally outperforms ONCRPO, positioning it as the best algorithm in this setting. DDPGLag sustains its poor performance for the additional environments as well. Furthermore, while GTEPEnv and UCEnv-v1 are trained to reasonable optimality, the evaluation costs for ASUEnv remain high for all the algorithms, indicating the issues faced by current algorithms to get good feasible solutions. This further underscores the persistent challenges inherent in such constrained

settings and highlights the urgent need for more refined and robust approaches to safe reinforcement learning in these domains.

Table 3: Evaluation results for 10 episodes

| Environment | Optimal Reward | CPO Reward | Cost | DDPGLag Reward | Cost |
|---|---|---|---|---|---|
| GTEPEnv | -267.7 | -313 | 0 | -19 | 689140 |
| UCEnv-v1 | -264104 | -319469 | 0 | -575239 | 216 |
| ASUEnv | -6597.5 | -7295.45 | 25563.34 | -7743.56 | 13130.41 |

| Environment | OnCRPO Reward | Cost | P3O Reward | Cost |
|---|---|---|---|---|
| GTEPEnv | -298 | 0 | -346 | 79.28 |
| UCEnv-v1 | -313897 | 0 | -1035242 | 0 |
| ASUEnv | -7330.32 | 17706.08 | -7427.55 | 12624.76 |

| Environment | TRPOLag Reward | Cost | FOCOPS Reward | Cost |
|---|---|---|---|---|
| GTEPEnv | -352 | 0 | -367 | 0 |
| UCEnv-V1 | -553465 | 4.21 | -589754 | 0 |
| ASUEnv | -7337.64 | 16018.8 | -7416.93 | 11747.68 |

| | SACPID | | SACLag | |
|---|---|---|---|---|
| GTEPEnv | -403 | 10725.05 | -475 | 600 |
| UCEnv:V1 | -390289 | 153.06 | -414884 | 195.9 |
| ASUEnv | -7870.69 | 16713.63 | -8329.49 | 28864.44 |

## B.2 CONSTRAINT-WISE VIOLATION ANALYSIS ACROSS SAFE-RL ALGORITHMS

In the following subsection we interpret the plots that report the *mean episode-level constraint breaches per epoch* observed during training with eight safe-rl algorithms. To help the reader connect each trend to its control philosophy, we first give a concise description of every algorithm and its reward–cost balancing mechanism.

- **Constrained Policy Optimization (CPO):** At each update, CPO solves a small trust-region *quadratic program* (QP) that maximizes expected return while ensuring the new policy remains within a cumulative-cost constraint set. This guarantees monotonic improvement in reward without violating the safety constraints.

- **Trust-Region Policy Optimization with a Lagrange Multiplier (TRPOLag):** TRPOLag enhances standard *Trust-Region Policy Optimization* (TRPO) by introducing an on-policy Lagrange multiplier, updated after each batch. This multiplier penalizes excessive costs within the Kullback–Leibler trust region, balancing reward optimization and safety.

- **On-Policy Constrained Reinforcement Policy Optimization (OnCRPO):** OnCRPO alternates between maximizing reward using a standard *Proximal Policy Optimization* (PPO) objective when constraints are satisfied, and minimizing costs through a dedicated surrogate objective when constraints are breached, thus explicitly balancing reward and safety.

- **Penalty-based Proximal Policy Optimization (P3O):** P3O integrates an *adaptive exterior penalty* into PPO's clipped-surrogate objective, dynamically adjusting penalty strength based on constraint violations. The penalty increases when cumulative cost exceeds its budget and decreases otherwise, gradually guiding the policy towards feasibility while prioritizing reward.

- **Deep Deterministic Policy Gradient with a Lagrange Multiplier (DDPGLag):** DDPGLag employs an off-policy deterministic actor–critic architecture, augmented with stability enhancements from *Twin Delayed Deep Deterministic Policy Gradients* (TD3). It concurrently learns a Lagrange multiplier, policy, and critic from replay-buffer data, effectively balancing reward and constraint satisfaction through deterministic policy gradients.

- **Soft Actor–Critic with Lagrangian Penalty (SACLag):** An off-policy actor–critic that augments the Soft Actor–Critic (SAC) objective with a learned Lagrange multiplier, adapting the penalty on expected cost online.

- **Soft Actor–Critic with PID Control (SACPID):** Extends SAC with a proportional–integral–derivative controller on cumulative cost, automatically tuning penalty strength via PID updates to balance reward and safety.

- **First-Order Constrained Policy Optimization (FOCOPS):** A trust-region method that linearizes both reward and cost objectives and solves a first-order approximation via a closed-form update, yielding a policy that enforces cost constraints with minimal computational overhead.

### B.2.1 RTNEnv

Figure 11 illustrates mean episode-level constraint violations per epoch for the RTNEnv. For *inventory-level* constraints, projection-based algorithms—**CPO**, **OnCRPO**, and **TRPOLag**—consistently reduce violations, converging swiftly to minimal violation levels. **P3O** achieves comparable compliance more gradually, while **DDPGLag** consistently exhibits the highest residual violations. Both **SACLag** and **SACPID** demonstrate strong reductions in inventory violations, reaching compliance at rates comparable to projection-based methods, whereas **FOCOPS** shows slower improvement and stabilizes with moderately higher residual violations. *Equipment-usage* violations are inherently less frequent owing to the simpler nature of the constraints compared to inventory management; all methods maintain near-baseline levels throughout training, with CPO, OnCRPO, TRPOLag, and the SAC-based methods (SACLag, SACPID) achieving compliance earliest, followed slightly later by P3O, and with DDPGLag and FOCOPS maintaining modestly higher yet infrequent violation rates.

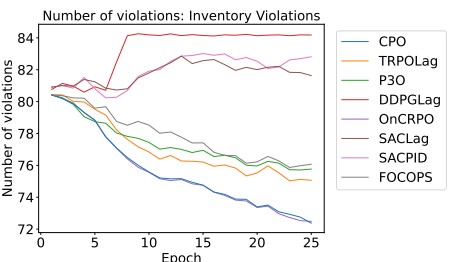 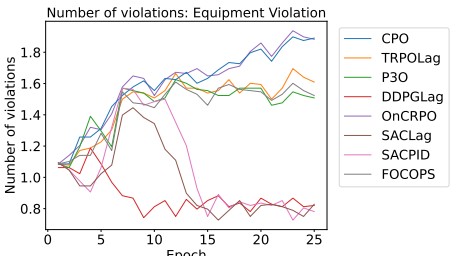

Figure 11: Average number of episode violations for different epochs for RTNEnv

### B.2.2 STNEnv

Figure 12 presents mean episode-level constraint breaches per epoch for the STNEnv. Regarding *inventory-level* safety, the projection-driven algorithms—**CPO**, **OnCRPO**, and **TRPOLag**—steadily minimize violations, stabilizing at the lowest counts observed. **P3O** demonstrates a slower convergence, while **DDPGLag** retains a significantly higher residual violation count. Both **SACLag** and **SACPID** show strong reduction in inventory violations, with convergence patterns comparable to projection-based methods, whereas **FOCOPS** achieves improvement but stabilizes at slightly higher violation levels. *Equipment-usage* breaches are consistently low for all algorithms due to

reasons similar to the RTN environment, with immediate practical compliance from CPO, OnCRPO, TRPOLag, and the SAC-based methods (SACLag, SACPID), followed shortly thereafter by P3O, while DDPGLag and FOCOPS maintain modestly higher yet infrequent violation rates.

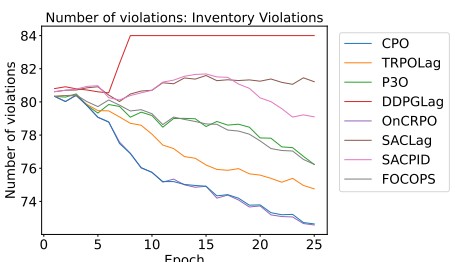 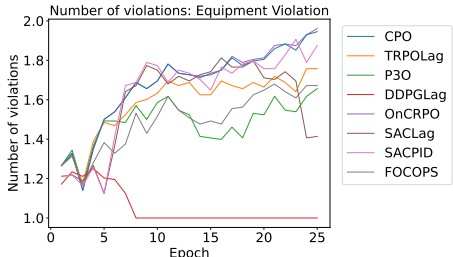

Figure 12: Average number of episode violations for different epochs for STNEnv

### B.2.3 UCEnv

**Single-bus system without network constraints (UCEnv-v0):** Figure 13 reports mean episode-level constraint violations per epoch for the single-bus UCEnv problem, covering *minimum up-time*, *minimum down-time*, and *ramping-rate* constraints. Projection-based algorithms—**CPO**, **OnCRPO**, **TRPOLag**, and **FOCOPS**—rapidly reduce violations (in about 100 epochs), converging to the lowest residual counts. **P3O** achieves compliance more gradually as its adaptive penalty strengthens, while **DDPGLag** is the quickest to reduce violations (in less than 10 epochs) early on due to aggressive Lagrangian penalty updates and off-policy learning but plateaus with ramp-up residual violations due to higher variance and instability. In contrast, SAC-based methods, **SACLag** and **SACPID**, exhibit considerably higher residual violation for the ramp-up/ramp-down constraints than other algorithms.

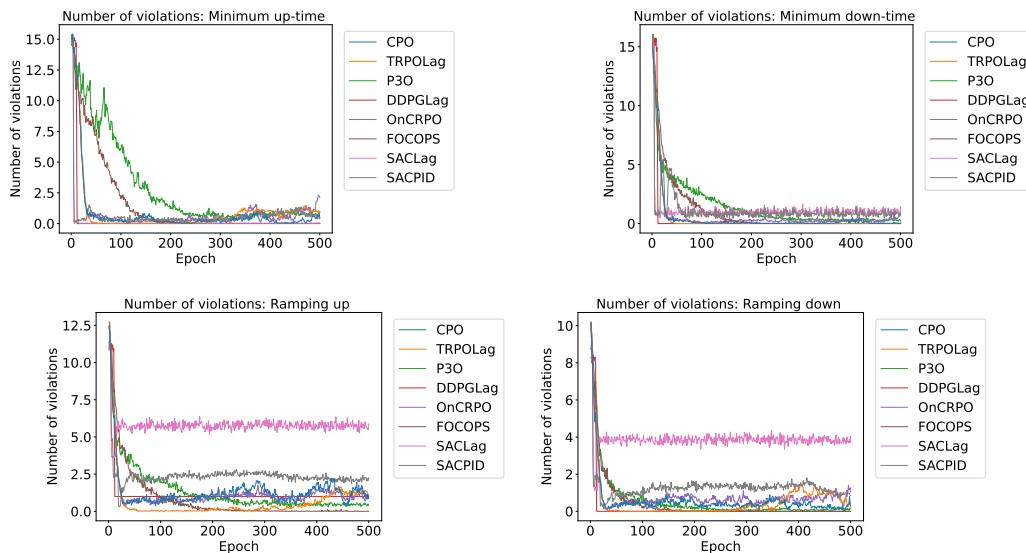

Figure 13: Average number of episode violations for different epochs for UCEnv-v0

**Multiple-bus system with network constraints (UCEnv-v1):** Figure 14 illustrates mean episode-level breaches per epoch for the multi-bus UCEnv problem, incorporating *minimum up-time*, *minimum down-time*, *ramping-rate*, and network feasibility constraints. The qualitative performance of the algorithms remains the same as described for UCEnv-v0.

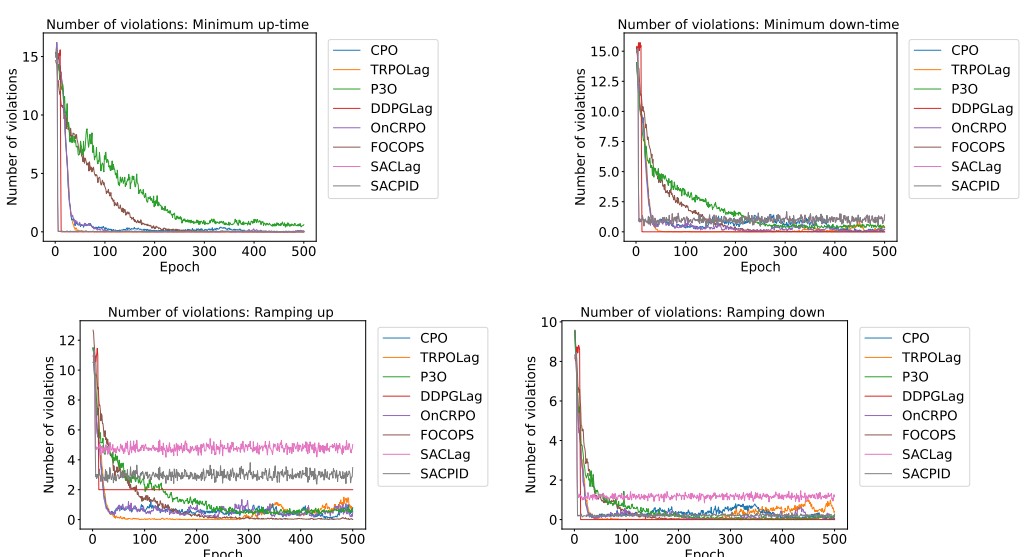

Figure 14: Average number of episode violations for different epochs for UCEnv-v1

### B.2.4 GTEPENV

Figure 15 illustrates mean episode-level constraint violations per epoch for the generation and transmission expansion planning task. The projection-based methods, **CPO**, **OnCRPO**, and **TRPOLag**, rapidly reduce generator-bound violations while keeping demand violations low, highlighting their effectiveness in per-update safety enforcement. **P3O** and **FOCOPS** achieve comparable compliance more gradually, converging to a non-zero level of bound violations. In contrast, **SACPID** and **SACLag** exhibit an initial reduction in generator-bound violations but persistently exceed the bounds thereafter, though they maintain relatively few demand violations. Finally, **DDPGLag** sustains non-zero bound violations and shows a persistent demand violation.

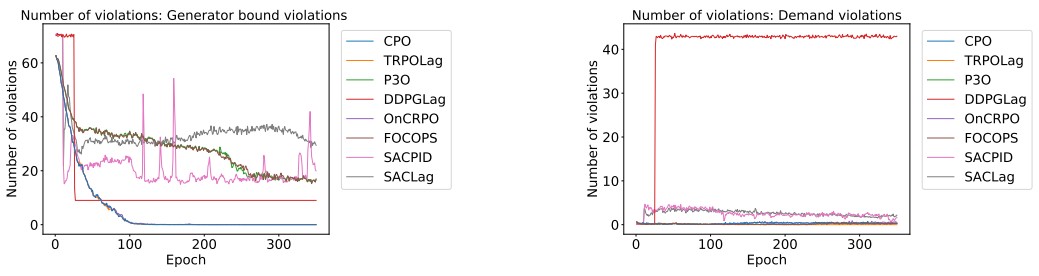

Figure 15: Average number of episode violations for different epochs for GTEPEnv

### B.2.5 BLENDINGENV

Figure 16 shows the mean episode-level constraint violations per epoch during training for the Blending environment. For inventory bound violations, **CPO**, **TRPOLag**, and **OnCRPO** rapidly converge to zero violations. **P3O** and **FOCOPS** exhibits a gradual decline with minor oscillations before stabilizing at a small but non-zero level. **DDPGLag**, **SACPID** and **SACLag** consistently oscillates without clear convergence. In contrast, for the in-out rule violations, **CPO**, **TRPOLag**, and **OnCRPO** show significant increases, stabilizing at substantial violation levels. **P3O** and **FOCOPS** displays gradual increases with **P3O** showing pronounced occilations. **DDPGLag** maintains persistent oscillations around a steady level. **SACPID** and **SACLag** maintain a steady level with very occasional osccilations. Property violations slightly increase over time for **P3O**, **CPO**, **TRPOLag**, **OnCRPO**

and **FOCOPS** with minor fluctuations, whereas **DDPGLag** demonstrates clear oscillations with an upward trend. In contrast, **SACPID** and **SACLag** oscilates around 0.

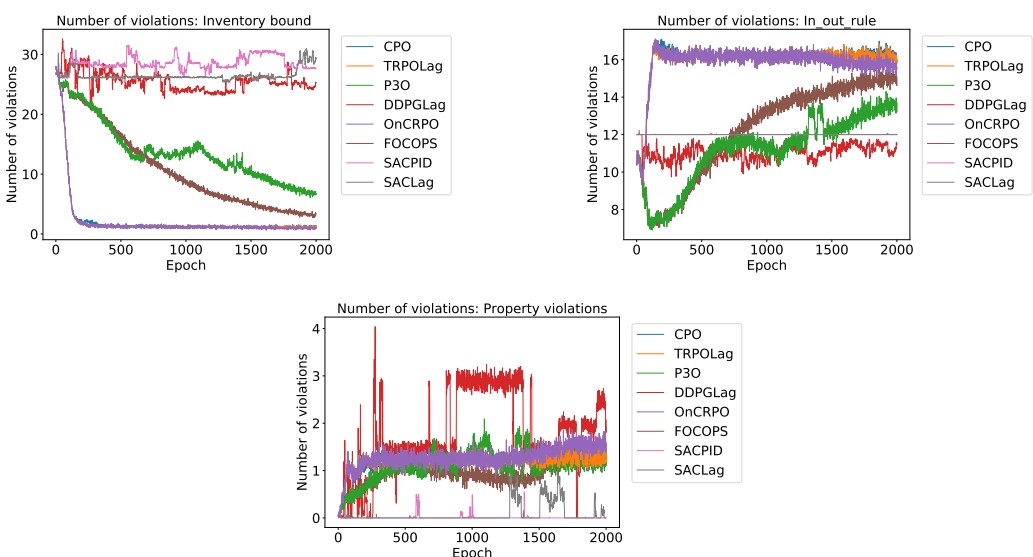

Figure 16: Average number of episode violations for different epochs for BlendingEnv with prop strategy

### B.2.6    INVMGMTENV

Figure 17 plots episode-level reordering-quantity bound violations per epoch in InvMgmtEnv across several methods. InvMgmtEnv includes bounds constraints on reordering quantities, on-hand inventory, pipeline inventory, backlog, and sales. The on-policy convex optimization method CPO and primal method OnCRPO, along with the on-policy primal-dual method TRPOLag, reduce violations rapidly and sustain low levels; the on-policy penalty function method P3O follows a similar path but needs a few extra epochs to recover from early spikes. The off-policy primal-dual methods DDPGLag and SACLag, the off-policy PID-based methods SACPID, and the on-policy convex optimization method FOCOPS also trend downward over training and exhibit zero to near-zero violations. All methods achieve near-zero violations for the on-hand inventory, pipeline inventory, sales, and backlog bounds constraints.

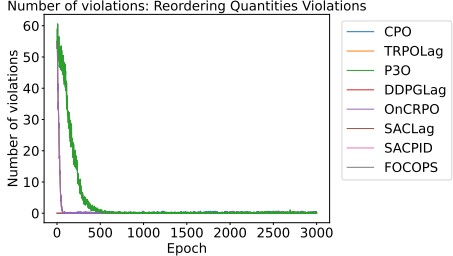

Figure 17: Average number of episode violations for different epochs for InvMgmtEnv

### B.2.7    GRIDSTORAGEENV

Figure 18 illustrates the mean episode-level violations per epoch for the GridStorageEnv, which includes bounds constraints on *generator power limits*, *battery charge rates*, *battery discharge rates*, *load shedding*, *bus-angle bounds*, *battery state of charge (SOC)*, and the *slack-bus angle*. The on-policy convex optimization method CPO, primal method OnCRPO, and primal-dual method

TRPOLag rapidly mitigate violations and quickly stabilize compliance; the on-policy penalty function method P3O converges more slowly, with temporary mid-training peaks before aligning with the leading methods. Among the other methods, apart from the on-policy convex optimization method FOCOPS, which in the initial stages shows an increasing trend in the number of violations, the off-policy primal-dual methods DDPGLag and SACLag, along with the off-policy PID-based method SACPID, remain consistently at (or near) zero across all violations. Violations for battery SOC and slack-bus voltage angles remain consistently at zero across all methods, indicating complete compliance from the outset.

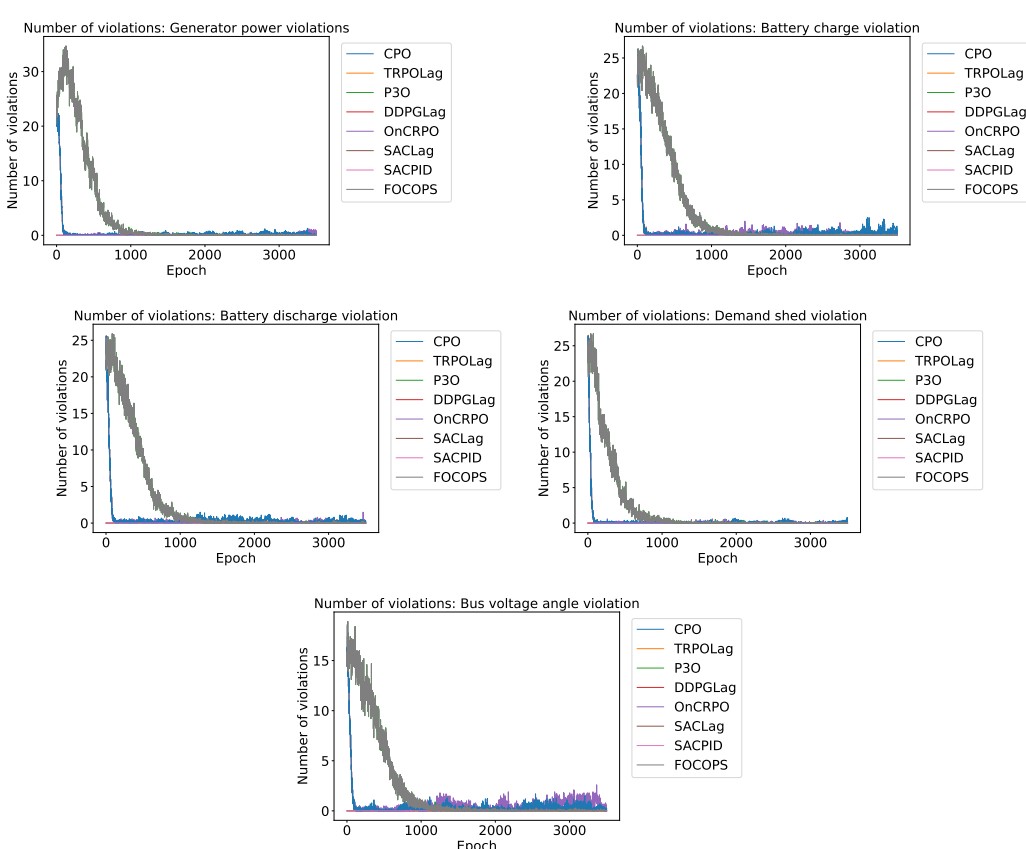

Figure 18: Average number of episode violations for different epochs for GridStorageEnv

### B.2.8 SCHEDMAINTENV

**Deterministic integrated scheduling and maintainence environemnt (SchedMaintEnv-v0):** Figure 19 illustrates mean episode-level violations per epoch for the Integrated Scheduling & Maintenance benchmark, covering constraints on *maintenance-duration*, *maintenance-failure*, *early-maintenance*, *ramping-in-maintenance*, and *demand-unsatisfaction*. Projection-based methods—**CPO**, **OnCRPO**, and **TRPOLag**—swiftly reduce violations across all constraints, establishing stable and minimal breach levels. **P3O** follows a similar trend but exhibits transient spikes in violations for almost all constraints during mid-training, except for demand-unsatisfaction, where it consistently fails to learn full compliance. For the other constraints, it eventually achieves violation levels comparable to the projection-based algorithms. **DDPGLag** consistently displays higher residual violations, notably in duration and demand-unsatisfaction categories, highlighting variability from its replay-buffer updates. Maintenance-failure violations remain consistently low for all methods. Among the remaining methods, **FOCOPS** performs on par with **P3O**. **SACLag** is only slightly better than **DDPGLag**, but it persistently yields the highest rates of maintenance failures and unmet demand. **SACPID** has perfomance similar to **SACLag** (overlapping in this case).

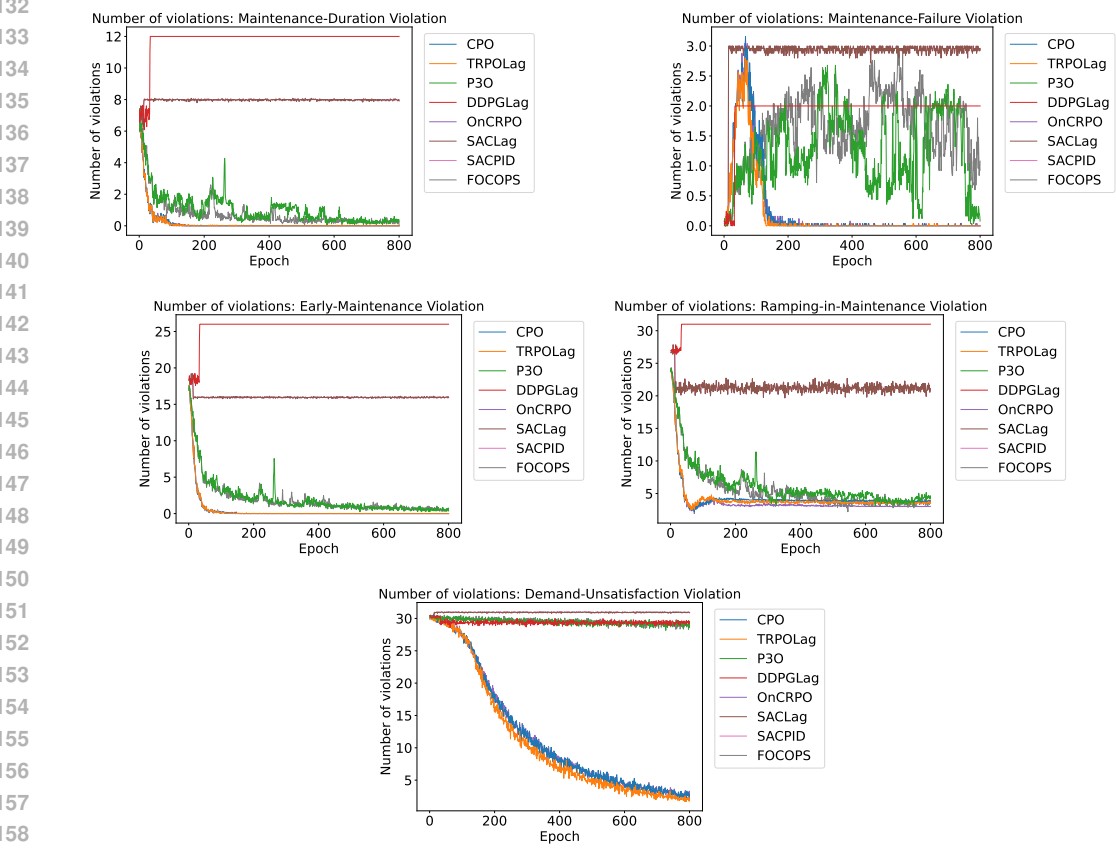

Figure 19: Average number of episode violations for different epochs for SchedMaintEnv-v0

**Integrated scheduling and maintainence environemnt with stochasticity (SchedMaintEnv-v1):** Figure 20 illustrates the mean episode-level breaches per epoch for the stochastic variant of the integrated scheduling and maintenance environment. The figure captures all maintenance-, production-, and demand-related constraints, as in Figure 19. The qualitative performance of the algorithms remains consistent with that observed for SchedMaintEnv-v0.

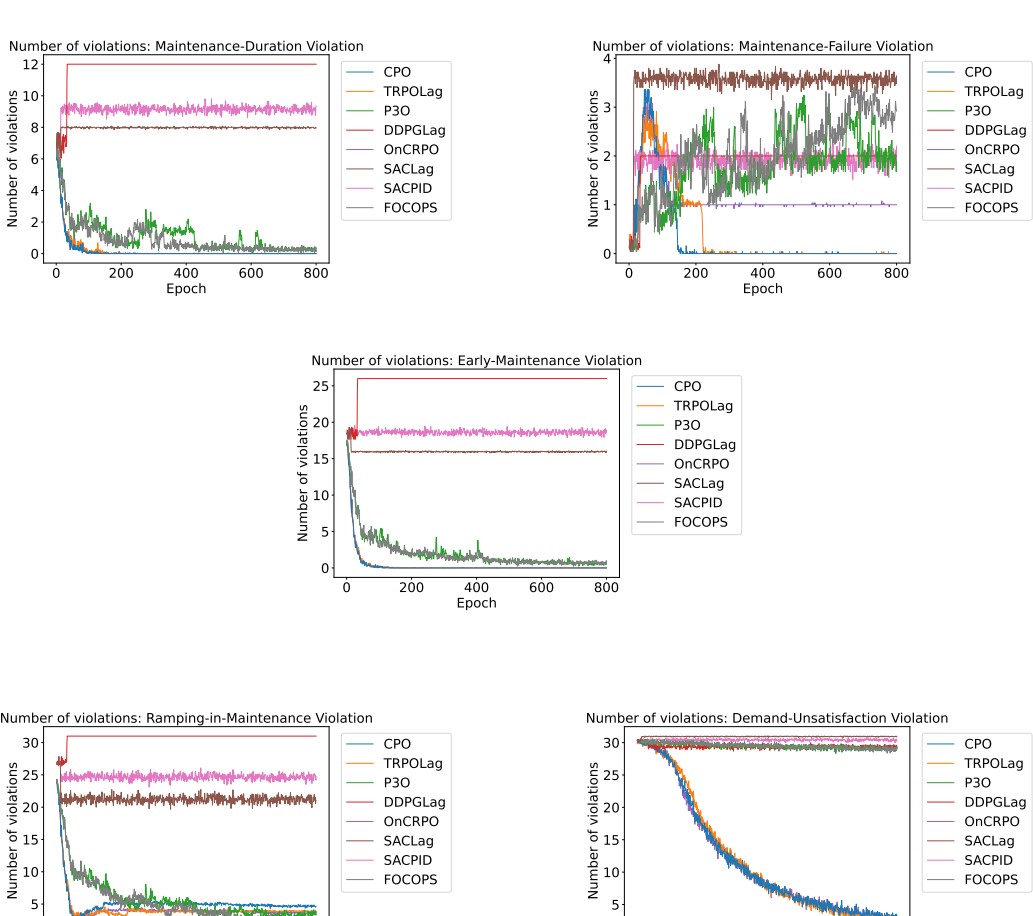

Figure 20: Average number of episode violations for different epochs for SchedMaintEnv-v1

### B.2.9 ASUENV

Figure 21 shows the mean episode-level constraint violations per epoch in the Air Separation Unit environment, focusing on *inventory limits* and *demand satisfaction*. Projection-based methods—**CPO**, **OnCRPO**, and **TRPOLag**—rapidly suppress inventory violations, converging to stable minimal levels (approximately 9 episodes per epoch by around 170 epochs). **P3O** and **FOCOPS** exhibit monotone declines but plateau with substantial residual violations (roughly 16–17) by the end of training. Among off-policy methods, **DDPGLag** remains elevated (near 23) throughout, **SACLag** is consistently the highest and flat (about 26–27), and **SACPID** holds very high counts (around 28) until a late drop near epoch $\sim 180$, after which it stabilizes at roughly 18.

Demand violations emerge only in the latter half of training—after approximately 150–160 epochs—because the agent begins to encounter underproduction violations once it has learned to avoid overproduction. Consequently, methods that tighten inventory most aggressively (**CPO**, **TRPOLag**, **OnCRPO**) exhibit sub-unit average demand-violation rates with intermittent spikes, whereas **P3O**, **FOCOPS**, **DDPGLag**, **SACLag**, and **SACPID** remain near zero—an artifact of maintaining slack inventories rather than superior constraint balancing. Overall, projection-based

methods best reconcile the inventory–demand trade-off; off-policy methods—especially **SACLag** and **DDPGLag**—struggle to reduce inventory violations in ASUEnv, and **SACPID** improves late but does not match the leaders.

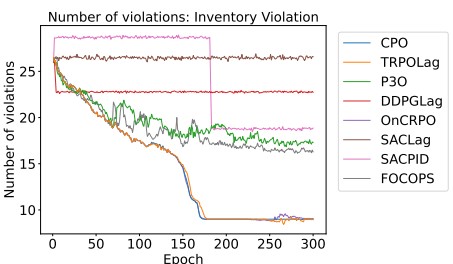 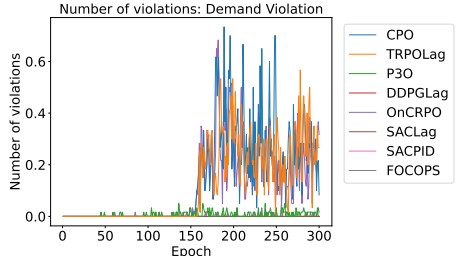

Figure 21: Average number of episode violations for different epochs for ASUEnv

### B.3 BENCHMARKING CLASSICAL RULE-BASED POLICIES IN INVMGMTENV

We evaluated two classical rule-based policies—$(s, S)$ and $(r, Q)$—in `InvMgmtEnv`. Under an $(s, S)$ policy, an order is placed whenever the inventory position falls to or below a reorder point $s$, and the order raises the position to a target level $S$. Under an $(r, Q)$ policy, a fixed lot $Q$ is ordered whenever the position falls to or below a threshold $r$. We impose lower bounds $s \geq s_{\min}$ and $r \geq r_{\min}$ on these triggers; with $s_{\min} = 20$ the $(s, S)$ controller achieves a reward of 7,610.42, while with $r_{\min} = 20$ the $(r, Q)$ controller attains 10,948.9. With the optimal reward being 11265.97493; $(r, Q)$ lies within $\approx 2.8\%$ of the optimum, whereas $(s, S)$ is $\approx 32\%$ below. Relative to the learned SafeRL policies reported for this environment (e.g., ONCRPO $\approx 7,599$, CPO $\approx 7,303$, TRPOLag $\approx 7,198$), the $(s, S)$ baseline is comparable to the best-performing RL result, and the $(r, Q)$ baseline exceeds all RL methods by roughly $40\% - 50\%$. In our 10-episode evaluation for `InvMgmtEnv` (Table 2), SACLAG achieves a reward of $-14,386.99$ with cost 0, SACPID 5,555.74 with cost 0, and FOCOPS $-6,434.03$ with cost 0; hence $(r, Q)$ improves upon SACPID by $\approx 97\%$ and $(s, S)$ exceeds SACPID by $\approx 37\%$, while both rule-based policies obtain higher rewards than SACLAG and FOCOPS.

### B.4 COMPUTATIONAL TIME FOR TRAINING DIFFERENT ENVIRONMENTS WITH VARIOUS ALGORITHMS

Table 4 lists the wall-clock training time **(hours)** required by each algorithm on every environment. All experiments were run on an AWS `g4dn.xlarge` instance—4 vCPUs, 16 GB RAM, and a single NVIDIA T4 GPU.

**Runtime trends across algorithms** The projection–trust-region trio—**TRPO-Lag**, **On-CRPO**, and **CPO**—shows virtually identical runtimes, reflecting their shared on-policy update pattern with lightweight trust-region sub-problems. **P3O** matches this group closely, incurring only a modest overhead for its adaptive penalty update. **FOCOPS** is also in this regime, with runtimes comparable to the projection methods thanks to its focused update structure that avoids heavy critic–actor coupling. By contrast, the off-policy **DDPG-Lag** is consistently the slowest: replay-buffer sampling, twin-critic evaluation, and deterministic actor updates roughly double the wall-clock time relative to the on-policy methods. **SAC-Lag** and **SAC-PID** exhibit similar off-policy behaviour, with entropy-regularised objectives and adaptive dual updates that inflate runtime beyond DDPG-Lag in some environments, though SAC-PID generally yields more stable trajectories at the cost of additional computation. Overall, the results indicate that enforcing safety through on-policy projections or dual updates delivers both strong constraint compliance and favourable computational efficiency, whereas off-policy dual learning trades longer runtimes for greater sample reuse.

Table 4: Wall-clock training time (hours) across environments and algorithms

(a) P3O, DDPGLag, TRPOLag, OnCRPO, CPO

| Environment | P3O | DDPGLag | TRPOLag | OnCRPO | CPO |
|---|---|---|---|---|---|
| RTNEnv | 0.13 | 0.26 | 0.12 | 0.12 | 0.12 |
| STNEnv | 0.13 | 0.26 | 0.13 | 0.13 | 0.13 |
| UCEnv-v0 | 4.66 | 9.18 | 4.43 | 4.43 | 4.46 |
| UCEnv-v1 | 4.58 | 3.71 | 4.44 | 4.44 | 4.43 |
| GTEPEnv | 0.33 | 1.00 | 0.31 | 0.31 | 0.32 |
| BlendingEnv | 1.20 | 3.62 | 1.16 | 1.18 | 1.23 |
| InvMgmtEnv | 0.93 | 2.60 | 0.82 | 0.83 | 0.88 |
| GridStorageEnv | 0.94 | 2.70 | 0.86 | 0.84 | 0.95 |
| SchedMaintEnv-v0 | 0.57 | 1.70 | 0.56 | 0.56 | 0.58 |
| SchedMaintEnv-v1 | 1.49 | 3.35 | 1.45 | 1.51 | 1.42 |
| ASUEnv | 2.38 | 8.80 | 2.42 | 2.39 | 2.42 |

(b) FOCOPS, SACLag, SACPID

| Environment | FOCOPS | SACLag | SACPID |
|---|---|---|---|
| RTNEnv | 0.21 | 0.35 | 0.34 |
| STNEnv | 0.17 | 0.35 | 0.36 |
| UCEnv-v0 | 3.71 | 7.71 | 7.52 |
| UCEnv-v1 | 3.24 | 5.72 | 5.91 |
| GTEPEnv | 0.32 | 1.11 | 0.96 |
| BlendingEnv | 1.11 | 3.49 | 3.51 |
| InvMgmtEnv | 1.25 | 3.07 | 2.67 |
| GridStorageEnv | 1.03 | 2.60 | 2.50 |
| SchedMaintEnv-v0 | 1.45 | 4.02 | 3.60 |
| SchedMaintEnv-v1 | 1.19 | 2.94 | 2.89 |
| ASUEnv | 13.57 | 5.72 | 9.91 |

# C  COMPARISON WITH OTHER OPEN-SOURCE REPOSITORIES

Table 5: Comparison of Gym environments with constraint handling. "Mixed Space" refers to presence of both discrete and continuous variables.

(a) Environment class, constraints, and application domains

| Work | Env. Class | Constraint Handling | Application Domain |
|---|---|---|---|
| OR Gym | Gynasium | Truncation, Reward | Classical OR problems |
| SustainGym | Gynasium | Reward Penalties | Sustainable Energy Systems |
| SafeOR Gym | Gynasium + CMDP wrapper | Truncation, Reward Penalty, Cost | Supply Chain, Chemical Production, Network scheduling, Power Systems |

(b) Compatibility, structural properties, and observation/action sizes

| Work | Compatible with SafeRL Algorithms | Nonconvex constraints | Mixed state action space | Obs / Action (mean, max) |
|---|---|---|---|---|
| OR Gym | ✗ | ✗ | ✓ | (242, 2501) / (57, 200) |
| SustainGym | ✗ | ✗ | ✓ | (79, 150) / (33, 72) |
| SafeOR Gym | ✓ | ✓ | ✓ | (86, 4280) / (32, 272) |

**Discussion.**    Table 5 systematically evaluates four representative open-source reinforcement learning environments that incorporate constraint management. We classify each framework by its API, enforcement mechanism (e.g. truncation, reward/cost penalties, or formal CMDP wrappers), application areas, compatibility with SafeRL algorithm libraries, the dimensionality of observation and action spaces, and support for nonconvex and mixed decision variables.

The key differences of our work compared with

1. **Constraint handling.** OR-Gym and SustainGym mainly rely on simple truncation to enforce basic constraints, such as bounds on state variables. More complex constraints are handled indirectly by assigning negative rewards for violations. SafeOR-Gym uses a more principled approach:

   - For non-safety-critical constraints (e.g., delayed product delivery in a supply chain problem), penalties are applied to the reward signal.
   - For safety-critical or hard physical constraints (e.g., preventing negative inventory levels, which are physically infeasible), SafeOR-Gym introduces explicit costs to guide Safe RL algorithms to rigorously respect these constraints.

   In contrast, the purely reward-based handling in other frameworks can lead to violations of safety-critical constraints.

2. **Environment and algorithm compatibility.** Both OR-Gym and SustainGym are implemented in Gymnasium and are primarily compatible with standard RL algorithms (e.g., PPO, DDPG from Stable-Baselines), treating all problems as unconstrained Markov Decision Processes (MDPs). SafeOR-Gym extends Gymnasium with a Constrained MDP (CMDP) wrapper, enabling compatibility with Safe RL algorithms such as those provided in OmniSafe, which explicitly handle constraints.

3. **Type of constraints.** SafeOR-Gym includes environments with nonlinear, nonconvex constraints and more complex logical relationships between variables. SustainGym focuses on linear and convex constraints, while OR-Gym environments are limited to relatively simple linear constraints.

4. **Problem difficulty.** While the problem dimensions (observation and action space sizes) are of similar magnitude across the frameworks, SafeOR-Gym instances are significantly harder to solve due to their more intricate and realistic constraint structures.

5. **Application domains.** OR-Gym implements classical OR problems such as knapsack and traveling salesman, where well-known greedy heuristics can often provide near-optimal solutions without violating constraints. SafeOR-Gym focuses on more complex, realistic problems where simple heuristics typically violate feasibility constraints. SustainGym targets sustainable energy systems, whereas SafeOR-Gym spans a wider set of domains, including supply chains, chemical production, scheduling, and power systems.

