# OpenReview forum: "SafeOR-Gym: A Benchmark Suite for Safe Reinforcement Learning Algorithms on Practical Operations Research Problems"
_ICLR.cc/2026/Conference — ICLR 2026 Conference Withdrawn Submission_

### Official Review · Reviewer_Jg43 · 2025-10-17

**Soundness:** 2
**Presentation:** 2
**Contribution:** 2
**Rating:** 2
**Confidence:** 3

**Summary:**

The work proposes a new reinforcement learning (RL) benchmark based on operations research (OR) tasks. The main motivation is to provide tools for evaluating RL approaches beyond standard control and robotics domains. In my review, I focused primarily on the relevance of these tasks for RL, as well as on the paper’s presentation quality and reproducibility. I used large language models (LLMs) to polish the writing after crafting the first draft by hand; none of the opinions expressed were artificially generated.

**Strengths:**

The key point is the provision of a set of different tasks with solid and well-documented representations, and strong baselines for comparison. The goal is to provide the community with a testbed beyond robotics and control tasks, which often treat constraint violations as merely negative rewards. The paper is mostly well written and covers similar works available in the literature.

**Weaknesses:**

Although the work is undoubtedly well motivated, several aspects could be improved to enhance clarity and strengthen the contribution.

**Introduction:**
In the second paragraph, the discussion around OmniSafe occupies considerable space and may appear premature. At this early stage, readers who are not already familiar with the framework might find this section confusing, since its relevance to the benchmark’s motivation is not yet established. This content would be better placed later in the paper, perhaps in the methodology or implementation section.

**Results and Motivation:**
The results shown in Figure 1 are a primary concern. In several environments, particularly (f), some RL algorithms fail to outperform what would likely be a randomly initialized policy. In other cases, the convergence is extremely fast, which could reflect limited task complexity, insufficient exploration, or inadequate network architecture.
Moreover, the authors’ explanation that poor performance stems from the nonconvexity of the underlying problems is not entirely convincing. RL is generally robust to nonconvex optimization landscapes; the key issue may instead lie in problem formulation or algorithmic design. This raises a broader question: **if RL methods perform poorly on these tasks, does the benchmark truly represent a suitable or fair testbed for RL research?** The authors should clarify whether the environments are intentionally designed to expose current limitations or whether they are inherently ill-suited for RL-based approaches.

**Technical Content:**
Some detailed technical descriptions currently in the main text could be moved to the appendix. This would help streamline the narrative and maintain focus on the conceptual and empirical contributions.

**Figure 1:**
The figure is difficult to interpret. The overlapping curves, and dense information make it challenging to extract meaningful insights.

**Minor remarks:**
Although code release is not mandatory, providing a supplementary file with the benchmark implementation and trained models would substantially improve transparency and allow reviewers to verify reproducibility.

**Questions:**

**Q1.** The paper emphasizes the importance of handling mixed discrete–continuous action spaces, yet the benchmark evaluation relies primarily on algorithms developed for continuous domains. Could the authors clarify how these algorithms interact with the discrete components of the tasks, and whether any relaxation or encoding strategy was applied?

**Q2.** The paper distinguishes between modelling safety through reward penalties and using explicit cost signals within a CMDP framework. Could the authors elaborate on the conceptual and practical differences between these two formulations, and explain how explicit cost modelling leads to improved learning stability or constraint satisfaction in practice?

**Q3.** Table 1, the authors refer to “classical OR” tasks. Could the authors clarify what qualifies as a “classical” OR task in this context, and how the proposed SafeOR-Gym environments differ from or extend these traditional formulations?

**Details Of Ethics Concerns:**

The paper is not subject to major ethical concerns.

---

### Official Review · Reviewer_t6HV · 2025-10-28

**Soundness:** 2
**Presentation:** 1
**Contribution:** 2
**Rating:** 2
**Confidence:** 3

**Summary:**

The paper proposes SafeOR-Gym, a benchmark suite for safe reinforcement learning in realistic operations and planning settings. Unlike prior safe RL benchmarks that focus on robotics-style continuous control with safety penalties, these environments model industrial decision problems. The environments feature hybrid discrete–continuous actions, long-horizon coupling through forecasts, and hard operational constraints; constraint violations are exposed as explicit cost signals via a CMDP interface compatible with OmniSafe/Gymnasium. The results show that current safe RL methods are still far from being reliable for industrial decision-making.

**Strengths:**

1.The paper systematically evaluates CMDP-based safe RL algorithms in environments with structured constraints, mixed-integer decision structure, and hybrid discrete–continuous actions, spanning planning, power systems, chemical process control, and maintenance scheduling. The results clearly show that many widely used algorithms fail on these tasks.

2.The benchmark suite is implemented to be directly usable: it exposes a CMDP interface through OmniSafe while staying compatible with standard Gymnasium-style RL interfaces. This lowers the barrier for the community to reproduce the experiments and extend them.

**Weaknesses:**

1. The paper devotes a large amount of space to domain-specific operational details of each environment (e.g., industrial process assumptions, power system structure, scheduling rules), but provides relatively little insight into why current CMDP-style algorithms fail. The work reports outcomes (“algorithm X fails here, succeeds there”) but does not analyze which aspects of the tasks (e.g., mixed integers, nonconvex feasible sets, long-horizon credit assignment under feasibility penalties) are responsible for failure, nor how existing safe RL algorithms might need to be modified to handle them.

2. The experimental section mostly presents success/failure across algorithms and tasks, but does not attempt to explain the observed behaviors. For example, why do certain algorithms collapse in BlendingEnv. Without this analysis, it is difficult to extract algorithmic lessons for the safe RL community.

3. It is difficult for a typical RL researcher (without deep OR / energy / process-control background) to judge whether a learned policy is actually making reasonable decisions. All environments are highly domain-specific, and the paper does not provide qualitative rollouts, interpretable visualizations, or sanity checks of agent behavior over time. This limits the accessibility and impact of the benchmark.
4. The paper argues that these environments are realistic and safety-critical, but also states that strong optimization solvers (e.g., Gurobi) can compute globally optimal solutions while strictly satisfying all constraints in the deterministic settings. This weakens the motivation for using RL in these tasks.

5. Many plots appear effectively single-run (visually almost no variance band), and it is unclear how many random seeds were used.

**Questions:**

1. You state that state-of-the-art optimization solvers (e.g., Gurobi) can solve the deterministic nonconvex problems globally while strictly enforcing constraints. Under what conditions is RL actually necessary?

2. In RTNEnv and STNEnv, all safe RL algorithms fail to converge to good solutions. Should we interpret this as evidence that these tasks are fundamentally out of reach for current policy-gradient methods? Or do you believe the tasks are learnable with RL but require different training schemes. Why is PPO not explicitly included as a baseline for “can vanilla RL solve this at all,” independent of safety?

3. How many random seeds were used for Figure 1 and the other reported learning curves? Some curves (e.g., Figure 1(a)) appear to have essentially zero visible variance. Please report the number of seeds and whether shaded regions represent standard deviation or standard error.

4. The benchmark spans very different domains. How did you validate that the state transitions, action constraints, and reward/cost definitions are implemented correctly in each environment? From an RL researcher’s perspective, it is difficult to audit the correctness of these environments without deep domain expertise.

5. In several plots in Figure 1, some algorithms’ reward curves become almost perfectly flat very early in training. Please explain what causes these early plateaus.

6. In Figure 1(f) (BlendingEnv), most algorithms show reward collapse and large fluctuations.  Any insight here would help clarify what specifically makes this environment challenging.

7. How you define the cost threshold for each task?

---

### Official Review · Reviewer_h9oT · 2025-10-29

**Soundness:** 3
**Presentation:** 3
**Contribution:** 2
**Rating:** 4
**Confidence:** 4

**Summary:**

This paper introduces SafeOR-Gym, a benchmark suite of nine operations research (OR)–inspired environments designed to evaluate safe reinforcement learning (RL) algorithms. The environments model structured decision problems such as unit commitment, scheduling, blending, and inventory management, each formulated as a Constrained MDP and fully compatible with the OmniSafe framework. The authors benchmark several state-of-the-art safe RL algorithms (CPO, TRPOLag, P3O, OnCRPO, DDPGLag) and compare their performance against optimal solutions obtained with mathematical programming solvers such as Gurobi.

**Strengths:**

The main strength of this work lies in its attempt to model complex and structured OR problems within a safe RL framework. The environments are well-motivated, grounded in realistic operational settings, and go beyond traditional control benchmarks like Safety Gym. Integrating these tasks with OmniSafe provides immediate usability and reproducibility, which makes SafeOR-Gym a valuable contribution to the community.

The paper also highlights an important empirical finding: most current safe RL algorithms struggle to learn effective policies when faced with hybrid discrete–continuous decisions and nonconvex constraints. This observation underlines the benchmark’s relevance as a stress test for future algorithmic advances.

**Weaknesses:**

The structure of the paper is not fully clear. While nine environments are introduced, only a few (two or three) are described in sufficient detail in the main text, with the rest delegated to the supplementary material. A concise comparative summary (e.g., a table of environment sizes, horizon lengths, constraint types, and stochasticity) would help readers understand their diversity and modeling differences.

Moreover, although the authors refer to these tasks as real-world, most environments are fully deterministic, with only the blending task including a stochastic component. This weakens the “realistic” claim, since real operational systems usually involve uncertainty in demand, resource availability, or failures. Including stochastic variants would strengthen the benchmark’s practical realism.

The results section shows that most algorithms struggle to reach satisfactory policies. While this is an important observation, the paper could provide a deeper analysis of why this occur

Finally, I would have appreciated a clearer discussion on what makes each environment “real-world” beyond the fact that it represents a classical OR problem. Highlighting the specific characteristics that differentiate these environments from standard RL benchmarks would make the contribution stronger.

Missing code link in paper (find it on google)

**Questions:**

Could you provide a concise comparative summary of all nine environments (e.g., action/observation dimensions, horizon, constraint types, stochasticity) to help readers better understand their diversity and complexity?

Since most environments are currently deterministic, do you plan to introduce stochastic variants (e.g., uncertain demand or resource availability) to better reflect real-world operational uncertainty?

You note that most algorithms struggle to learn effective policies—could you elaborate on the underlying causes (e.g., infeasibility, gradient instability, constraint scaling issues) and whether any diagnostics were performed?

Could you clarify what specific structural or operational features (e.g., mixed-integer decisions, coupling over time, nonconvex safety constraints) make these environments representative of practical OR systems rather than abstract RL benchmarks?

---

### Official Review · Reviewer_VMo5 · 2025-10-31

**Soundness:** 3
**Presentation:** 3
**Contribution:** 3
**Rating:** 6
**Confidence:** 3

**Summary:**

SafeOR-Gym presents a suite of nine operations-research (OR) inspired environments (e.g., RTN, STN, UC, GridStorage, Blending, InvMgmt, SchedMaint, ASU, GTEP) implemented as Gym/CMDP environments with native OmniSafe compatibility and explicit cost signals for constraint handling. The paper benchmarks a range of safe-RL algorithms (CPO, TRPOLag, P3O, OnCRPO, DDPGLag, SAC variants, FOCOPS, etc.), compares to optimization baselines (Gurobi) where applicable, and reports that many current safe-RL methods struggle on these structured OR problems.

**Strengths:**

- **Well-motivated contribution:** The authors target an important mismatch: popular safe-RL benchmarks are control/robotics focused, while many real safety problems are OR problems with mixed-integer structure and long horizons. SafeOR-Gym is clearly targeted to that gap.
- **Implementation quality:** Environments expose explicit cost channels (CMDP wrapper) rather than only reward penalties, making them directly useful for constraint-aware algorithms in OmniSafe and similar toolkits.
- **Empirical insight:** The results clearly show that most safe RL algorithms fail to handle mixed-integer or nonconvex constraints effectively, a useful diagnostic for future algorithmic design.
- **Comparative positioning:** The paper situates itself well relative to OR-Gym and SustainGym, emphasizing constraint modeling and CMDP compliance
- **Potential impact:** If released and maintained, SafeOR-Gym could become a standard benchmark for the intersection of operations research and RL.

**Weaknesses:**

- **Envirnonment diversity, but overlapping algorithmic challenges:** The nine environments span a good range of OR-inspired domains, from power systems to scheduling and process control. However, several share similar underlying learning characteristics — such as long-horizon decision-making, hybrid discrete–continuous actions, and feasibility-based safety constraints. For instance, RTN and STN represent closely related scheduling formulations. As a result, while the suite is cohesive and well-scoped for evaluating this particular class of problems, it may capture a narrower slice of safe-RL challenges (e.g., limited stochasticity, partial observability, or risk-based safety criteria). Expanding future versions to include such variations could further broaden its applicability.
- **Lack of real-world data and realism:** The environments rely on JSON instance files with synthetic parameters, but no real-world datasets or data provenance are provided. Moreover, most environments are deterministic, with stochasticity introduced only in one variant. This limits the practical realism claimed in the paper.
- **Reproducibility:** The main text does not provide a public code link, which hinders verification and community adoption (assuming it exists online).

**Questions:**

- I suggest to add a compact taxonomy table in the main text summarizing the 9 environments along axes: domain, obs/action sizes, discrete vs continuous decisions, presence of integer constraints, stochasticity, horizon length, and canonical safety constraints. This will immediately clarify diversity claims.
- Provide hyperparameter search ranges, seeds, evaluation episode counts, and computational budgets. Moreover, the appendix is really dense: a tighter edit would help.
- Include diagnostic analyses (e.g., constraint-violation breakdowns or action-feasibility rates) to better explain algorithmic failures.
- I would suggest selecting Benchmark and Datasets as the primary submission area.

---

### Note · Authors · 2025-11-18

I have read and agree with the venue's withdrawal policy on behalf of myself and my co-authors.